# H3.1K27me1 loss confers *Arabidopsis* resistance to Geminivirus by sequestering DNA repair proteins onto host genome

Zhen Wang [1,2], Claudia M. Castillo-González [1], Changjiang Zhao [1], Chun-Yip Tong[1], Changhao Li [1], Songxiao Zhong[1], Zhiyang Liu[1], Kaili Xie [1], Jiaying Zhu [1], Zhongshou Wu [3,4], Xu Peng [5], Yannick Jacob [6], Scott D. Michaels [7], Steven E. Jacobsen [3,4] & Xiuren Zhang [1,2,8] ✉

The H3 methyltransferases ATXR5 and ATXR6 deposit H3.1K27me1 to heterochromatin to prevent genomic instability and transposon re-activation. Here, we report that *atxr5 atxr6* mutants display robust resistance to Geminivirus. The viral resistance is correlated with activation of DNA repair pathways, but not with transposon re-activation or heterochromatin amplification. We identify RAD51 and RPA1A as partners of virus-encoded Rep protein. The two DNA repair proteins show increased binding to heterochromatic regions and defense-related genes in *atxr5 atxr6* vs wild-type plants. Consequently, the proteins have reduced binding to viral DNA in the mutant, thus hampering viral amplification. Additionally, RAD51 recruitment to the host genome arise via BRCA1, HOP2, and CYCB1;1, and this recruitment is essential for viral resistance in *atxr5 atxr6*. Thus, Geminiviruses adapt to healthy plants by hijacking DNA repair pathways, whereas the unstable genome, triggered by reduced H3.1K27me1, could retain DNA repairing proteins to suppress viral amplification in *atxr5 atxr6*.

Geminiviruses, a group of single-stranded circular DNA viruses, are increasing threats to worldwide crop yield and food security due to their broad host range[1,2]. Geminiviruses pack their DNA with host-encoded histone octamers to form episomes that are named the viral mini-chromosomes. Mini-chromosomes serve as intermediates for replication via rolling circle replication (RCR) and recombinant-dependent replication (RDR)[3]. Epigenetic modifications, including histone methylation, regulate various development processes and plant defense responses against microbes such as plant viruses[4,5]. For example, kryptonite (KYP)/SUVH4 catalyzes deposition of H3K9me2/3 on viral mini-chromosome to attenuate virus accumulation, whereas

Geminivirus-encoded transcriptional activation protein (TrAP/AL2) can impair KYP activity to counter the host defense[4]. Recently, various histone marks such as H4K3me3, H3K36me3, and H3K27me3 have been detected on Geminivirus mini-chromosomes[6,7], suggestive of regulatory roles for histone methylation in plant defense against Geminiviruses.

Arabidopsis Trithorax-Related Proteins 5 (ATXR5) and ATXR6 (ATXR5/6) redundantly catalyze K27 monomethylation specifically on the H3.1 variant (H3.1K27me1), and depletion of this mark mainly impacts heterochromatin[8–10]. Reduced levels of H3.1K27me1 in heterochromatic regions cause increased amount of H3.1K27ac

[1]Department of Biochemistry and Biophysics, Texas A&M University, College Station, TX 77843, USA. [2]Molecular and Environmental Plant Sciences, Texas A&M University, College Station, TX 77843, USA. [3]Department of Molecular, Cell, and Developmental Biology, University of California, Los Angeles, Los Angeles, CA 90095, USA. [4]Howard Hughes Medical Institute, University of California, Los Angeles, Los Angeles, CA 90095, USA. [5]Department of Molecular Physiology, College of Medicine, Texas A&M University, College Station, TX 77843, USA. [6]Department of Molecular, Cellular & Developmental Biology, Yale University, New Haven, CT 06511, USA. [7]Department of Biology, Indiana University, Bloomington, IN 47405, USA. [8]Department of Biology, Texas A&M University, College Station, TX 77843, USA. ✉e-mail: xiuren.zhang@tamu.edu

(acetylation) and H3.1K36ac, leading to transcriptional activation of transposable elements (TEs)[9,11]. Reduction of H3.1K27me1 also causes DNA amplification in heterochromatic regions of the genome[9,10]. DNA duplication in *atxr5 atxr6* is associated with the accumulation of DNA double-strand breaks (DSBs)[8,9,12,13]. DSBs can be repaired through homologous recombination repair (HRR) and non-homologous end joining (NHEJ) in eukaryotic cells. It has been found that HRR factors such as *Arabidopsis* radiation sensitive 51 (*RAD51*), breast cancer susceptibility gene 1 (*BRCA1*), and B1 type cyclin-dependent protein kinase (*CYCB1;1*) are transcriptionally upregulated in *atxr5 atxr6*[12-14]. Intriguingly, a genetic screen based on the readout of *RAD51* promoter-driven GFP in the *atxr5 atxr6* background recovered loss of function mutants of *METHYL-CpG BINDING DOMAIN PROTEIN 9* (*MBD9*) and *Yeast SAC3 HOMOLOG B* (*SAC3B*) as suppressors of *atxr5 atxr6* phenotypes. These lines provide an opportunity to investigate the mechanisms underlying the distinct molecular phenotypes of *atxr5 atxr6*[13,15].

Recent work has revealed that H3.1 interacts with TONSOKU (TSK), which is required for initiating HRR during replication to resolve stalled/broken replication forks and maintain genomic stability[16]. Interestingly, inactivation of TSK in *atxr5 atxr6* mutants suppresses heterochromatin amplification, whereas deletions of RAD51 and BRCA1 enhance this phenotype[13,16]. These results suggest that the roles of different HRR proteins in protecting genome stability may be distinct. Furthermore, there is still a gap in our understanding of the interplay between H3.1K27me1 depletion, heterochromatin amplification, and HRR.

A growing body of evidence highlights the involvement of HRR factors in viral DNA replication in human[17-22]. In plants, the roles of HRR factors in geminiviral propagation are perceived in different ways. Whereas proliferating cell nuclear antigen (PCNA) suppresses the enzyme activity of Rep, RAD54 promotes Rep function in vitro. Moreover, deficiency of PCNA or RAD51D impairs Geminivirus accumulation, but deletion of RAD54 or RAD17 does not affect infection[23-27]. The contrasting reports above indicate that the roles of plant HRR factors in viral DNA replication remain unclear. Of note, when the plant innate immune response is triggered by salicylic acid (SA), RAD51 can directly bind to promoter elements of defense genes and enhance gene expression in a BRCA2- and SA-dependent manner. Moreover, *RAD17* and *Rad-3-related* (*ATR*) are required to enhance the expression of SA-activated genes and deploy an effective immune response[28,29]. It has been reported that Cabbage Leaf Curl Virus (CaLCuV) infection can induce the expression of genes in the SA pathway[30], which in turn seesaws the battle between Geminiviruses and plants[31,32]. These results suggest HRR might regulate the plant defense against Geminiviruses through the innate immune response.

We have recently surveyed the viral susceptibility of numerous epigenetic mutants of *Arabidopsis*. Surprisingly, we found that *atxr5 atxr6* double mutants behaved differently and displayed a striking resistance to CaLCuV. Depletion of *SAC3B*, *MBD9*, or *BRCA1* in *atxr5 atxr6* restored susceptibility of the viral infection despite contrasting effects on TE reactivation and heterochromatin amplification. Transcriptome-wide association studies (TWAS) showed that reduced viral DNA replication correlated with upregulation of HRR-related genes in the mutants. We found that the viral protein Rep hijacked host RAD51 and replication protein A 1A (RPA1A) on the viral genome to promote viral amplification. Interestingly, RAD51 and RPA1A showed increased binding to unstable genomic DNA (e.g., rDNA and noncoding RNA (ncRNA) loci) in *atxr5 atxr6* vs Col-0. Moreover, RAD51 was enriched at plant defense genes, and its binding was coupled with their transcriptional upregulation in *atxr5 atxr6* upon viral inoculation. Additionally, we found that BRCA1, HOP2, and CYCB1;1 recruited RAD51 onto the host genome, and deletion of these factors restored the susceptibility of *atxr5 atxr6* to geminiviral infection. Thus, we propose that increased unstable genomic DNA, together with enhanced expression of defense-related gene loci, sequesters RAD51

via BRCA1, HOP2, and CYCB1;1 to prevent the loading of HRR factors onto viral genome, leading to poor viral amplification in *atxr5 atxr6*. This study provides a new idea to manipulate the routing of plant HRR factors to defend viral infection to improve agricultural traits.

## Results

### The *atxr5 atxr6* mutant displays increased resistance to CaLCuV inoculation

To investigate the effect of epigenetic modifications on viral pathogenesis, we inoculated numerous mutants in epigenetic silencing pathways of *Arabidopsis* with CaLCuV. The symptoms induced by CaLCuV-inoculation included chlorosis, curled leaf and plant growth arrest (Supplementary Fig. 1). Loss-of-function mutants of H3K9me2/3 methyltransferases (MTases) (*suvh4/5/6*), H3K27me3 MTases in Polycomb Repressive Complex 2 (PRC2) (*clf-28*) and DNA MTases (*drm1 drm2 cmt3*) displayed increased viral susceptibility compared to wild type (WT) Col-0, implying their critical roles in inhibiting viral propagations in plants (Supplementary Fig. 1). In contrast, inoculation of *atxr5 atxr6* (with compromised H3.1K27me1 deposition) resulted in significantly fewer infected plants. When infected, *atxr5 atxr6* also displayed milder symptoms compared to either of the single mutants of *atxr5* and *atxr6* or Col-0 (Fig. 1a, b). These results indicated that *atxr5 atxr6* did not offer a permissible environment for Geminivirus propagation and/or replication (Fig. 1a, b).

As the infection was conducted through agrobacteria-mediated infiltration, we first examined whether the initial plasmid delivery in planta was compromised in *atxr5 atxr6*. We collected inoculated plants at 3, 6, 9, and 13 days post inoculation (dpi). Southern blot, semi-qPCR, and qPCR results showed that the amount of delivered plasmids in Col-0 and *atxr5 atxr6* was comparable at 3, 6, 9, and 13 dpi. In contrast, the amount of replicated viral DNA was strikingly lower in *atxr5 atxr6* vs. Col-0 starting at 13 dpi (Supplementary Fig. 2). These results indicated that viral DNA amplification rather than plasmid transfection was suppressed in *atxr5 atxr6*.

ATXR5 and ATXR6 deposit K27me1 specifically on the replication-dependent H3.1 variant, which prevents heterochromatin amplification[10]. H3.1 is encoded by five genes in *Arabidopsis thaliana*, and inactivation of H3.1 in plants leads to sterility and strong pleiotropic phenotypes[10,16]. Deletion of *FASCIATA2* (*FAS2*), which encodes a subunit of CHROMATIN ASSEMBLY FACTOR 1 (CAF1), prevents the normal deposition of H3.1 during replication[10]. Of note, while *fas2* and *h3.1* mutants lose both H3.1K27me1 and the H3.1 variant, the *atxr5 atxr6* mutants only display a reduced level of H3.1K27me1 without major changes to H3.1 deposition[10]. To examine the impact of H3.1K27me1 deficiency on viral replication, we challenged *fas2* and numerous hypomorphic *H3.1* mutants with CaLCuV. The *fas2* plants showed a more severe yellow mosaic phenotype than Col-0, whereas the H3.1 single mutants and quadruple mutants showed similar infection ratio and symptom severity to those in Col-0 (Supplementary Fig. 3). Of note, inactivation of *FAS2* or *H3.1* genes did not result in any defect on heterochromatic DNA stability due to the concurrent loss of H3.1K27me1 and H3.1[10,16]. These results imply that concomitant reduction of H3.1K27me1 and H3.1 does not mimic the suppression of viral DNA amplification observed in *atxr5 atxr6* where only H3.1K27me1 is reduced.

### Viral resistance of *atxr5 atxr6* is not directly related to TE reactivation and heterochromatic DNA amplification

The *atxr5 atxr6* mutant has three main molecular phenotypes: TE reactivation, heterochromatin amplification, and activation of DNA repair pathways[10,13,33]. It has been reported that loss of function mutations of *MBD9* or *SAC3B* in *atxr5 atxr6* rescue the loss of H3.1K27me1, and suppress heterochromatin amplification and TE re-activation[13,15]. By contrast, loss of function mutation of *BRCA1*, a gene involved in replication fork stability and DNA repair, enhances

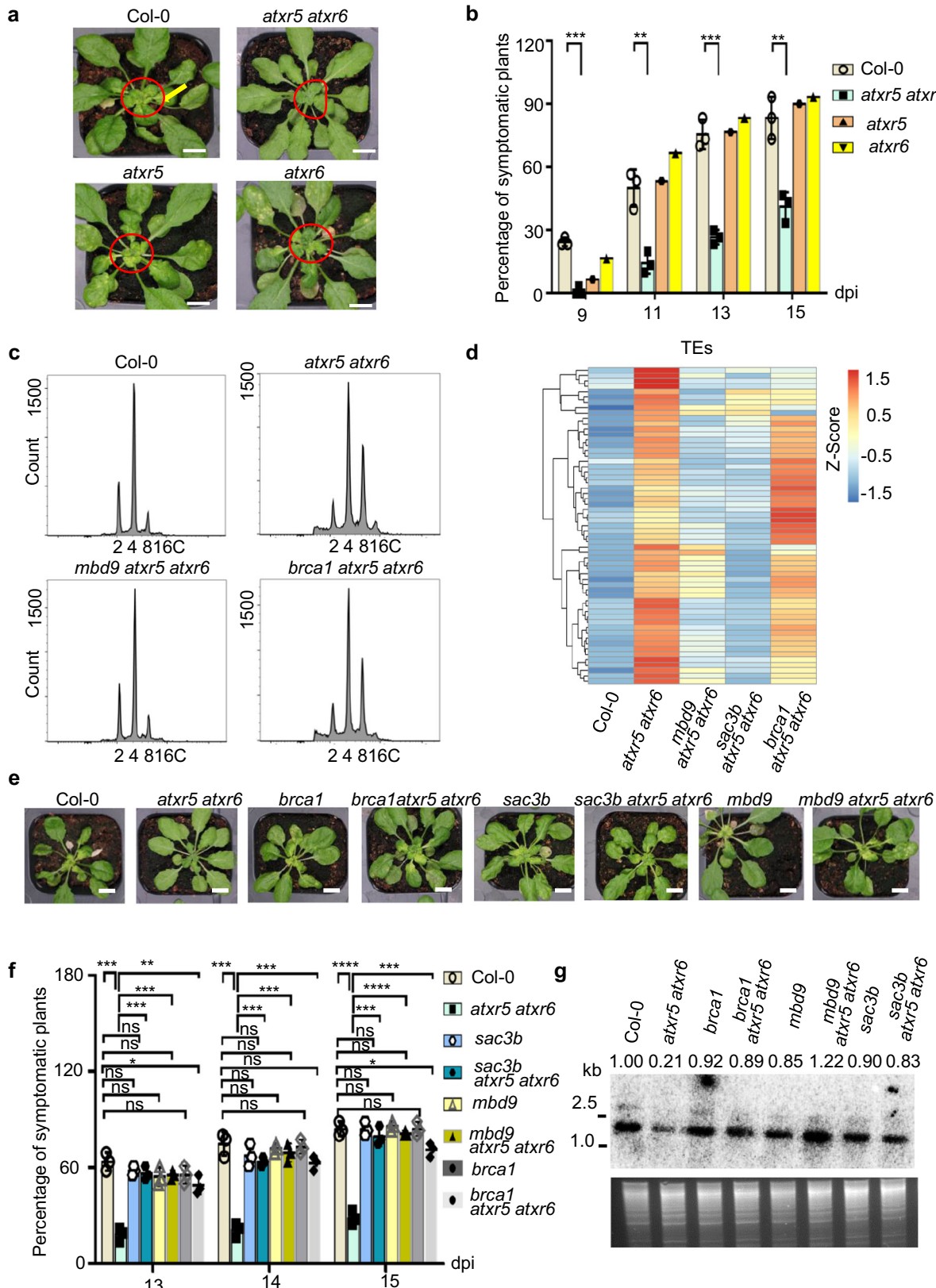

both heterochromatin amplification and TE re-activation in *atxr5 atxr6*[13,15]. We revisited these experiments using leaves #1-#6 of five-week-old mock-treated plants and obtained similar results as the previous report, which used cotyledons in their experiments[13] (Fig. 1c, d and Supplementary Fig. 3g). These triple mutant lines provided an opportunity to investigate the relevance of distinct

molecular phenotypes of *atxr5 atxr6* with geminiviral pathogenesis in plants.

We inoculated *mbd9, sac3b, brca1, atxr5 atxr6, mbd9 atxr5 atxr6, sac3b atxr5 atxr6, brca1 atxr5 atxr6,* and Col-0 with CaLCuV. Mutations of *BRCA1, MBD9,* or *SAC3B* in the wild-type background did not affect the plant viral susceptibility (Fig. 1e–g, and Supplementary Fig. 4a–c).

**Fig. 1 | *atxr5 atxr6* displays robust viral resistance phenotype that is superficially uncoupled with re-activation of TEs and DNA replication. a** Loss-of-function mutants of *atxr5 atxr6* show resistance to CaLCuV. Photographs of CaLCuV-inoculated Col-0, *atxr5*, *atxr6*, and *atxr5 atxr6* plants at 16 dpi (days post-inoculation). Scale bars, 1 cm. **b** Percentages of symptomatic plants induced by CaLCuV-inoculation at 9, 11, 13, and 15 dpi. Each dot in the bar plot represents one replicate, the inoculation experiments were performed with 30 plants/replicate with 6 mock-treated plants as a control. Data are presented as mean ± SD (*n* = 3 biological replicates for Col-0 and *atx5 atxr6*; n = 1 for *atxr5* and *atxr6*). The experiments were repeated three times with similar results. **c** Flow cytometry assay shows that loss of *MBD9* but not *BRCA1* could rescue DNA re-replication phenotype in *atxr5 atxr6*. **d** Heat map shows that loss of *MBD9*, or *SAC3B*, but not BRCA1, could suppress transcriptional re-activation of TEs in *atxr5 atxr6*. The quantification was conducted by DESeq2. **e** The loss of *MBD9*, *SAC3B*, or *BRCA1* could all increase viral susceptibility of *atxr5 atxr6*. Photographs of CaLCuV-inoculated plants were taken at 15 dpi. Scale bars, 1 cm. **f** Percentages of symptomatic plants in different backgrounds induced by CaLCuV- inoculation at 13, 14, and 15 dpi. Each dot in the bar plot represents one replicate, experiments were performed with 36 plants/replicate. Data are presented as mean ± SD (n = 3 biological replicates). **g** Southern blot assay shows differential accumulation of viral DNA A in CaLCuV-inoculated plants with different genotypes at 15 dpi. Experiments were repeated twice with similar results. The titers of viral DNA A were first normalized with the loading control (EcoRI-digested input DNA, Bottom panel), and then to Col-0 where the amount was arbitrarily set as 1. Statistics in Fig. 1b, f were performed with unpaired two-tailed student t-test, *, **, *** and ****, *P* < 0.05, 0.01, 0.001, and 0.0001, respectively. Source data are provided in the Source Data File.

Intriguingly, *mbd9 atxr5 atxr6*, *sac3b atxr5 atxr6*, and *brca1 atxr5 atxr6* mutants all showed significantly higher percentages of symptomatic plants with severe chlorosis compared to *atxr5 atxr6* (Fig. 1e–g, and Supplementary Fig. 4a–c). Consistently, titers of viral DNA in the three triple mutants were higher than the amount in *atxr5 atxr6* (Fig. 1g, Supplementary Fig. 4c). Thus, the susceptibility of *brca1 atxr5 atxr6*, *mbd9 atxr5 atxr6* and *sac3b atxr5 atxr6* mutants was attributed to the specific effect of the loss of *BRCA1*, *SAC3B* and *MBD9* on molecular features of *atxr5 atxr6*. Given that three triple mutants had similar viral infection profiles but displayed different effects on the heterochromatin amplification and TE re-activation (Fig. 1c, d, Supplementary Fig. 4d, and Supplementary Table1), thus, heterochromatin amplification and TE re-activation were not directly responsible for Geminivirus resistance in *atxr5 atxr6* mutants. In line with this result, *suvh4/5/6* and *drm1 drm2 cmt3* mutants that have TE re-activation[33] showed hyper-susceptibility to viral infection (Supplementary Fig. 1).

### Viral resistance of *atxr5 atxr6* is coupled with the enhanced expression of genes involved in DNA repair

To pinpoint the genetic pathways that attributed to viral resistance of *atxr5 atxr6*, we mined public RNA-seq data of *mbd9 atxr5 atxr6*, *sac3b atxr5 atxr6*, *atxr5 atxr6* and Col-0 from a cotyledon stage to perform transcriptome-wide association studies (TWAS)[13]. Mutations of *MBD9* and *SAC3B* suppress enhanced expression of 240 protein-coding genes in *atxr5 atxr6* (Supplementary Fig. 4e, f). Gene ontology (GO) analysis reveals that significant enriched biological processes belonged to immune system process, response to virus and DNA repair (Supplementary Fig. 4g). These three pathways might individually or synergistically contribute to viral resistance of *atxr5 atxr6*.

To investigate how the transcriptome is reprogramed upon virus infection, we performed comprehensive TWAS with high-quality reads (Supplementary Fig. 5, quality score > 30). When the samples from mock and virus inoculation treatments were considered, approximately 4800 differentially expressed genes (DEGs, fold change ≥ 2, FDR ≤ 0.05) were recovered (Supplementary Fig. 6a–c). Among the DEGs, 1136 genes showed enhanced expression upon virus inoculation (Supplementary Fig. 6d, e). Among the virus inoculation-activated genes, we selected 365 genes that were expressed at higher levels in *atxr5 atxr6* compared to Col-0, *brca1 atxr5 atxr6*, *mbd9 atxr5 atxr6* and *sac3b atxr5 atxr6*, in both mock-treated and virus-inoculated samples (Fig. 2a). GO analysis classified the top three enriched biological processes into DNA damage response (DDR,19 genes), DNA repair (17 genes, included in the list of 19 DDR genes) and DNA recombination (12 genes) (Fig. 2b and Supplementary Table 2). Of note, the 18 of 19 DDR genes and 16 of 17 genes related to DNA repair were upregulated in Col-0 upon virus inoculation (Supplementary Fig. 6f). A similar result was also observed in an early microarray assay[30]. Importantly, the expression of DDR genes induced by virus inoculation was further enhanced in *atxr5 atxr6*; and these DDR genes including *HOP2*, *CYCB1*, and *BRCA1* belong to HRR rather than NHEJ (Fig. 2c)[34]. Overall, TWAS revealed a significant association between viral resistance of *atxr5 atxr6* and enhanced expression of DDR genes.

### DDR factors are required for efficient amplification of viral genome

In our study, virus inoculation activated the expression of 53 genes related to DDR in *atxr5 atxr6*. Among them, Ataxia-telangiectasia mutated (ATM) and ATR are the kinases that redundantly associate with the majority of DDR factors activated during Geminivirus inoculation (Fig. 2d). One gene, the suppressor of gamma response 1 (SOG1), can govern transcriptional activation of many DDR factors[35]. To decipher the relationship between geminiviral amplification and DDR activation in plants, we performed virus inoculation assays with Col-0, *atm*, *atr*, *sog1*, *atxr5 atxr6*, and *atm atxr5 atxr6* plants. Unlike *sog1*, *atm*, and *atr* showed reduced ratios of symptomatic plants, milder symptoms and less viral DNA compared to Col-0 (Supplementary Fig. 7a–c; Fig. 2e–g). Remarkably, no significant difference was observed in the ratio of symptomatic plants and the amount of viral DNA between *atxr5 atxr6 atm* and *atxr5 atxr6* (Fig. 2f, g). This epistatic phenotype suggests that *atm* and *atxr5 atxr6* function in the same pathway in relation to viral DNA amplification.

### Rep recruits DNA repair proteins to facilitate viral DNA replication

Since ATM is related to HRR, we hypothesized that some HRR factors might directly promote viral DNA amplification. To test this, we conducted yeast two-hybrid (Y2H) screening of 18 selected DDR factors using Rep (Fig. 3a and Supplementary Fig. 7d), the essential viral replication protein, as a bait. Y2H screening recovered RPA1A, RAD51, and PCNA1 as binding partners of Rep (Fig. 3a and Supplementary Fig. 7d). We validated the interaction of Rep with RPA1A and RAD51 through co-immunoprecipitation (Co-IP) experiments in *N. benthamiana* (Fig. 3b). We hypothesized that if RPA1A and RAD51 were recruited to viral DNA to facilitate viral amplification, they should associate with the viral mini-chromosomes. Indeed, chromatin immunoprecipitation-qPCR (ChIP-qPCR) readily detected the enrichment of RPA1A and RAD51 on the viral genome (Fig. 3c).

We also found that the expression of *RAD51* and *RPA1A* was upregulated in *atxr5 atxr6* vs. WT and further increased upon virus inoculation in both backgrounds (Fig. 3d). Importantly, mutations in *RAD51* or *RPA1A* resulted in lower ratio of symptomatic plants and reduced viral DNA accumulation compared to Col-0 (Fig. 3e–g). On the other hand, mutations of other HRR factors such as *HOP2*, *BRCA1*, or *CYCB1* did not affect the viral pathogenesis (Supplementary Fig. 7e, f). These results indicate that RAD51 and RPA1A are essential for Geminivirus amplification in *Arabidopsis*. Remarkably, loss of *RAD51* did not have an additive effect on the viral resistance phenotype of *atxr5 atxr6* (Fig. 3f, g). Given that *atxr5 atxr6* and *atxr5 atxr6 rad51* had the same viral resistance phenotype, we concluded that RAD51 and RPA1A were downstream effectors that accounted for reduced viral amplification in *atxr5 atxr6*.

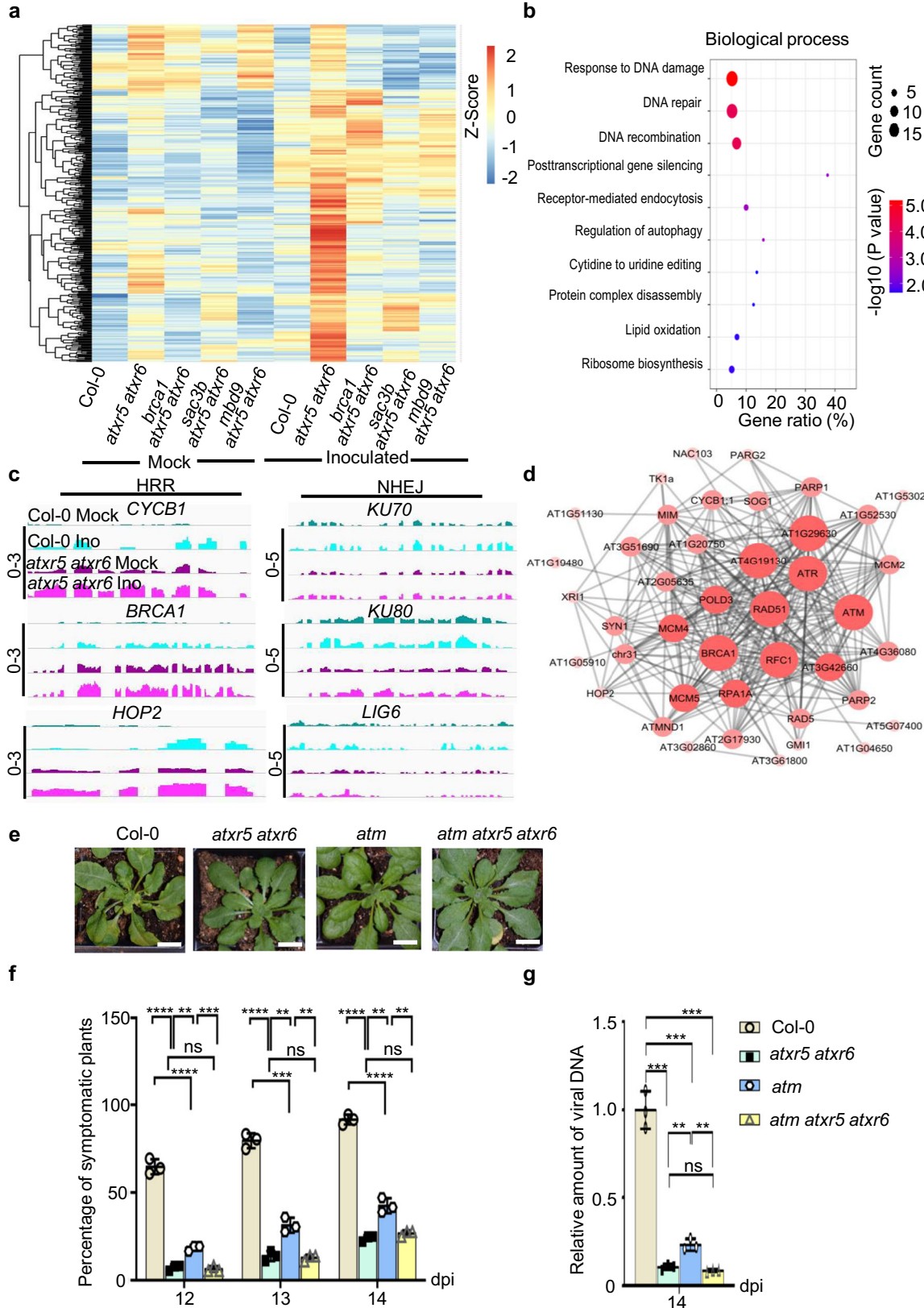

## Over-replicated *45S rDNA* in *atxr5 atxr6* recruits more RAD51 and RPA1A to the loci

A recent study shows that TSK, one upstream HRR factor, preferentially binds to the replication-dependent H3 variant H3.1 to repair the DSBs[16]. Increased DSBs and RAD51 foci have been observed in over-replication associated centers (RACs), a structure observed during

remodeling of heterochromatin, suggesting that RAD51 is involved in DNA repair in *atxr5 atxr6*[12].

We next assessed the distribution of RAD51 and RPA1A signal over the genome in Col-0 and *atxr5 atxr6*. We first validated the specificity of anti-RAD51 and RPA1A antibodies (Supplementary Fig. 8a–f), and then performed genome-wide chromatin immunoprecipitation-sequencing

**Fig. 2 | Homology directed repair (HDR) pathway contributes to viral resistance of *atxr5 atxr6*. a** Heat map shows the accumulation change of 365 transcripts, selected from a total of 4800 DEGs based on the clustering analysis, in mock-treated and virus-inoculated Col-0, *atxr5 atxr6*, *sac3b atxr5 atxr6* and *mbd9 atxr5 atxr6*. The quantification is conducted by DESeq2. **b** Bubble plots from Gene Ontology (GO) analysis show the enrichment of 365 genes from **2a** in different biological processes. **c** IGV files show changes of transcript levels of indicated genes in mock-treated and virus-inoculated (Ino) Col-0 and *atxr5 atxr6* (Normalized by RPKM). Scales for the distinct loci were shown in left as solid lines. Ino, Inoculated. **d** Protein-protein interaction (PPI) network of DNA repair-related proteins encoded by DEGs upon the virus infection. **e** Representative phenotypes of CaLCuV-inoculated Col-0, *atxr5 atxr6*, *atm*, atm *atxr5 atxr6*. Photographs were taken at 14 dpi. Scale bars, 1 cm. **f** Percentages of symptomatic plants induced by CaLCuV-inoculation in indicated backgrounds at 12, 13, and 14 dpi. **g** q-PCR shows the relative amount of viral DNA A in CaLCuV-inoculated plants in indicated backgrounds at 14 dpi. The relative amount of viral DNA A was first normalized to *UBQ10* control, and then to that of Col-0 where the mean was arbitrarily assigned a value of 1. In (**f**) and (**g**), each dot in the bar plot represents one replicate, experiments were performed with 36 plants/replicate. Data are presented as mean ± SD (*n* = 3 biological replicates). Statistics In Fig. 2f, g were performed with unpaired two-tailed student t-test, *, **, *** and ****, *P* < 0.05, 0.01, 0.001, and 0.0001, respectively. Source data are provided in the Source Data File.

(ChIP-seq) for the two proteins in Col-0 and *atxr5 atxr6*. Sample distance clustering analysis of ChIP-seq datasets (Supplementary Fig. 8g) and Venn diagram (Supplementary Fig. 9a, b) showed high reproducibility among replicates, indicating the reliability of our ChIP-seq. When we counted total mapping reads that had multiple mapping locations in genome, we observed numerous peaks in heterochromatin. However, these peaks were not observed when only uniquely mapping reads were aligned (Supplementary Fig. 9c, d). We observed that a majority of RPA1A-bound loci coincided with RAD51-enriched regions, despite the fact that much fewer peaks were identified for the RPA1A ChIP-seq (Fig. 4a, and Supplementary Fig. 9c, d). These results suggested that the two proteins might coordinate with each other during HRR. Indeed, their physical interaction could be readily validated in our Co-IP experiments (Fig. 4b). These results were also consistent with earlier reports showing that concomitant absence of RPA1A and RPA1C mimics *rad51* phenotypes (i.e., sterile)[36].

We found that RAD51 and RPA1A were widely distributed over euchromatic regions that contain numerous PCGs (protein-coding genes) and intergenic regions in Col-0 and *atxr5 atxr6* (Fig. 4c and Supplementary Fig. 9c–e). Interestingly, this pattern is reminiscent of ChIP-seq patterns of RAD51 and RPA in *Mus musculus*[37] (Supplementary Fig. 9f), suggestive of their important functions in eukaryotes. Both RAD51 and RPA1A could also bind to heterochromatic regions including rDNA, TEs, and loci corresponding to non-coding RNAs (ncRNAs) (Fig. 4c and Supplementary Fig. 9c–e). We also compared peak numbers in heterochromatic elements in *atxr5 atxr6* and Col-0. The overall numbers of RAD51 and RPA1A-bound peaks over rDNA, TEs, and non-coding RNAs (ncRNAs) among the others seemed not to be affected by the loss of *ATXR5/6* relative to Col-0 (Supplementary Fig. 9e). One possible reason is that peaks numbers over heterochromatic regions represent a relatively small fraction of the total called peaks. We further performed density profiling of RAD51- and RPA1A-occupied regions by calculating the fractions of total reads for peaks from the different categories in total reads corresponding to all RAD51- and RPA1A-enriched loci. Interestingly, we observed a substantial increase in RAD51 and RPA1A occupancy at the loci corresponding to unknown genes, rDNA (Fig. 4c–f, and Supplementary Fig. 10a), and ncRNAs (Supplementary Fig. 10b, c) in *atxr5 atxr6* vs. Col-0. In other words, RAD51 and RPA1A displayed a robust increase in read coverage over rDNA among other classes in *atxr5 atxr6* compared to Col-0 (Fig. 4c–f and Supplementary Fig. 10a–c). Of note, the enrichment of RAD51 or RPA1A on TEs was not increased in *atxr5 atxr6* vs Col-0, likely because the plant materials were five- or six-week-old, and the TE amplification does not show an obvious difference in the mutant at this stage (Supplementary Fig. 10d, e).

Emerging evidence shows the association between *RPA, RAD51, BRCA1,* and *RAD51*-associated protein 1 (*RAD51AP1*) with transcription processes[38,39]. It has been also shown that reduced H3K27me1 on the *45 S rDNA* loci induces the expression of *45 S rRNA* variants, which is accompanied by higher copy number of *45 S rDNA* in 8 C nuclei of *atxr5 atxr6*[40]. In our hands, we found that RAD51 and RPA1A were significantly enriched on two sites of rDNA and the enrichment was well correlated with increased copy number of rDNA (Fig. 4d–f,

Supplementary Fig. 10d, e) in *atxr5 atxr6* compared to Col-0. Furthermore, the increased rDNA and the DNA that transcribes ncRNAs in *atxr5 atx6* vs Col-0 were concomitant with reduced level of H3K27me1 (Fig. 4d–f, Supplementary Fig. 10a–e). These results suggested that H3K27me1 might act as a repressive marker to regulate the replication of rDNA, and recruitment of HRR factors onto rDNA. By contrast, reduced H3K27me1 over heterochromatic regions observed in *atxr5 atxr6* was restored in *mbd9 atxr5 atxr6* and *sac3b atxr5 atxr6* mutants[15]. In parallel, the copy number of *45 S rDNA* decreased in *mbd9 atxr5 atxr6* and *sac3b atxr5 atxr6* vs. *atxr5 atxr6*(Supplementary Fig. 11a). Collectively, these results strongly support a model where reduced H3K27me1 leads to the heterochromatin amplification in *atxr5 atxr6*, especially rDNA loci, which in turn recruits RAD51 and RPA1A among other HRR factors to repair the increased DNA damage caused by heterochromatin amplification.

## Coordination between reduced HRR occupancy and impaired transcription in *atxr5 atxr6*

Besides rDNA loci, RAD51 and RPA1A occupied numerous PCG loci (Fig. 4c–e, and g). GO analysis showed that a large number of RAD51- and RPA1A-enriched genes belonged to several genetic pathways, such as response to cold, salt stress, oxidative stress, and jasmonic acid (JA)-mediated signaling (Supplementary Fig. 11b, c). In addition, RAD51 also bound genes related to defense response, response to SA, cell communication, immune system, fatty acid, and other important biological processes (Supplementary Fig. 11b). Interestingly, the PCGs enriched with RAD51 or RPA1A showed very low levels of H3K27me1 in Col-0 and *atxr5 atxr6*, supporting an active chromatin status at these loci (Fig. 4d, e, g). Further analysis showed that the signals of RAD51 and RPA1A tended to be evenly distributed over gene bodies rather than enriched at promoters in Col-0 and *atxr5 atxr6* (Fig. 4d, e, and g). In contrast to *rDNA* loci, RAD51 signal tended to be reduced over the bodies of PCGs in *atxr5 atxr6* vs. Col-0 (Fig. 4d, g).

We then selected 366 PCGs that showed lower RAD51 ChIP signal in *atxr5 atxr6* vs Col-0 (analyzed by DESeq2, log2 FC < −0.5, *P* value < 0.05) and assessed their transcription levels. Among the selected genes, the transcripts of 207 PCGs were detectable. Interestingly, 71.0% of the PCGs not only showed decreased RAD51 ChIP signal but also showed reduced transcript accumulation in *atxr5 atxr6* relative to Col-0 (Fig. 4h). Thus, our data indicated that reduced occupancy of RAD51 over otherwise actively transcribed regions coincided with their decreased transcript accumulation in *atxr5 atxr6*. The correlation between DNA repair and transcription observed here is reminiscent of a recent discovery of transcription-associated homologous recombination repair (TA-HRR)[39]. As one detrimental byproduct of transcription, unprocessed R-loops often cause the formation of DSBs, which in turn inhibit local ongoing transcription[41]. Supporting the importance of HRR factors in promoting transcription, RAD51-associated protein 1 (RAD51AP1) induces the formation of R-loop and favors RAD51-mediated D-loop formation to restore active transcription over these regions[38]. The fact that normally active chromatin regions in *atxr5 atxr6* showed lower RAD51 signals co-occurred with suppressed transcription suggests that the availability of HRR factors is limiting in *atxr5*

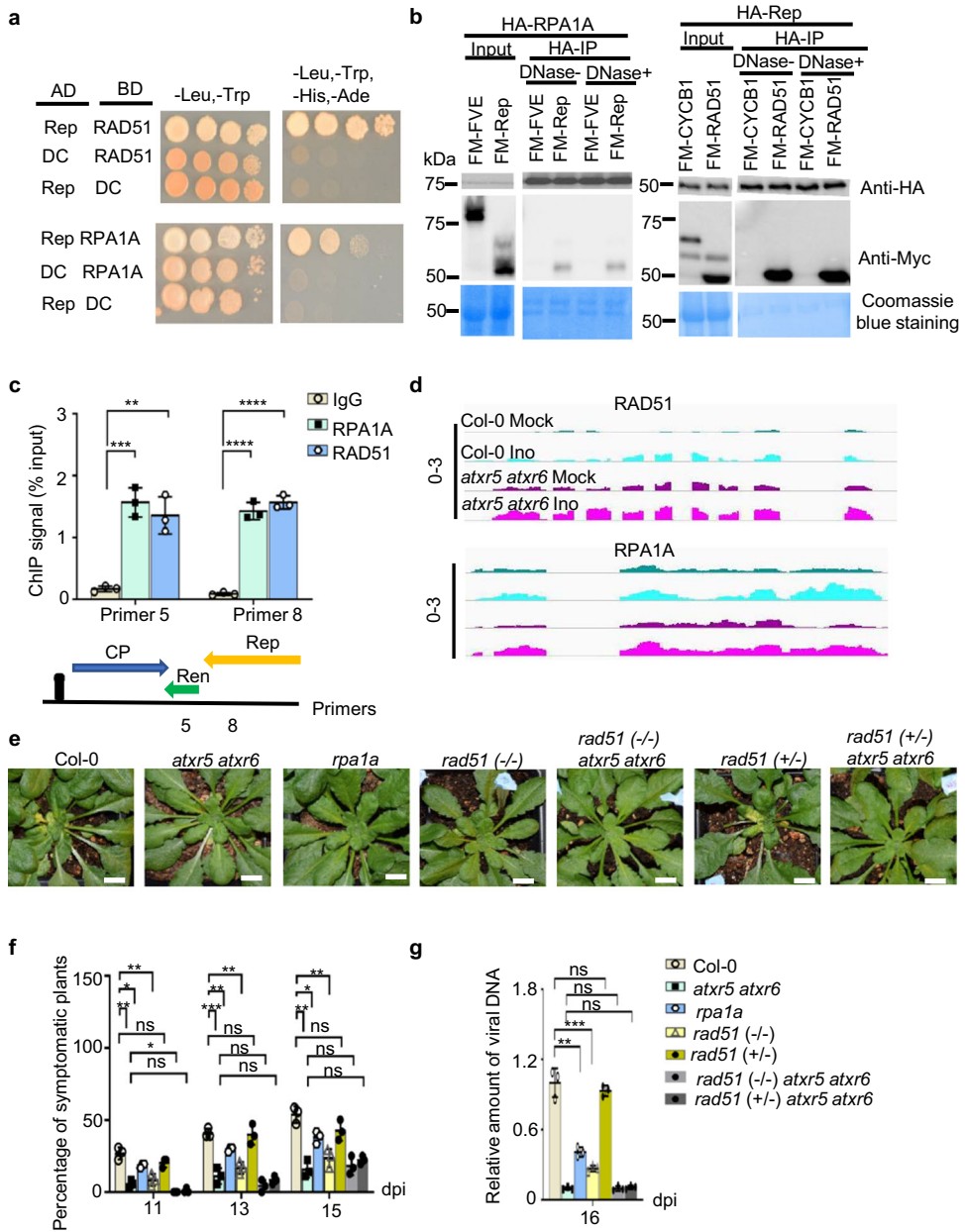

**Fig. 3 | Rep hijacks RAD51 and RPA1A to facilitate viral DNA amplification. a** Y2H screening pinpointed RAD51 and RPA1A as targets of Rep protein. The negative controls are AD/BD empty vectors. At least 16 independent colonies for each combination were tested and showed similar results. **b** Co-IP assay validated interactions of Rep with RAD51 and RPA1A in planta. FM-FVE[77], FM-CYCB1, and Coomassie blue staining of blots serve as negative and loading controls, respectively. The experiments were repeated twice with similar results. **c** ChIP-qPCR assay shows the binding of RAD51 and RPA1A on viral genome. IgG is a negative control. Each dot in the bar plot represents one replicate, experiments were performed with 36 plants/replicate. Data are presented as mean ± SD (n = 3 biological replicates). **d** IGV file shows transcript levels of *RAD51* and *RPA1A* in mock-treated or virus-inoculated Col-0 and *atxr5 atxr6* (normalized by RPKM). Scales for the distinct loci are shown on left as solid lines. **e** Representative phenotypes of CaLCuV-inoculated

Col-0, *atxr5 atxr6*, *rpa1a*, *rad51 (-/-)*, *rad51 (+/-)*, *rad51 (-/-) atxr5 atxr6*, and *rad51 (+/-) atxr5 atxr6* plants. Photographs were taken at 16 dpi. Scale bars, 1 cm. **f** Percentages of symptomatic plants induced by CaLCuV-inoculation in indicated backgrounds at 11, 13, and 15 dpi. **g** q-PCR assays show the amount of viral DNA A in CaLCuV-inoculated plants indicated at 16 dpi. In (**f**) and (**g**), each dot in the bar plot represents one replicate, most experiments were performed with 36 plants/replicate except for *rad51*(-/-) and *rad51*(-/-) *atxr5 atxr6* where 24 and 15 plants were used for each replicate, respectively. Data are presented as mean ± SD (n = 3 biological replicates). Normalization of viral DNA was conducted as in Fig. 2g. Statistics in Figs. 3c, f, and g were performed with unpaired two-tailed student t-test, *, **, *** and ****, *P* < 0.05, 0.01, 0.001, and 0.0001, respectively. Source data are provided in the Source Data File.

*atxr6* mutants, despite increased expression of these factors in this mutant background.

**Reduced binding of RAD51 and RPA1A to viral DNA in *atxr5 atxr6***
The ChIP-seq results suggested that HRR factors are recruited to the unstable genomic elements in *atxr5 atxr6*. On the other hand, RAD51 and RPA1A among other HRR factors that are de novo viral Rep

partners are essential to promote viral DNA amplification. These facts raised the possibility that Geminivirus might compete with host RACs in *atxr5 atxr6* for a limiting amount of RAD51, RPA1A and other HRR factors. To test this, we performed RAD51 and RPA1A ChIP-qPCR for the mock-treated and virus-inoculated samples at an early stage. We found that the loading of RAD51 and RPA1A on viral genome, reflected by the ratios of ChIP signal to the viral titer, was significantly higher in

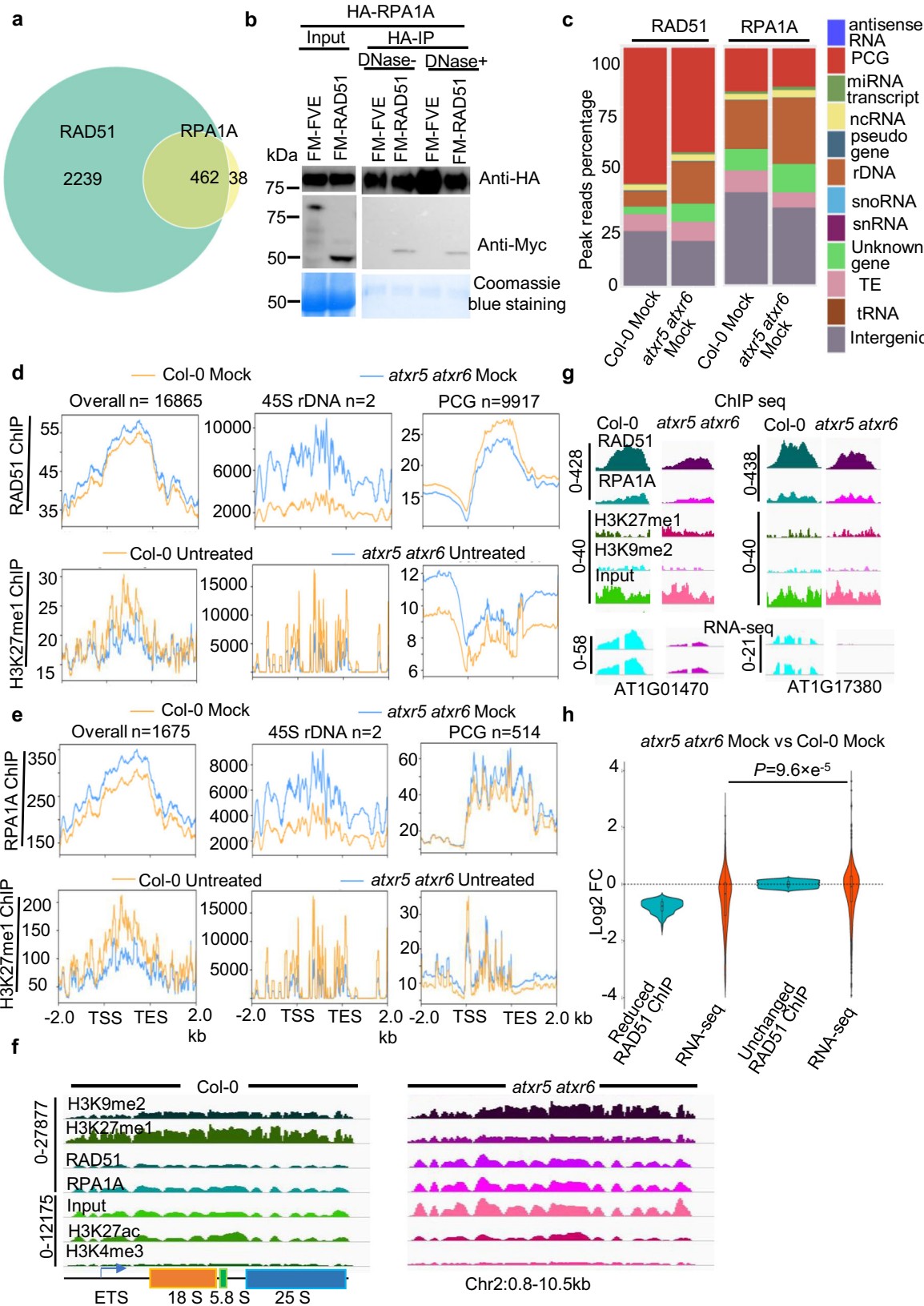

Col-0 than that in *atxr5 atxr6* (1.69% vs 0.48%) (Fig. 5a). By contrast, we observed a significant enrichment of RAD51 over host rDNA in *atxr5 atxr6* than in Col-0, regardless of mock-treated or virus-inoculated samples (Fig. 5a).

We next aimed to study how RAD51 and RPA1A were distributed to the host genome. Since *atxr5 atxr6* had significantly reduced

symptomatic plant rate and resultant lower viral titers (12% amount of Col-0), we purposely increased the number of symptomatic *atxr5 atxr6* plants to artificially mimic the symptomatic plant ratio of Col-0 for convenience of ChIP-qPCR assays and ChIP-seq. In this scenario, the virus titers in *atxr5 atxr6* could reach to 80% level of Col-0. Interestingly, ChIP-qPCR assays showed that the loading of RAD51 and RPA1A

**Fig. 4 | ChIP-seq assays of RAD51 and RPA1A show that unstable genomic DNA reshapes the distribution of HRR events in *atxr5 atxr6* vs. Col-0. a** Venn graph shows the overlap between RAD51 and RPA1A enriched regions in host genomes. **b** Co-IP assay validated the interaction between RPA1A and RAD51 in planta. FM-FVE[77] and Coomassie blue staining of blots serve as negative and loading controls, respectively. The experiments were repeated twice with similar results. **c** Peak reads distributions of RAD51 and RPA1A ChIP-seq in various locus categories in Col-0 and *atxr5 atxr6*. The y axis represents the percentage of reads mapped to loci of various categories. ncRNA and TEs represent non-coding RNA and transposable elements, respectively. **d**, **e** Distribution of normalized ChIP-signal of RAD51 and RPA1A (normalized with reads of the internal control mitochondrial DNA) and H3K27me1 (RPKM) from Col-0 and *atxr5 atxr6* over different categories. H3K27me1 ChIP-seq data were mined from published data (GSE111814). **f** IGV files of normalized ChIP

signals (RPKM) of H3K9me2, H3K27me1, RAD51, RPA1A, H3K27ac, and H3K4me3 on a *45 S rDNA* locus on chr2. H3K9me2 and H3K27me1, H3K4me3, and H3K27Ac ChIP-seq data were mined from published data GSE111814, GSE166897, and GSE146126, respectively. Scales for the distinct loci are shown on the left as solid lines. **g** IGV files of normalized ChIP signals (RPKM) of H3K27me1, H3K9me2, RAD51, and RPA1A over selected loci. The bottom panel displayed the IGV files of normalized transcript levels (RPKM) on selected loci from RNA-seq. Scales for the distinct loci are shown on the left as solid lines. **h** Violin plot shows transcript expression changes from the loci with reduced and unchanged RAD51 ChIP signal in mock-treated *atxr5 atxr6* vs Col-0. Horizontal lines in the bar plots display the 75th, 50th and 25th percentiles, respectively. Whiskers represent the minimum and maximum values. *P* value is calculated by unpaired two-tailed Welch's approximate *t*-test. Source data are provided in the Source Data File.

was still significantly decreased over viral genome whereas increased signal of RAD51 was detected over host rDNA and PCG loci in *atxr5 atxr6* than that in Col-0 (Fig. 5b). Be noted that the increased loading efficiency of RAD51 and RPA1A onto viral genome vs the natural condition was the trade-off of purposely increasing the number of virus-infected *atxr5 atxr6* vs Col-0. Thus, we concluded that RAD51 and RPA1A were indeed poorly recruited to viral genome in the mutant vs Col-0.

Using these samples where virus titers in *atxr5 atxr6* were purposely increased to 80% level of Col-0, we next performed ChIP-seq of RAD51 and RPA1A to profile their distributions cross host genome in Col-0 and *atxr5 atxr6* upon the virus inoculation. We normalized the ChIP-seq reads to the internal control of mitochondrial DNA where RAD51 and RPA1A do not bind so that we could compare the patterning changes of RAD51 and RPA1A associations with host genomes across different samples and treatments.

For Col-0, virus inoculation resulted in clearly reduced signal of RAD51 and RPA1A over host genome in virus-inoculated plants vs mock (Fig. 5c, d) despite that the expression of *RAD51* and *RPA1A* was elevated upon virus inoculation (Fig. 3d). The decreased signals of RAD51 mainly originated from the PCG loci and only marginally from rDNA regions (Fig. 5c, d and Supplementary Fig. 11d). This difference was likely due to the lack of DNA re-replication and relative lower DNA damage in Col-0 vs. *atxr5 atxr6*. The robust loss of RAD51 over PCG loci is also in line with the fact that RAD51 is predominantly associated with PCG loci in a normal condition. Of note, the genes with reduced RAD51 occupancy upon viral inoculation were related to well-known defense pathways involving JA, fatty acid, and s-adenosylmethionine (SAM) (Supplementary Fig. 12a). Moreover, the steady-state transcript levels from those loci were also reduced in inoculated Col-0 plants when compared to the mock (Supplementary Fig. 12b). These results suggested that Geminivirus might suppress transcription of the immune-related genes to attenuate the host defense system, leading to release of RAD51 from the loci and being re-routed onto viral DNA (Supplementary Fig. 12b). Altogether, the accumulated RAD51 in host cells and the RAD51 detained from the genome were all recruited to the viral genome to promote the viral DNA amplification in Col-0 upon virus inoculation (Fig. 3d, Fig. 5c, d and Supplementary Fig. 12a, b).

In contrast to Col-0, RAD51 ChIP signal displayed a robust increase in host genome and the signal covered 16865 loci in *atxr5 atxr6* upon the virus inoculation (Fig. 5c, d). This result suggested that the accumulated RAD51 was largely recruited onto the host genome in *atxr5 atxr6* upon virus inoculation rather than viral genome as observed in Col-0. RPA1A ChIP signal was also significantly distributed onto host genome in *atxr5 atxr6* vs Col-0. Despite this, the signal of RPA1A somehow showed a marginally decrease in the genome of *atxr5 atxr6* upon the virus inoculation vs mock treatment. This pattern was slightly different from that of RAD51, likely because RPA1A-ChIP-seq only recovered one-tenth of RAD51-bound loci (Fig. 5d). Alternatively, RPA1A has four more orthologs that might surrogate its function.

Detailed plotting of ChIP signals showed the increased RAD51 signal covered over rDNA, ncRNAs and PCGs in *atxr5 atxr6* upon virus inoculation relative to the mock (Fig. 5d and Supplementary Fig. 12c–e). The rDNA loci harbor relative high copy number of DNA sequences, and are hot spots of DNA damage due to tandem repeat sequences[42] (Supplementary Fig. 10d, e). Thus, over-replicated rDNA and increased DNA damage in *atxr5 atxr6* retain RAD51 in the loci[12]. Notably, we found virus inoculation significantly enhanced the RAD51 signal over 270 PCGs (analyzed by DESeq2, Log2 FC > 0.5, *P < 0.05*) in *atxr5 atxr6* whereas only 18.1% of them showed increased RAD51 signal in Col-0 (Supplementary Fig. 13a, Fig. 5a, b). Among 270 PCGs, 177 displayed detectable transcripts. Interestingly, 78% of 177 PCGs also displayed accumulated transcripts in the inoculated *atxr5 atxr6* plants (Fig. 5e). These genes are mainly related to defense and immune responses (23 genes), SA (53 genes), and JA (8 genes). Likely, RAD51 contributes to the upregulation of the defense-related loci through a TA-HRR mechanism (Fig. 5e and Supplementary Fig. 13b). Importantly, ChIP signals of RAD51 and RPA1A were significantly enriched over both heterochromatin (exemplified by rDNA) and PCGs in the infected *atxr5 atxr6* mutants compared to those of infected Col-0 plants (Fig. 5a–d).

Oppositely, the accumulated RAD51 and RPA1A could not be efficiently loaded to the viral mini-chromosomes for amplification in *atxr5 atxr6* (Fig. 5a, b). These results suggested that the occupancy of RAD51 and RPA1A in the heterochromatic regions and PCGs both contributed to the viral resistance of *atxr5 atxr6* vs Col-0 (Fig. 5a–d). To further test our model, we generated Col-0:*35S-FM-RAD51* and *atxr5 atxr6:35S-MYC-RAD51* and inoculated the stable transgenic lines with the virus (Supplementary Fig. 13c). Indeed, overexpression of RAD51 in both Col-0 and *atxr5 atxr6* clearly increased the ratio of symptomatic plants and viral DNA amount compared to their corresponding controls (Fig. 5f–h). Altogether, we concluded that re-replicated DNA and the loci of defense-related PCGs in *atxr5 atxr6* take up HRR components, leading to a shortage of the components essential for viral amplification in *atxr5 atxr6* vs Col-0.

### Increased DNA damage interferes with the interaction between Rep and RAD51

To further validate whether DNA damage attracted more HRR factors onto the host genome and prevented their loading to the virus genome in *atxr5 atxr6* vs Col-0, we co-transfected nYFP-HOP2 and cYFP-RAD51 into protoplasts of Col-0 and *atxr5 atxr6*, then evaluated YFP signal intensity as a proxy of protein-protein interaction level. The complementation of nYFP-HOP2 and cYFP-RAD51 showed YFP signal in the nucleus, reminiscent of the previously reported interaction between HOP2 and RAD51 in mammals[34]. Indeed, a higher YFP signal was detected in the protoplasts of *atxr5 atxr6* than those in Col-0, suggesting that more HRR factors are entailed to repair the DNA damage in *atxr5 atxr6* vs Col-0 (Fig. 6a, b and Supplementary Fig. 13d). In parallel, we assessed the interaction between RAD51 and Rep in this

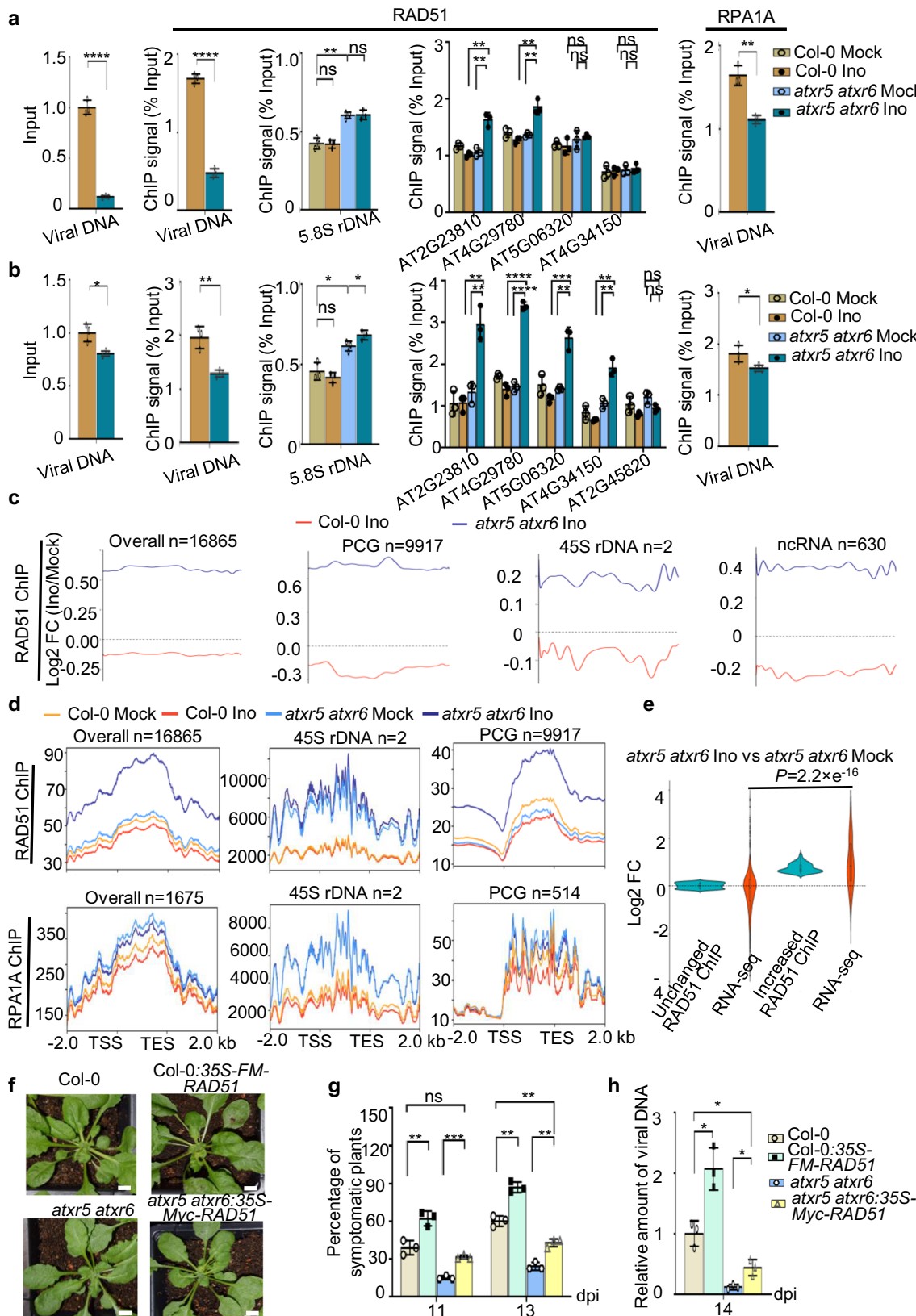

scenario. Indeed, BiFC assays showed significantly reduced YFP signals in *atxr5 atxr6* vs Col-0, indicative of a weaker interaction of RAD51 and Rep in the mutant (Fig. 6c, d and Supplementary Fig. 13e). These results, together with our ChIP-seq, indicated that HRR factors would be efficiently sequestered to the genome of *atxr5 atxr6*, preventing them from being recruited to viral genome for amplification.

We next assessed whether the scenario that DNA damage could retain HRR factors in host DNA, preventing the viral replication could go beyond *atxr5 atxr6* mutants. It has recently been reported that transgenic plants expressing H3.1S28A, but not H3.1S28 A31T, show heterochromatin amplification and activation of DNA repair pathway[16]. In line with this, we could readily detect significantly increased rDNA in

**Fig. 5 | Recruitment of HRR factors onto unstable genomic DNA and defense related genes prevents the loading of HRR factors on viral genome. a**, **b** ChIP-qPCR assays show that RAD51 enrichment was significantly increased over rDNA and selected PCGs but decreased over viral DNA in *atxr5 atxr6* vs Col-0 in a natural condition (**a**) or in a condition where increased number of symptomatic *atxr5 atxr6* plants were purposely collected (**b**). *AT2G45820* serves as a negative control. **c** Overview of RAD51 signal change in *atxr5 atxr6* and Col-0 upon virus inoculation. **d** Distributions of RAD51 and RPA1A ChIP signal (normalized with the reads of mitochondrial DNA) in different locus categories in mock-treated and virus-inoculated plants. **e** Violin plot shows transcription changes from the loci with unchanged and increased RAD51 ChIP signal in virus-inoculated vs mock-treated *atxr5 atxr6*. Horizontal lines in the bar plots display the 75th, 50th and 25th percentiles, respectively. Whiskers represents the minimum and maximum values.

*P* value is calculated by unpaired two-tailed Welch's approximate *t*-test. **f** Representative pictures of the virus-inoculated plants in indicated lines. Photographs were taken at 14 dpi. Scale bars, 1 cm. **g** Percentages of symptomatic plants of virus-inoculated plants in indicated backgrounds at 11 and 13 dpi. **h** Constitutive expression of RAD51 promotes the viral DNA amplification in Col-0 and *atxr5 atxr6*. qPCR assays show increase of virus titers in RAD51 overexpression lines vs their reference backgrounds. Samples were collected at 14 dpi. Normalization of viral DNA was conducted as Fig. 2g. In (**a**), (**b**), (**g**) and (**h**), each dot in the bar plot represents one replicate, the experiments were performed with 36 plants/replicate. Data are presented as mean ± SD (n = 3 biological replicates). Statistics in Figs. 5a, b, g, and h were performed with unpaired two-tailed student t-test, *, **, *** and ****, *P* < 0.05, 0.01, 0.001, and 0.0001, respectively. Source data are provided in the Source Data File.

transgenic plants expressing H3.1S28A compared to Col-0 and transgenic plants expressing H3.1S28A A31T (Fig. 6e). Importantly, the H3.1S28A transgenic plants showed robust virus resistance against CaLCuV, whereas the H3.1S28A A31T lines did not (Fig. 6f–h). Thus, in the H3.1S28A line, unstable genomic elements could also retain HRR factors and prevent them from being recruited onto the viral genome, leading to suppression of viral DNA amplification.

Since SA triggers DNA damage and also induces the expression of defense genes, we consider this treatment as a natural condition, reminiscent of the physiological condition in *atxr5 atxr6*. In our hands, the exogenous SA treatment activated the DNA damage response and suppressed the viral DNA amplification (Fig. 6i, j). Moreover, the application of DNA damage reagent bleomycin (BLM) enhanced the immunity of wild-type plants against virus inoculation (Supplementary Fig. 14a, b). These results consolidated our model that DNA damage induced by unstable DNA regions can request HRR factors to the host genome to prevent them from being used by the viral genome for efficient viral DNA amplification.

### BRCA1, HOP2 and CYCB1 are bona fide partners of RAD51

We next aimed to identify the factors that potentially recruited RAD51 or RPA1A to the host genome. Through Y2H assays, we recovered BRCA1, and CYCB1;1 as partners of RAD51 from a few candidate genes (Fig. 7a, and Supplementary Fig. 14c). Neither protein appeared to directly interact with RPA1A in the assays (Fig. 7a, b). RAD51 could be readily detected in the co-immunoprecipitated products of CYCB1;1 and BRCA1 but not in control IPs (Fig. 7b). In addition, homologous pairing protein 2 (HOP2) interacted with RAD51 in Co-IP, but not in Y2H assay (Fig. 7b, Supplementary Fig. 14c). Of note, all three proteins are intrinsically disordered or contain disordered segments; and these features might contribute to their interaction with RAD51 (Supplementary Fig. 14d).

### Roles of BRCA1, HOP2 and CYCB1 in loading RAD51

Interactions of BRCA1, HOP2, and CYCB1;1 with RAD51 detected here and also in mammalian cells and plants[34,43,44] suggested the potential roles of these proteins in the recruitment of RAD51 onto unstable DNA. A robust increase of viral DNA was detected in *brca1 atxr5 atxr6* compared to that in *atxr5 atxr6*, and no obvious difference in the viral titer was detected between *brca1* and Col-0 (Fig. 1e–g). These contrasting results suggested that RAD51-centered DNA repair components were not able to reach the unstable loci in *atxr5 atxr6*; rather, the components were re-routed to viral mini-chromosomes for DNA amplification in *brca1 atxr5 atxr6* compared to *atxr5 atxr6* (Fig. 7c, d). Differently, DNA repair components were largely recruited to viral genome in the single mutant *brca1* and Col-0 where host DNA is intact in a normal physiological condition. Thus, BRCA1 might regulate viral DNA amplification in a manner dependent on RAD51 and other HRR factors in the *atxr5 atxr6* background.

To test whether deletions of *HOP2* and *CYCB1;1* would also interrupt recruitment of HRR factors between host DNA and viral genome

in *atxr5 atxr6*, we generated *cycb1 atxr5 atxr6* and *hop2 atxr5 atxr6* mutants through genetic crossing and tested their susceptibility to viral inoculation. When challenging the higher order mutants with CaLCuV, *hop2 atxr5 atxr6* and *cycb1 atxr5 atxr6* mutants, like *brca1 atxr5 atxr6*, had higher percentages of symptomatic plants and more severe chlorosis phenotype than those in *atxr5 atxr6* (Fig. 7e–g). Moreover, *hop2 atxr5 atxr6* and *cycb1 atxr5 atxr6* mutants showed higher viral DNA content relative to *atxr5 atxr6* (Fig. 7e–g). These results indicated that BRCA1, HOP2, and CYCB1;1 were essential for the virus resistance phenotype in *atxr5 atxr6*, with the suggestion that the three proteins might be responsible for the recruitment of RAD51 to the unstable loci and PCGs in the host genome. Of note, differential susceptibilities among the triple mutants (Fig. 7e–g) suggested that their contributions to the loading of RAD51 or other HRR factors onto damaged DNA might be different. Alternatively, additional paths for recruitment to RAD51 and other HRR factors might compensate for the loss of the three components in different degrees.

To further test our model, we selected several representative loci on heterochromatic regions and PCG loci to perform ChIP-qPCR. Indeed, RAD51 signals over heterochromatic regions including *5.8 S rDNA* were all lower in the triple mutants than that in *atxr5 atxr6* under mock treatment and inoculation conditions. This result indicated the deficient recruitment of RAD51 onto host heterochromatic regions in the triple mutants vs *atxr5 atxr6* (Fig. 7c and Supplementary Fig. 14e). Moreover, deletions of *CYCB1;1*, *HOP2* and *BRCA1* in *atxr5 atxr6* also prevented the loading of RAD51 on PCGs upon virus inoculation (Fig. 7c). On the other hand, more efficient loading of RAD51 on viral genome was observed in *hop2 atxr5 atxr6*, *brca1 atxr5 atxr6* and *cycb1 atxr5 atxr6* compared with that of *atxr5 atxr6* (Fig. 7d). In contrast, there was barely heterochromatin amplification and HRR events in WT background. As a consequence, the amount of RAD51 was sufficient for viral amplification, deletions of *CYCB1;1*, *HOP2* and *BRCA1* did not affect the viral pathogenesis and accumulation (Supplementary Fig. 7e, f). Thus, we concluded that BRCA1, HOP2 and CYCB1;1 indirectly regulate viral DNA replication through controlling the availability of RAD51 to the viral genome in the *atxr5 atxr6* background (Fig. 7h).

## Discussion

Here we reported a novel regulatory role of histone MTases on DNA virus amplification. Different from other histone MTases that deposit repressive marks on viral genomes and typically repress viral infections, ATXR5 and ATXR6 coincidentally promote viral amplification as a trade-off to maintaining host genome integrity. This unique mode of action in virus-inoculation is highlighted by the fact that loss-of-function mutants of *atxr5 atxr6* become resistant to the virus compared to Col-0. The underlying mechanism for the viral resistance is that mutations of *ATXR5* and *ATXR6* cause unstable heterochromatic regions exemplified by rDNA and ncRNAs loci, which activate the HRR pathway. These unstable heterochromatic regions together with defense-related PCGs in euchromatin regions, in turn, sequester HRR

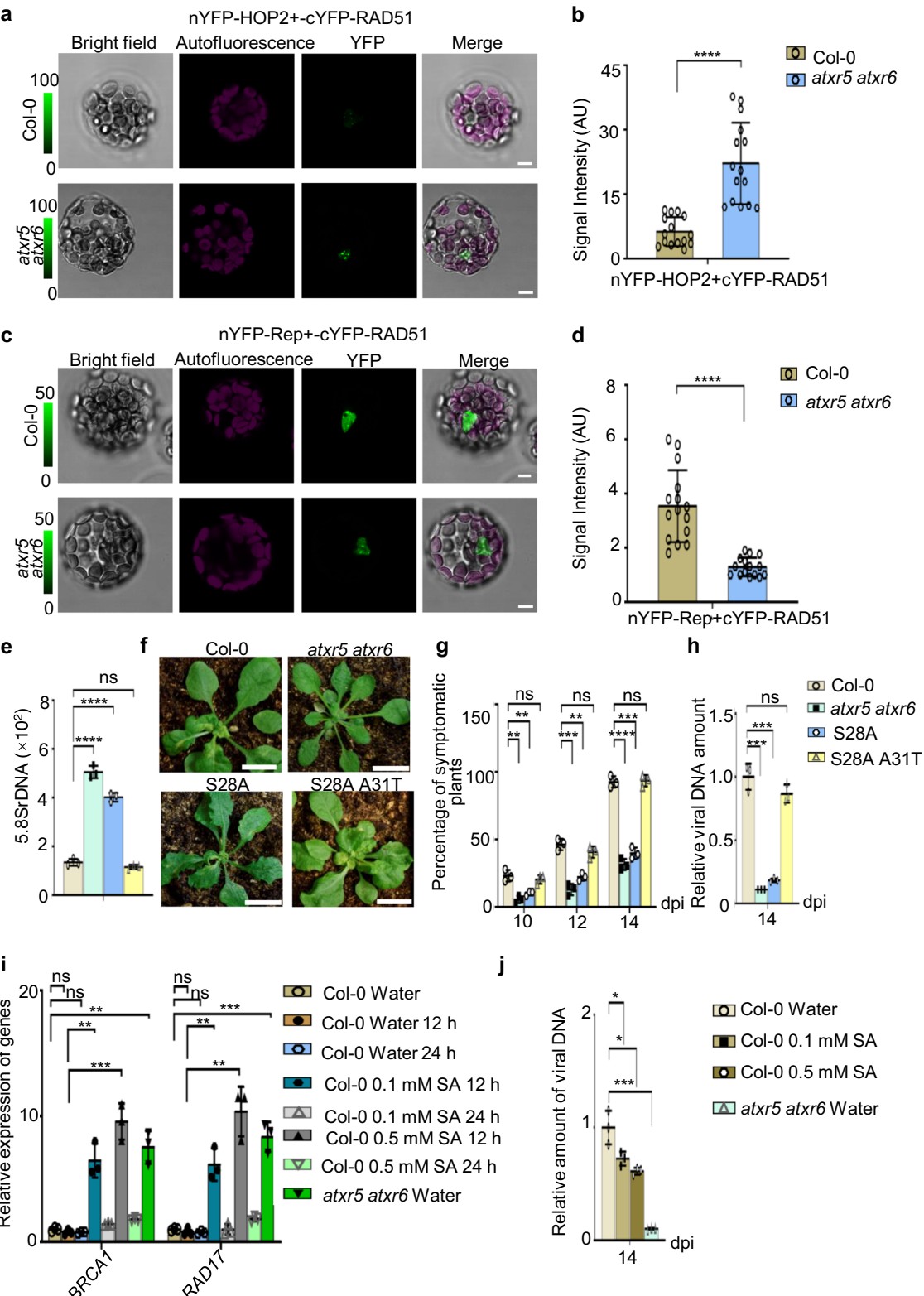

factors and preclude them from being hijacked by the virus-encoded Rep protein, leading to suppression of the viral replication (Fig. 7h). Several pieces of evidence support our model: 1) DNA repair factors such as ATM is required for efficient viral DNA amplification (Fig. 2e–g); 2) the viral protein Rep hijacks RAD51 and RPA1A among other DNA repair components to facilitate viral amplification (Fig. 3c–g); 3) although *MBD9, SAC3B,* and *BRCA1* play opposite roles in regulating heterochromatin amplification and TE reactivation, loss of

any of these genes interrupts activation of HRR and restores viral amplification in *atxr5 atxr6* to similar levels observed in Col-0. These observations suggest that heterochromatin amplification and TE reactivation in *atxr5 atxr6* do not function as a prophylactic system against CaLCuV infection (Fig. 1c–g); 4) reduced H3K27me1 causes extra copies of heterochromatic DNA, including rDNA and ncRNA loci, which induces the recruitment of HRR factors such as RAD51 and RPA1A at heterochromatic loci in the host (Fig. 4c–f); 5) deficient

**Fig. 6 | Increased DNA damage promotes RAD51 interaction with host HOP2 but not viral Rep. a** BiFC of YFP assays showed that mutations of ATXR5/6 promote the interaction between RAD51 and HOP2. See Supplementary Fig. 13d for negative controls. 15 independent protoplasts were examined for the interaction and showed similar results. Scale bars, 5 μm. **b** Statistical analysis of YFP signal intensity of each protoplast. **c** BiFC of YFP assays showed that interaction between RAD51 and Rep was compromised in *atxr5 atxr6*. See Supplementary Fig. 13e for negative control. 15 independent protoplasts were examined for the interaction and showed similar results. Scale bars, 5 μm. **d** Statistical analysis of YFP signal intensity of each protoplast. In (**b**) and (**d**), each dot in the bar plot represents one protoplast. Data are presented as mean ± SD (*n* = 15 biologically independent protoptlasts). **e** q-PCR assays show the amount of rDNA in different lines. The relative amount of rDNA was normalized against *UBQ10*. **f** Representative phenotypes of CaLCuV-inoculated Col-0, *atxr5 atxr6*, and third generation of (T3) lines expressing

H3.1 variants. Photographs were taken at 14 dpi. Scale bars, 1 cm. **g** Percentages of symptomatic plants induced by CaLCuV-inoculation in indicated backgrounds at 10, 12, and 14 dpi. **h** q-PCR assays show the amount of viral DNA A in CaLCuV-inoculated plants at 14 dpi. Normalization of viral DNA was conducted as Fig. 2g. **i** SA treatment activates the DNA damage response. qRT-PCR assays show the transcription level of HRR factors in indicated genotypes and treatments. **j** q-PCR assays show the relative amount of viral DNA A in the indicated genotypes and treatments at 14 dpi. In (**e**), (**g**), (**h**), (**i**), (**j**), each dot in the bar plot represents one replicate. Experiments were performed with 36 plants/replicate except in (**i**) 6 plants/replicate was used. Data are presented as mean ± SD (n = 3 biological replicates). Statistics In Figs. 6b, d, e, g, h, i and j were performed with unpaired two-tailed student t-test, *, **, *** and ****, *P* < 0.05, 0.01, 0.001, and 0.0001, respectively. Source data are provided in Source Data File.

loading of RAD51 and RPA1A on viral genome is detected in *atxr5 atxr6* relative to Col-0, despite higher accumulation of the proteins in the mutant and stronger ChIP signal over the *atxr5 atxr6* genome relative to Col-0 (Fig. 5a–d); and 6) BRCA1, HOP2, and CYCB1;1 promote the recruitment of RAD51 onto the host genome, whereas depletions of *HOP1*, *BRCA1*, and *CYCB1* in *atxr5 atxr6* cause inefficient loading of RAD51 onto the unstable host genome and PCGs related to defense, and re-routing of RAD51 onto viral genome in *atxr5 atxr6*, leading to hyper-susceptibility to virus inoculation in the triple mutants vs *atxr5 atxr6* (Fig. 7c–g). Thus, we conclude that RAD51 and RPA1A, among other HRR factors, are the cornerstones in the battle of host and virus. If the virus fails to recruit RAD51 and RPA1A, it will fail to parasitize the host, as seen in *atxr5 atxr6*.

DSBs activate HRR to faithfully repair DNA during replication to maintain genome stability[45]. As parts of heterochromatic elements, rDNA is considered a hot spot for recombination events due to their repetitive elements. During HRR, the RPA complex coats resected ends to prevent their degradation. RAD51 then replaces RPA and mediates the D-loop formation and strand invasion[45]. Robust loading of RAD51 and RPA1A onto rDNA and ncRNAs loci accompanies lower H3.1K27me1 levels in *atxr5 atxr6* (Fig. 4d, e). Similarly, reduced H3.1K27me1 correlates with higher rDNA and ncRNA copy number (Fig. 4d, e, Supplementary Fig. 10b–e). These results suggest that RAD51 and RPA1A participate in maintaining genome stability at heterochromatic regions in the absence of H3.1K27me1. In our study, BRCA1 and HOP2 promote the recruitment of RAD51 over rDNA regions to facilitate DNA repair in *atxr5 atxr6*. A similar mechanism has been reported in mammalian cells, where BRCA1 and HOP2 complexes stabilize the RAD51-ssDNA filament and promote RAD51-mediated homologous DNA pairing process in vitro[34,44]. In *Arabidopsis*, CYCB1;1-CDKB1 complex phosphorylates RAD51 in vitro[43]. Phosphorylation of RAD51 is required for its efficient DNA binding[46]. Remarkably, loss-of-function mutations of *BRCA1, HOP2, and CYCB1;1* in *atxr5 atxr6* compromised viral resistance of *atxr5 atxr6* to different extents (Fig. 7c–g), suggesting robust loading of RAD51 over heterochromatic region in *atxr5 atxr6* mutants is required for its viral resistance.

RAD51 is also distributed along the gene body of PCGs with low H3K27me1 under physiological conditions (Fig. 4d). Furthermore, decreased RAD51 ChIP signals over PCGs are concordant with reduced transcripts levels in virus-inoculated Col-0 (Supplementary Fig. 12b). Notably, the RAD51-enriched PCGs are dedicated to the defense response, response to hypoxia, SA, JA and cold and salt stresses. These observations suggest that RAD51 might coordinate TA-HRR to regulate the transcription, while coping with physiological stresses (Supplementary Fig. 11b). Exogenous SA treatment induces DNA breaks and promotes RAD51 binding to the promoters of pathogenesis-related genes (PR). Conversely, depletion of BRCA2A[28] and ATR[29], two partners of RAD51, suppresses the expression of SA-induced defense-related genes and efficient immune response. These results further imply that RAD51 is required for efficient transcription regulation and defense

response. Importantly, the fact that PCGs display reduced occupancy of RAD51 while producing lower expression of defense-related transcripts in infected Col-0 plants implies a dual role of Rep during infection (Supplementary Fig. 12a, b): hijacking the HRR factors to facilitate the viral genome amplification whereas attenuating the plant defense system in WT plants. By contrast, RAD51 occupancy over defense-related genes and transcription of the corresponding loci were enhanced in *atxr5 atxr6* upon virus inoculation (Fig. 5c–e, Supplementary Fig. 13b), implying that the elevated transcription of defense genes in *atxr5 atxr6* compete with the virus for a limited amount of RAD51, restricting virus amplification and enhancing the immune response, thus resulting in viral resistance. During the parasitic lifestyle, DNA viruses activate the host DNA repair mechanism and hijack the replication machinery to the viral genome. Here, *rad51* and *rpa1a* mutants displayed resistance to Geminivirus inoculation, indicating that the proteins are critical for viral amplification. Consistently, RAD51 interacts with the Rep encoded by mungbean yellow mosaic India virus (MYMIV) and promotes geminiviral DNA replication in the heterologous system *Saccharomyces cerevisiae*[47]. Geminiviruses replicate their genomes through rolling circle replication and recombination-dependent replication[2]. RAD51 might function as a recombinase to directly facilitate the rolling circle replication of the virus[48]. RPA1A might also contribute to the rolling circle replication of geminiviruses, reminiscent of RPA-mediated enhancement of the replication of simian virus 40[49,50]. As key components for homologous recombination, RAD51, and RPA1A might also directly promote the recombination-dependent replication of Geminiviruses. Supporting this model is that RAD51D, a paralog of RAD51, has been reported to promote the recombination-dependent replication of Geminivirus[51]. In our studies, although virus inoculation enhances the expression of RAD51 and RPA1A in *Arabidopsis*, the ChIP signals of RAD51 and RPA1A over the host genome were reduced in inoculated vs mock-treated Col-0 plants (Figs. 3d and 5c, d). Furthermore, Rep interacts with RAD51 and RPA1A in planta (Fig. 3b). All these observations clearly indicate that Rep hijacks RAD51 and RPA1A to facilitate viral amplification during the infection process.

In summary, we propose that robust retention of RAD51 and RPA1A onto unstable host DNA, along with increased RAD51 accumulation over the upregulated defense-related PCGs, prevent the efficient loading of the replication-essential factors onto the viral genome, leading to a resistance phenotype in *atxr5 atxr6*. This study implies that the virus might adapt to the healthy host to hijack host DNA-repairing components for the viral replication. On the other hand, an unstable genome could retain DNA-repairing proteins onto host genome to gain plant resistance to viral infection. One implication of this study is that one might apply a certain level of genome toxicity stress on crops or insert certain pieces of RAD51-favored unstable DNA elements into crop genomes[52]. Such actions might be able to trap HRR factors onto host genomes to a certain degree, while granting viral resistance, ensuring agricultural production.

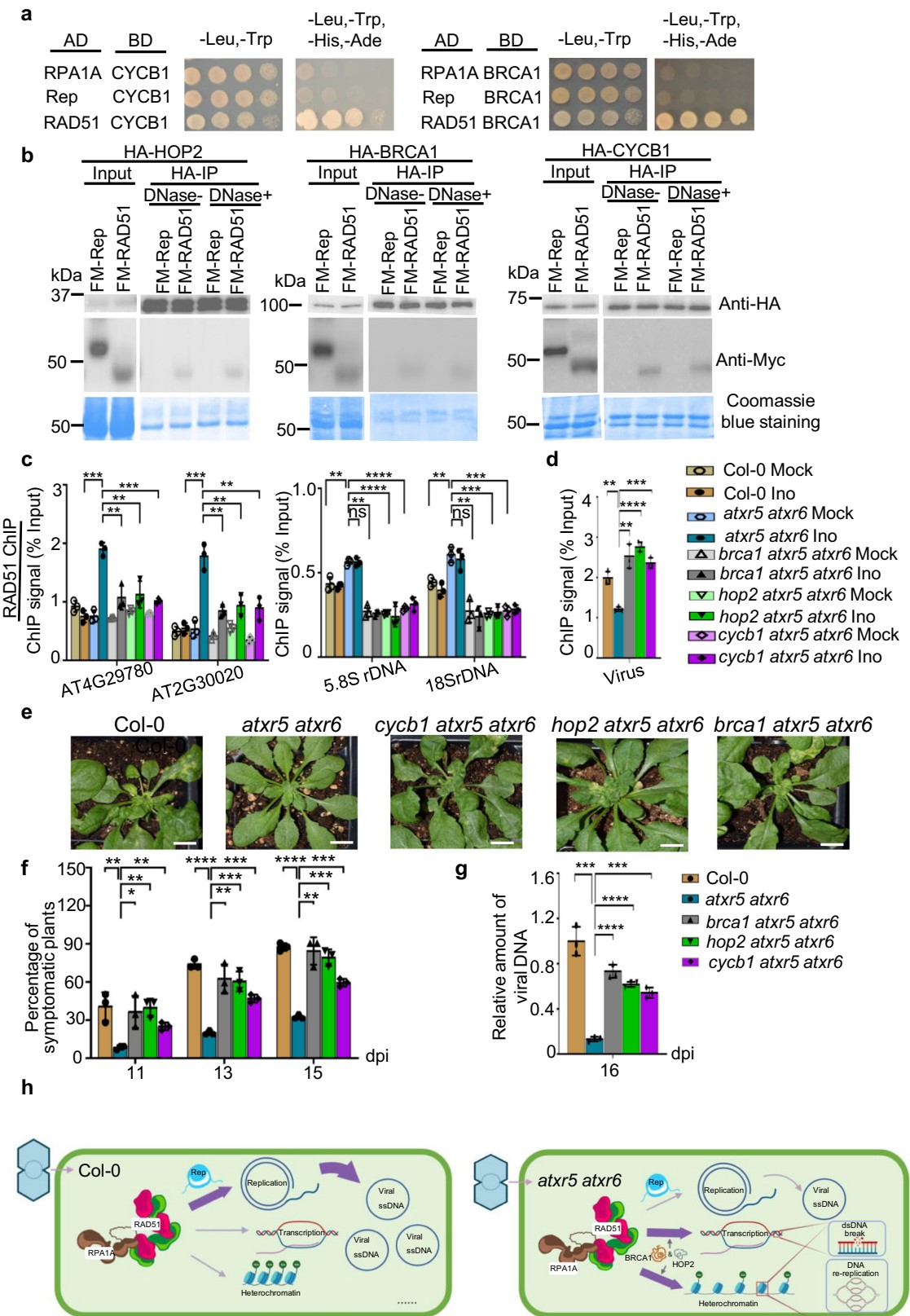

Similarly, disabling H4K20me1 transferases in human, the functional homologs of ATXR5/6, can also activate TONSOKU-LIKE and BRCA1 machinery[53,54]. Host retention of the DNA repairing machinery to correct genome instability would be expected to increase host immunity and impair the DNA virus amplification, thus providing an opportunity to defend against the viruses. In addition,

breaks on rDNA lead to the loss of rDNA repeats and exacerbate genomic instability during aging[55,56]. Overexpression of RAD51 restores the accumulated DSBs in aged cells and extends the replicative life span of yeast[56,57], implying the potential application of H4K20 and RAD51 in curing rDNA-associated human diseases such as amyotrophic lateral sclerosis, and Huntington's disease[58].

**Fig. 7 | BRCA1, HOP2 and CYCB1 regulate the antiviral defense in an RAD51-dependent manner. a** Y2H assays validated the interactions of RAD51 with CYCB1 and BRCA1. **b** Co-IP assay validated the interactions of RAD51 with CYCB1, BRCA1, and HOP2 in planta. FM-Rep and Coomassie blue staining of blots serve as negative and loading controls, respectively. The experiments were repeated twice with similar results. **c** ChIP-qPCR assay shows that BRCA1, HOP2 and CYCB1 are required for recruitment of RAD51 over rDNA and PCGs in *atxr5 atxr6*. **d** ChIP-qPCR assay shows that BRCA1, HOP2, and CYCB1 are required for reduced binding of RAD51 over the viral genome in *atxr5 atxr6*. **e** Representative pictures of the virus-inoculated plants in indicated backgrounds. Photographs were taken on at 15 dpi. Scale bars, 1 cm. **f** Percentages of symptomatic plants of virus-inoculated plants in indicated backgrounds. **g**, q-PCR assays show the amount of viral DNA A in CaLCuV-inoculated plants indicated at 16 dpi. Normalization of viral DNA was conducted as Fig. 2g. In (**c**), (**d**), (**f**), and (**g**), each dot in the bar plot represents one replicate,

experiments were performed with 32 plants/replicate. Data are presented as mean ± SD (*n* = 3 biological replicates). **h** A proposed model for Geminivirus competition with host genome for HRR to facilitate viral amplification. In the WT plants with a stable genome the virus inoculation suppresses the transcription of defense-related genes to evict RAD51 and RPA1A from host genome, and then viral-encoded Rep protein hijacks RAD51 and RPA1A for efficient viral replication. However, in *atxr5 atxr6*, unstable host DNA retains large amount of DNA repairing factors and promotes the transcription of defense-related genes. Consequently, unstable host heterochromatic elements coordinate with the upregulated defense-related genes to retain a large amount of DNA repairing machinery to prevent its rerouting to viral genome, leading to low-efficiency virus amplification. Figure 7h was created with BioRender.com. Statistics in Figs. 7c, d, f, and g were performed with unpaired two-tailed student *t*-test, *, **, *** and ****, *P* < 0.05, 0.01, 0.001, and 0.0001, respectively. Source data are provided in the Source Data File.

## Methods
### Plant materials and growth condition
*A. thaliana* plants were grown under cool-white, fluorescent light (120 µmol m$^{-2}$ s$^{-1}$) in a short-day condition (8 h light/16 h dark) with 22 °C and 50% humidity. Col-0 was used as a wild type and *atxr5 atxr6, atxr5, atxr6, mbd9, mbd9 atxr5 atxr6, sac3b, sac3b atxr5 atxr6, brca1, brca1 atxr5 atxr6, atm, atr, sog1, atm atxr5 atxr6, fas2, htr1, htr2, htr3, htr9, htr1 htr2 htr3 htr9, suwh4 suwh5 suwh6* were described previously[10,12,13,16,33]. Seeds of *drm1 drm2 cmt3* (CS16384), and *clf-28* (SALK_139371) were obtained from Arabidopsis biological center (ABRC). T-DNA insertion mutants of *CYCB1 (SALK_200647), HOP2 (SALK_136002), CHR31 (SALK_204501), TK1A (SALK_097767), RAD51 (SAIL_873_C08), RPA1A (SALK_017580)* were purchased from ABRC and genotyped by PCR. High order mutants *cycb1 atxr5 atxr6, hop2 atxr5 atxr6* and *rad51* (+/-) *atxr5 atxr6* were generated by genetic crossing. The triple mutant of *rad51* (-/-) *atxr5 atxr6* was generated from selfing *rad51* (+/-) *atxr5 atxr6* and genotyped by PCR before the virus infection assays due to the sterility of *rad51* (-/-). For *cycb1 atxr5 atxr6, hop2 atxr5 atxr6*, we used F4 generation of homozygotes to do viral infection. H3.1 transgenic lines expressing H3.1S28A or H3.1 S28A A31T were in T3 generation as previously described[16]. Transgenic materials were obtained by floral dipping with GV3101 containing pBA-Flag-4Myc-RAD51 in Col-0 or floral dipping with GV3101 containing pCambia1300-Myc-RAD51 in *atxr5 atxr6* as described[59].

### Plasmid constructs
Full-length coding DNA sequences (CDSs) of *RAD51* (AT5G20850), *CYCB1* (AT4G37490), *Rep* (encoded by CaLCuV) were cloned from CaLCuV-infected Col-0 and *HOP2* (AT1G13330) was cloned from U83877 (obtained from ABRC) into pENTER/D-TOPO (Thermo Fisher) vectors and confirmed by sequencing. The PCR was performed with *Thermococcus kodakaraenis* (KOD) DNA polymerase (Novagen).

Full-length CDSs of *BRCA1* (AT4G21070, stock G24692), *PARP1* (AT2G31320, G19185), *EMB1379* (AT5G21140, G63718), *TK1A* (AT3G07800, G10015), *TSO2* (AT3G27060, G11902), *XRI1* (AT5G48720, G63543), *HTA3* (AT1G54690, G13465), *HTA5* (AT1G08880, G13529), *PCNA1* (AT1G07370, G21468), RPA1A (AT2G06510, G61064), *PARG1* (AT2G31870, G16153), *WIP3* (AT4G10265, G60564) ligated with pENTER223 were obtained from ABRC and confirmed with enzyme digestion.

For Y2H constructs, full-length CDSs of *RAD51, CYCB1, Rep, HOP2, BRCA1, PARP1, EMB1379, TK1A, TSO2, XRI1, HTA3, HTA5, PCNA1, RPA1A, PARG1* and *WIP3* in PENTER-223 were cloned into pGADT7-DC and pGBKT7-DC by LR reaction. Plasmids confirmed by enzyme digestion were used for yeast transformation. pGADT7-ATXR5 and pGBKT7-ATXR5 were previously described[14]. For transient expression constructs, full lengths CDSs of *RAD51, BRCA1, RPA1A, HOP2,* and *CYCB1;1* were cloned into pBA-HA3-DC by LR reaction. Full length CDSs of *RAD51, CYCB1, and Rep* were cloned into pBA-Flag-4Myc-DC. Plasmids

confirmed by enzyme digestion were transferred to GV3101 for transient expression in *N. benthamiana*.

For Bimolecular fluorescence complementation (BiFC) constructs, full-length CDSs of *Rep* and *HOP2* were cloned into pBA-nYFP-DC by LR reaction. *RAD51* CDS was cloned into pBA-cYFP-DC. Plasmids were confirmed by enzyme digestion, extracted as previously described[14], and transfected to protoplasts.

For stable transgenic plants, pBA-Flag-4Myc-RAD51 was obtained as described above. To obtain pCambia1300-Myc-RAD51, pCambia1300-Myc-DC-Nluc was digested by PstI and dephosphorylated with Calf Intestinal Alkaline Phosphatase (CIP). The missed destination cassette DC fragments was obtained by PCR of another normal DC cassette followed by PstI-digesttion. The resultant fragment was ligated into PstI/CIP treated pCambia1300-Myc-DC-Nluc to generate pCambia1300-Myc-DC. Full length CDS of RAD51 was transferred into pCambia1300-Myc-DC by LR reaction.

### CaLCuV inoculation assays
Plants were grown on MS plate under 8-h light/16-h dark conditions for 10 days and then transferred to soil under 8-h light/16-h dark conditions for around 14 days to reach eight true-leaf developmental stage. We first extracted the plasmid from *E. coli* (DH5α) that contains the geminivirus genome and transformed it into agrobacteria. Fresh agrobacteria containing geminivirus genome together with silicon carbide powder was sprayed by a pump onto the center of *Arabidopsis* (which were newly emerged rosette leaves) at 80 psi (pound per square inch). Plants with different genotypes were infected by agroinfiltration of CaLCuV infective clones of pNSB1090 DNAA and pNSB1091 DNAB. The symptoms of plants were monitored and evaluated daily since we began to observe the yellow mosaic and chlorosis in viral infected plants. The SA treatment was conducted as previously described[60]. The concentration of bleomycin (BLM) treatment was selected as previously described[29] and carefully conducted in a hood with proper protection. To assess systemic infection, we harvested the eight newly emerged rosette leaves of CaLCuV-inoculated plants. We collected 36 CaLCuV-inoculated plants as one biological replicate to perform the following assays for the majority results. For *rad51* and *rad51 atxr5 atxr6*, we collected 24 or 15 homozygous plants as one biological replicate (*rad51* is sterile, genotype is confirmed by PCR after viral inoculation) respectively. We collected 36 CaLCuV-inoculated plants and Mock-treated Col-0 and *atxr5 atxr6* as one biological replicate to perform the assays in Fig. 6a. For the assays in Fig. 6b, we collected 36 Mock-treated Col-0 and *atxr5 atxr6* and 36 CaLCuV-inoculated Col-0 as one biological replicate. To purposely increase the symptomatic plants in CaLCuV-inoculated *atxr5 atxr6*, we first inoculated 360 *atxr5 atxr6* together with 108 Col-0 (36 Col-0 in one tray as one biological replicate), and selectively collected *atxr5 atxr6* with CaLCuV-induced phenotypes from 360 CaLCuV-inoculated *atxr5 atxr6* to make sure the number of symptomatic plants in the *atxr5 atxr6* (selected 36 CaLCuV-inoculated plants as one biological

replicate) was the same to that in Col-0. To assess the plasmid transfection efficiency, we collected the 12 CaLCuV-inoculated whole plants except for cotyledons at 3, 6, 9, 13, and 16 dpi.

## Flow cytometry

Flow cytometry profiles were generated as described[14] with some modifications. We collected around 0.3 g rosette leaves from mock-treated plants with distinct genotypes at 14 dpi and finely chopped in 3 ml freshly made nuclear extraction buffer (45 mM MgCl$_2$, 30 mM sodium citrate, 20 mM pH 7.0 MOPS, 0.1% Triton X-100, 5 mM sodium metabisulfite, 5 μl/ml β-mercaptoethanol, 100 μg/ml RNaseA) and filtered through 40 μm cell strainer (Sigma) to release the nuclei. Nuclei were stained by adding 150 μl 1 mg/ml propidium iodide (Sigma) with gentle mixing by pipetting 10 times. Flow cytometry profiles were obtained from BD FORTESSA X-20 (College of Medicine Cell Analysis Facility, Texas A&M).

## Southern blot analyses

CaLCuV-infected plants were lysed in CTAB buffer (100 mM Tris-HCl pH 8.0, 20 mM EDTA, pH 8.0, 1.4 M NaCl, 3% cetyltrimethyl ammonium bromide, 2% β-mercaptoethanol) and then extracted with phenol:chlorophorm:isoamyl alcohol (25:24:1, pH 8.0) and precipitated with iso-propanol. Total DNA was treated with 50 μg/ml RNaseA and then purified with phenol:chloroform:isoamyl alcohol (25:24:1, pH 8.0). High quality DNA was obtained by precipitating with ethanol and sodium citrate. High quality DNA from plant was digested by EcoRI at 37 °C overnight. Digested DNA was transferred by capillarity to a Hybond-N membrane (GE Healthcare) and hybridized with $^{32}$P-labeled probe which is specific to CaLCuV DNA A. Probe for southern blot was amplified with primers listed in table S1 and then labeled using [α-$^{32}$P] 2′-deoxycytidine 5′-triphosphate (dCTP) (PerkinElmer) with Klenow fragment (3′−5′ exo-; NEB). Hybridization signals were obtained using Typhoon FLA 7000 (GE Healthcare) and visualized in Photoshop.

## Western blot analyses

Western blot analyses were performed as previously described[61]. Membranes were first incubated with antibodies against Myc (Sigma, C3956), H3 (Agrisera, AS10 710), or HA (Sigma, H9658) and then incubated with goat-developed anti-rabbit (GE Healthcare, NA934) or goat-developed anti-mouse immunoglobulin G (GE Healthcare, NA931). Membranes were developed with ECL+ and signals were obtained using ChemiDoc XRS+ and analyzed by Image Lab software (Bio-Rad).

Western blot analyses with anti-RAD51 and anti-RPA1A antibodies were performed as previously described[61] with minor modifications. For extracts from nuclei, membranes were first incubated with antibodies (1:1000) against RAD51 (PHYTOAB, PHY1804A) or anti-RPA1A (PHYTOAB, PHY1813S) overnight at 4 °C and then incubated with goat-developed anti-rabbit (1:5000) (GE Healthcare, NA934). For proteins eluted from immunoprecipitates, membranes were first incubated with antibodies (1:2000) against RAD51 (PHYTOAB, PHY1804A) or anti-RPA1A (PHYTOAB, PHY1813S) overnight at 4 °C and then incubated with goat-developed anti-rabbit (1:10000) (GE Healthcare, NA934).

## Strand specific RNA sequencing library preparation

Total RNA was extracted from virus-inoculated and mock-treated plants at 16 dpi with TRIzol and then treated with TURBO Dnase (Thermo Fisher, AM2238). Messenger RNA was enriched with Dynabeads™ Oligo(dT)$_{25}$ (Invitrogen™, 61005) according to the manufacturer's manual. First strand synthesis was completed with Random primer (Invitrogen™, 48190011), RNaseIn (Invitrogen™, AM2696), SuperScript™ III Reverse Transcriptase (Invitrogen™, 18080044) after fragmentation. Second strand synthesis was conducted with dUTP mixture (20 mM dUTP, 10 mM dATP, dCTP, and dGTP), RNase H (NEB, M0297), DNA polymerase I (NEB, M0209) and *E. coli* DNA ligase (NEB,

M0205). End repair was conducted with NEBNext Ultra™ End repair/dA-Tailling Module (NEB, E7442) followed by the adapter ligation with ligation mix and adapter (Illumina, TruSeq Kits, 15026773) according to the manufacturer's manual. Enriched mRNA from 1 μg total RNA was used as the starting material and 15 cycles were used to amplify the library.

## Chromatin Immunoprecipitation (ChIP) Assays

The ChIP assays were performed as previously described[4] with some modifications. We collected eight newly emerged rosette leaves of mock-treated and viral-infected plants with different genotypes. Three grams of materials were crosslinked with 1% formaldehyde for 20 min by vacuum infiltration at 4 °C, and the reaction was stopped with a final concentration of 100 mM Glycine by vacuum infiltration for 10 min at 4 °C. Plants were rinsed 5 times with pre-chilled sterilized water, flash-frozen in liquid nitrogen, and thoroughly grounded with TissueLyser II (QIAGEN, 85300). The powder was suspended in pre-chilled 6 volume (1.5 g with 9 ml) nuclei isolation buffer (15 mM PIPES-KOH pH 6.8, 0.25 M sucrose, 0.9% Triton X-200, 5 mM MgCl$_2$, 60 mM KCl, 15 mM NaCl, 1 mM CaCl$_2$, 1 mM PMSF, 1 pellet/50 ml EDTA free Protease inhibitor [Roche] and 25 μM MG132) on ice for 6 min. The mixture was filtered through two-layers of Miracloth and centrifuged at 11000 g for 10 min at 4 °C and carefully rinsed with 2 ml nuclei isolation buffer after discarding the supernatant. The white pellet was resuspended with 1 ml nuclei lysis buffer (40 mM Tris-HCl pH 8.0, 150 mM NaCl, 5 mM EDTA pH 8.0, 0.2% SDS, 0.1% Sodium Deoxycholate, 0.6% Triton X-100, 1 mM PMSF, 1 pellet/50 ml EDTA free Protease inhibitor [Roche] and 25 μM MG132) on ice for 6 min and then sonicated with 30 s ON and 60 s OFF for 16 cycles with Bioruptor Pico sonication device (Diagenode SA, B01060010) at 4 °C. Supernatant was collected after centrifuge for 10 min at 15000 rpm at 4 °C and diluted with the same volume nuclei dilution buffer (40 mM Tris-HCl pH 8.0, 150 mM NaCl, 5 mM EDTA. 0.2% Triton X-100, 1 mM PMSF, 1% Glycerol, 1 pellet/50 ml EDTA free Protease inhibitor [Roche] and 25 μM MG132). 100 μl diluted mixture was used as input, then the rest was split into two tubes with addition of 30 μl prewashed Protein A Agarose beads (Sigma, 11134515001), 3 μg anti-RAD51 (PHYTOAB, PHY1804A) and 3 μg anti-RPA1A (PHYTOAB, PHY1813S), and incubated at 4 with mild rotation for 6 hours. The beads-protein-DNA complex was washed as previously described[4]. Elution was repeated twice at 1200 rpm, 65 °C with 250 μl elution buffer (0.1 M NaHCO$_3$, 1% SDS) for 30 min. De-crosslinking was performed with 100 NaCl at 65 °C for 12 hours, followed by RNase A treatment and Protease K treatment. Purification was performed with phenol:chlorophorm:isoamyl alcohol (25:24:1, pH 8.0), precipitation of DNA was performed with 1/10 volume of 3 M sodium citrate (pH 5.2), 2.5 volume of ethanal and 1 μl GlycoBlue (Invitrogen ™, AM9516). DNA was dissolved in 1×TE buffer (10 mM Tris-HCl pH 8.0, 1 mM EDTA pH 8.0).

## Nuclear isolation and immunoprecipitation of RAD51 and RPA1A

Chromatin isolation was performed according to the ChIP assays above with minor modifications. The white pellet (isolated nuclear) was gently resuspended with 9 ml nuclei isolation buffer, centrifuged at 11000 g for 10 mins, and then resuspended with 3 ml lysis buffer (40 mM Phosphate buffer pH 8.0, 150 mM NaCl, 5 mM EDTA pH 8.0, 0.2% SDS, 0.1% Sodium Deoxycholate, 0.6% Triton X-100, 1 mM PMSF, 1 pellet/50 ml EDTA free Protease inhibitor [Roche] and 25 μM MG132) on ice for 6 mins. The supernatant was collected after centrifuging for 10 min at 15000 rpm at 4 °C and diluted with the same volume nuclei dilution buffer (40 mM Phosphate buffer pH 8.0, 150 mM NaCl, 5 mM EDTA. 0.2% Triton X-100, 1 mM PMSF, 1% Glycerol, 1 pellet/50 ml EDTA free Protease inhibitor [Roche] and 25 μM MG132) to perform immunoprecipitation.

Anti-RAD51 or anti-RPA1A were conjugated to protein A magnetic beads (Thermo Fisher, 88845) with BS3 crosslinker (C$_{16}$H$_{18}$N$_2$O$_{14}$S$_2$Na$_2$)

as previously described with minor modifications[62]. Protein A magnetic beads (150 μl) were washed with nuclei dilution buffer five times, incubated with 15 μg anti-RAD51 or anti-RPA1A antibodies in nuclei dilution buffer overnight at 4 °C, and then washed with nuclei dilution buffer. The antibodies were conjugated to the beads with freshly prepared crosslink solution with 5 mM BS3 at room temperature for 1 hr. The reaction was terminated with 60 mM Tris-HCl pH 7.5 at room temperature for 20 mins. Then the beads were washed with nuclei dilution buffer three times and used for immunoprecipitation with isolated nuclei (4 °C for 4 hours). Finally, the immunoprecipitates were washed with nuclear dilution buffer three times and eluted with elution buffer (5 mM EDTA, 200 mM $NH_4OH$) for western blot analysis.

## BiFC assay
The transient gene expression in protoplasts from Col-0 and *atxr5 atxr6* for BiFC assays were performed as previously described[63]. pBA-cYFP-RAD51 was co-expressed with pBA-nYFP-Rep and pBA-nYFP-HOP2 in the protoplasts. Sixteen hours after transfection, the fluorescence signals were captured and evaluated with Leica SP8 confocal microscope. The excitation light wavelength for YFP and chlorophyll autofluorescence was 514 nm and 633 nm, respectively. At least 15 individual protoplasts were examined for each transformation to obtain similar results.

## Quantitative PCR and qRT-PCR
The relative amount of viral DNA A in viral-infected plants and endogenous rDNA of *Arabidopsis* were examined by q-PCR. High-quality DNA was obtained as described in southern blot analyses. *Ubiquitin 10* served as an internal control for normalization. The enrichment levels of RAD51 and RPA1A on specific loci on viral genome and *Arabidopsis* after ChIP assays were also assessed by q-PCR. The q-PCR assays were performed with 10 μl system containing 5 μl SYBR Green Master Mix in 384-well plate. PCR conditions, signal detection, and quantification were performed as previously described[4].

Total RNA was extracted with TRI reagent (Sigma) from adult plants with eight true leaves. Normalized RNA was treated with Turbo DNases and then reverse-transcribed with reverse transcriptase (SuperScript III, Invitrogen). Reverse transcription was primed by oligo(dT). The following q-PCR assays were performed with 10 μl system containing 5 μl SYBR Green Master Mix in 384-well plate. PCR conditions, signal detection, and quantification were performed as previously described[4].

## ChIP sequencing library preparation
Half of ChIP-enriched DNA or one sixth of input DNA were used as starting materials for DNA end repair process for library construction with NEBNext Ultra™ End repair/dA-Tailing Module (NEB, E7442). Then the products were ligated to adapters (NEBNEXT Multiplex Oligos for Illumina, E7335, E7500 and E7710) with Blunt/TA Ligase Master Mix (NEB, M0367) according to the manufacturer's manual. For the library preparation, 9 cycles were used to amplify products from input DNA, 13 cycles were used to amplify RAD51 ChIP-enriched DNA and 16 cycles were used to amplify RPA1A ChIP-enriched DNA.

## Illumina sequencing and analysis
Libraries for strand-specific RNA sequencing were sequenced on the Novaseq system with pair-end 150 bp read length (Novogene). Adapters trimming, mapping, and counting process were performed as previously described[14]. Normalization and differential expressed analysis were calculated by DESeq2 (ver.3.15)[64]. The visualization for selected loci was shown in Intergrative Genomics Viewer (IGV)[65].

Heat-map clustering was performed based on sample-to-sample distances. To obtain the comprehensive DEG lists which contain both mock-treated and viral infected samples, we first compared mock-treated *atxr5 atxr6* to mock-treated Col-0 using Col-0 as a reference and then compared the other three triple mutants to *atxr5 atxr6* using *atxr5 atxr6* as a reference. The same analysis was also performed on viral-infected samples. Moreover, we also compared the viral infected Col-0 to mock-treated Col-0 using as a reference, then compared the other viral infected genotypes to their mock-treated samples. The DEGs were selected using the cut-off (Fold change > 2 and $p < 0.05$), but the genes with low expression (max reads > 10) were filtered. The pool of DEGs from the three groups mentioned above were the comprehensive DEG lists to perform heatmap clustering analysis.

Libraries for ChIP-seq were sequenced on the Novaseq system with pair-end 150 bp read length (Novogene). Adapter trimming and mapping processes were performed as previously described[14]. The leftover reads later were mapped by bowtie2 (ver. 2.4.4)[66] with perfect matches using Arabidopsis genome TAIR10 (http://www.arabidopsis.org/) as a reference genome. Reads uniquely mapped to the genome and reads mapped to multiple locations in the genome were separately extracted by Samtools (ver. 1.15.1)[67] for distinct downstream peak calling analyses. MACS2 (ver. 2.2.5)[68] was used for ChIP-seq peak calling using default parameters, both narrowPeak and broadPeak were called for each sample. We used the reads mapping to the genome which contains sequence mapped to multiple locations in the genome to perform the ChIP signal profile, peak number, peak reads percentage analysis and comparative analysis with DESeq2 (ver.3.15)[64] since the analysis with uniquely mapped reads barely detected signals in the majority over-replicated regions that harbor many repetitive sequences in *atxr5 atxr6*[9].

Peaks with $P$-value < 0.05 were selected for the following analysis, narrowPeak and broadPeak were merged by bedtools[69] (ver. 2.29.2). Annotation of peaks was conducted using bedtools. with TAIR10 gtf file as a reference. Peaks from biological replicates for the same genotype were merged and classified into different categories to calculate the relative peak number percentage.

The peaks belonging to distinct categories were first extracted with Samtools[67], and then reads over peaks from distinct categories were counted with Subread featureCounts (ver. 2.0.0)[70]. Total reads of peaks from specific categories were divided by total reads of peaks from all categories to generate peak reads percentages for individual specific categories. Normalization of the ChIP-seq signals was performed with concepts adopted from THOR[71]. Here, mitochondrial DNA was selected as "housekeeping" controls for the normalization of reads. Be noted that the reads mapping to mitochondria were relatively stabilized from 1.92% to 2.11% of total reads among our input samples and thus could serve as internal "housekeeping" controls. Furthermore, none of 16865 RAD51-enriched or 1675 RPA1A-enriched peaks were mapped into the mitochondrial genome, indicative of successful enrichment of RAD51 and RPA1A-bound chromatin in nuclei). The normalization based on mitochondrial (mt) DNA was performed first by counting mitochondrial reads in each sample. The percentage of the mtDNA reads in total mapped reads were listed in Supplementary Table 3. The scaling factor for each sample was calculated as the ratio of mtDNA reads /that of maximum mtDNA reads among all IP samples. Scaling factors were then applied to perform normalization by multiplying with total mapped read counts. Be noted that the ratios of mtDNA input in all samples were essentially the same and thus not used for the normalization. All downstream statistical analyses and plotted graphs were generated by R (ver. 4.0.2) and ggplot2[72]. All ChIP-seq profiles for RAD51, RPA1A, and H3K27me1 were drawn by deeptools (ver. 3.7.4)[73] using the default parameters. DESeq2 (ver.3.15)[64] with default parameters ($P$-value < 0.05) was implemented to output differential peaks between different groups of ChIP-seq samples. All violin plots were plotted by ggviolin package. All the plots and profiles for ChIP-seq were drawn by R. The visualization for selected loci was shown in Intergrative Genomics Viewer (IGV)[65].

## Yeast two-hybrid (Y2H) assays

Y2H were performed using the Gold Yeast Two-Hybrid System according to manufacturer's manual (Clontech). CDSs from different genes with the combination of pGADT7 and pGBKT7 were co-transformed into the yeast strain AH109. The yeast transformants were simultaneously plated on medium minus Leu, Trp, His, Ade to screen the interactions and medium minus Leu, Trp as controls under 28 °C.

## Co-immunoprecipitation (Co-IP) assays

GV3101 with tested constructs were co-transformed into *N. benthamiana* leaves by agroinfiltration. Materials were collected at two days post-transformation, grounded in liquid nitrogen, and stored at −80 °C. Total proteins were extracted from 0.35 g powder in 1.0 ml IP buffer and then centrifuged twice at 15,000 g for 15 min at 4 °C. IP buffer for HA-RPA1A and FM-Rep, HA-Rep and FM-RAD51, HA-RPA1A, and FM-RAD51 was 40 mM Tris-HCl pH 8.0, 150 mM NaCl, 5 mM $MgCl_2$, 1% Glycerol, 0.6% Triton X-100, 1 mM PMSF, 1 pellet/50 ml EDTA free Protease inhibitor [Roche] and 25 µM MG132. IP buffer for HA-BRCA1 and FM-RAD51, HA-HOP2 and FM-RAD51, HA-CYCB1 and FM-RAD51 was 40 mM Tris-HCl pH 7.0, 150 mM NaCl, 5 mM $MgCl_2$, 1% Glycerol, 0.6% Triton X-100, 1 mM PMSF, 1 pellet/25 ml EDTA free Protease inhibitor [Roche] and 50 µM MG132. Anti-HA-agarose beads or anti-FLAG-M2 magnetic beads were added to total protein extracts and then incubated at 4 °C for 2 h. Turbo DNases (10 U/ml) were added to IP buffer prior to incubation to conduct DNase treatment. The beads were washed four times at 4 °C for 5 min with IP buffer after incubation, then boiled with SDS loading buffer at 95 °C. Western blot analyses were carried out as described above with anti-Myc (Sigma, C3956) or anti-HA (Sigma, H9658) to detect IP products and co-precipitation levels of their partners.

## Quantification and statistical analysis

The images of Southern blot were quantified with Gel-Pro Analyzer (Media Cybernetics). For RNA-seq and ChIP-seq, an DESeq2 (version 3.15)[64] package was used to normalize gene expression levels with trimmed mean of M-values according to the false discovery rate. The significance cutoff for RNA-seq was Log2 FC > 1 or Log2 FC < −1, P-value < 0.05, and for ChIP seq, the cutoff was Log2 FC > 0.5 or Log2 FC < −0.5, P-value < 0.05.

For q-PCR, the data were presented as means of at least two replicates ± SD. For Figs. 2g, 3g, 5f, 5h, 6a, 6b, 6k, 7g, Supplementary Fig. 1d, Supplementary Fig. 3d, Supplementary Fig. 7c, the relative amount of viral DNA was initially normalized to that of *UBQ10*, and then to WT (Col-0) where the ratio was arbitrarily assigned a value of 1 with ±SD (n = 3, biologically independent replicates). For Fig. 5f, Supplementary Fig. 11a and Supplementary Fig. 14e the amount of rDNA was normalized to that of *UBQ10*. For the quantification of viral titers in Supplementary Fig. 2c, e, we first normalized viral DNA amount to the internal control *Tubulin* in individual samples, and then arbitrarily assigned the internally normalized viral titer of replicate 1 of virus-infected Col-0 as a value of 1 for each of the time points of 3, 6, 9, and 13 dpi, respectively.

## Graph drawing

Go enrichment analyses were performed using Metascape[74]. PPI network prediction was performed in STRING[75] and polished in Cytoscape[76]. Graphs with dot plots (individual data points) were drawn using GraphPad Prism 9.

## Reporting summary

Further information on research design is available in the Nature Portfolio Reporting Summary linked to this article.

## Data availability

RNA-seq related data were mined from GSE77735[13]. H3K9me2 and H3K27me1 related data were mined from GSE111814[14]. H3K27Ac related data were mined from GSE146126[11] and H3K4me3 related data were mined from GSE166897[15]. RAD51 and RPA related data were mined from GSE143582[37]. RNA-sequence data and ChIP-sequence data generated during this study has been deposited in GEO database (the accession number is GSE235158). Source data are provided with this paper.

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

## Acknowledgements

We thank D. Shippen for *atm*, *atr*, *sog1* seeds. The work was supported by grants from the NIH (GM127742) and NSF (MCB 2139857) to X.Z.; NIH (R35GM130272) to S.E.J.; NIH (R35GM128661) to Y.J.; Z.W., C.Z., Z.L., and K.X. were partially supported by China Scholar Council fellowships; S.E.J. is an investigator of the Howard Hughes Medical Institute.

## Author contributions

X.Z. conceived the project. Z.W., CC-G., and X.Z. designed the experiments. Z.W. performed most of the experiments, conducted the bioinformatic analysis for RNA-seq and analyzed the majority of data. CC-G. performed the early viral pathogenesis experiments and offered guidance and suggestions through the study. C.Z. performed the initial viral pathogenesis screening of epigenetic mutants. Z. L. genotyped mutants and validated the viral resistance phenotype. K.X. helped to generate high-order mutants. C.T. conducted the bioinformatic analysis for ChIP-seq with the guidance of C.L. J.Z. helped RNA-seq library construction and drew the model. S.Z. helped capture the confocal image. Z.S.W., S.E.J., S.D.M., and Y.J., provided a series of epigenetic mutants and provided intellectual advice. X.P. provided intellectual advice. Z.W. wrote the initial draft of the manuscript. X.Z. thoroughly edited the paper.

## Competing interests

The authors declare no competing interests.
