## [Peer Review File · Nature Communications]

H3.1K27me1 loss confers Arabidopsis resistance to Geminivirus by sequestering DNA repair proteins onto host genomeReviewer #1 (Remarks to the Author):

In this manuscript, Wang et al. investigated the impact of epigenetic modifications on plant-virus interaction using *Arabidopsis thaliana* as a host and the Geminivirus Cabbage Leaf Curl Virus (CaLCuV). They found that loss of function mutations in the H3 methyltransferases ATXR5 and ATXR6, which deposit H3.1K27me1 to heterochromatin, result in robust Geminivirus resistance. To investigate the underpinning molecular mechanism, the authors employed a plethora of molecular biology techniques ranging from yeast two-hybrid to RNA and Chromatin Immunoprecipitation Sequencing, and studied CaLCuV infection in the corresponding single, double, and triple mutant *Arabidopsis* plants. The cornerstone of their model (Fig 6h) is that the viral protein Rep hijacks RAD51 and RPA1A for efficient virus replication in the wild type (Col-0) plants. In the in *atxr5 atxr6* double mutant, unstable host DNA and defence-related genes retain a large amount of DNA repairing machinery including RAD51 and RPA1A, leading to reduced virus replication. However, the data supporting this hypothesis are mostly indirect. The only direct evidence is an immunoprecipitation essay (Fig 5f), which shows very little difference in the loading efficiency of RAD51 and RPA1A on viral genome between the infected Col-0 and *atxr5 atxr6*. In addition, further clarification is required to explain the corresponding data. To verify their model, the authors could generate Rep mutants with reduced RAD51 and RPA1A binding and analyse the composition of Geminivirus mini-chromosomes by mass spectrometry.

Additional comments:

Statistical analyses are missing in most figures!

Better introduction to Geminivirus replication cycle would help non-expert readers (minichromosomes etc.)

Page 2 lane 91 typo "a still a"

Transfection needs further clarification (Agrobacterium delivers t-DNA, not plasmid). What was detected in the Extended Data Fig2- integrated T-DNA or remaining bacterium DNA on the leaf surface including plasmid DNA? Reporter gene expression analysis, such as GUSint or GFPint would be a better method to assess the impact of *atxr5 atxr6* on agrobacterium-mediated gene transfer. Additionally, it is not indicated whether inoculated or systemic tissues were used. Furthermore, the presented data seems odd, since one would expect an increasing amount of viral DNA in Col-0 due to viral amplification and not more or less a flat line.

Page 6, lane 178. "Intriguingly, *mbd9 atxr5 atxr6*, *sac3b atxr5 atxr6* and *brca1 atxr5 atxr6* mutants all showed a similar ratio of symptomatic plants relative to Col-0" The conclusion for *mbd9 atxr5 atxr6* and *brca1 atxr5 atxr6* is not supported by the data presented in Fig1f.

It is not clear why southern-blot was used to quantify viral DNA level and not qPCR, the latter would be quantitative opposed to the former.

The authors should refrain from using the expressions such as comparable and similar. Use statistics.

Page 7, lane 183. "Susceptibility of *brca1 atxr5 atxr6*, *mbd9 atxr5 atxr6* and *sac3b atxr5 atxr6* mutants was attributed to effect of BRCA1, SAC3B and MBD9 on molecular features in *atxr5 atxr6*, but not to their distinct functions in the wild-type condition." Mdb9 shows higher susceptibility and viral titre (Fig 1fg compared to wild type). Hence the conclusion does not seem to be valid.

Page 7, lane 185. "Given that three triple mutants had similar viral infection profiles, thus, heterochromatin amplification and TE reactivation were not directly responsible for Geminivirus resistance in *atxr5 atxr6* mutants." Since the triple mutants had different heterochromatin amplification profile (Fig 1c) and TE amplification pattern (Fig 1d), it is not clear how the above conclusion can be made.

Why *sog1* was chosen for the analysis in Fig2? It has not been introduced.

Why the percentage of Col-0 symptomatic plants at 11 dpi differs between Fig 1 and Fig 2? (80 vs 30%)

Page 9, lane 245, CHR31 or TK1A were not introduced

Page 9, lane 248. "Given that *atxr5 atxr6* and *atxr5 atxr6 rad51* had the same viral resistance phenotype, we concluded that RAD51 and RPA1A were downstream effectors that accounted for reduced viral replication in *atxr5 atxr6*." However, RAD51 and RPA1A are overexpressed in the *atxr5 atxr6* double mutant (Fig 3d). Consequently, they should promote viral replication and not reduce it. Please clarify.

What does inf mean in Extended data Fig 8?

Page 10, lane 75. Typo figure 4C

Page 11, lane 301. "Collectively, these results strongly support a model where unstable DNA in *atxr5 atxr6*, especially rDNA loci, requires RAD51 and RPA1A, among other HRR factors, to maintain genome integrity." Unjustified conclusion since genome integrity was not investigated.

Page 12, lane 344. Awkward sentence: "This result clearly resulted from that the fact that.."

It is not clear how the authors calculated RAD51 and RPA1A loading on the viral genome (Fig 5f).

Extended data Fig 8. Why did the author use different constructs for the overexpression lines? FLAG-MYC FLAG-MYC-RAD51 in Col-0 or Myc-RAD51 in *atxr5 atxr6*.

Reviewer #2 (Remarks to the Author):

In this manuscript, Wang et al., discovered increased viral resistance in *atxr5 atxr6* compared with Col. They found that Rep interacts with RAD51 and RPA1A, and might hijack them for virus replication. Because in *atxr5 atxr6* mutant RAD51 and RPA1A are enriched at rDNA loci, this together with the less binding of RAD51 at viral DNA makes the authors propose that in *atxr5 atxr6*, unstable host DNA and defense-related genes retain a large amount of DNA repairing machinery to prevent its routing to viral genome, leading to low efficient replication of the virus. However, the data provided cannot support the conclusions regarding the molecular mechanisms, and also I have several concerns regarding the biological significance of the work and data analyses.

Major concerns

1. Line 40-42. Current results in this work only suggest that in a special case (*atxr5 atxr6* mutant), impaired genome stability might suppress viral infection. To demonstrate the claim that host genome could retain DNA repairing proteins via sacrificing its genome stability to suppress viral infection, relevant data need to be provided in virus-infected wild type plants, otherwise the biological significance of this work is missing.
2. Line 88-90. Results in reference 15 (Davarinejad et al., 2022, Science) show that loss of RAD51 in *atxr5 atxr6* does not enhance heterochromatin amplification.
3. It would be necessary to examine the DNA re-replication and TE-reactivation phenotypes in *atm;atxr5;atxr6* mutant to further decipher the connection between *atxr5;atxr6*-induced phenotypes, DDR and viral resistance.

4. The figure legends of Extended Data Fig. 4c, d does not explain what exactly is the difference between 4c and 4d. In addition, what is the "mock" condition in viral infection? Can an experiment from previous work (GSE77735) be considered "mock"?

5. Line 231-232. Are these 20 selected DDR factors differentially expressed in RNA-seq analysis? It is not clear what is the purpose of Extended Data Fig. 7a, in which transcripts of only six genes are shown. Moreover, PARP2 and CHR31 were not tested in Y2H assay, and PARP2 and TK1A are repeatedly shown in Figure 2c and Extended Data Fig. 7a.

6. Extended Data Fig. 7b. The interaction between REP and RAD51 was tested twice in yeast (4th row and 17th row), but different results were generated. This raises concerns regarding the quality of the data.

7. Figure 3f. There are two dark grey columns in Figure 3f.

8. Line 244-245 and Extended Data Fig. 7d. It seems that chr31 and tk1a mutants are less susceptible to viral infection, please provide statistics to support the conclusion that their mutations did not affect viral pathogenesis.

9. Line 256-257. It is not clear how this conclusion is reached. Results in Figure 3 only show that RAD51 and RPA1A associate with Rep and viral genome and are required for viral infection, but there is no evidence to support their participation in virus replication.

10. Line 260-262 and Extended Data Fig. 8a-c. It is very confusing regarding the antibody test here. Based on line 724-725, ChIP experiments were performed with anti-RAD51 and anti-RPA1A antibodies. But figure legends of Extended Data Fig. 8a-c do not provide any information in terms of how the antibody tests were performed and seem that Myc and HA but not RAD51 and RPA1A antibodies were used for blotting. 1) Extended Data Fig. 8a, what's the purpose of examining FLAG-Myc-RAD51 and Myc-RAD51 with Myc antibody here? 2) Extended Data Fig. 8b, these blots do not show the quality/specificity of the antibody. Why not simply use RAD51/RPA1A antibody for blotting with Col and rad51/rpa1a mutant? Since ChIP was performed with Col plants but not with these Myc-tagged lines. 3) Extended Data Fig. 8c, what's the purpose of detecting HA IPed HA-RPA1A with HA antibody? Overall, these antibody testing results raise concerns regarding the quality of RAD51 and RPA1A antibodies used in this study.

11. Line 264-266 and Extended Data Fig. 9a, b. Increased number of peaks in heterochromatin for what? RAD51 or RPA1A? But in Extended Data Fig. 9a, b RAD51 is clearly enriched at euchromatin, and RPA1A data are not shown. Also, it is not clear whether these data are derived from Col mock or any other conditions.

12. Line 273-278, Fig. 4c and Extended Data Fig. 9c. Data here does not tell whether RAD51 and RPA1A are enriched at any specific elements or heterochromatin or euchromatin as the percentage of different elements on the Arabidopsis/mouse genome is not provided and compared. Based on the text, RAD51 and RPA1A are localized at both heterochromatin and euchromatin, how does this general protein distribution pattern suggest conservative functions in eukaryotes?

13. Fig. 4e, f. Please describe what is "overall", "rDNA" and "PCG". Are they all elements, all rDNA loci and all PCG in the genome or RAD51/RPA1A-enriched "overall", "rDNA" and "PCG" in Col/atxr5 atxr6?

14. Extended Data Fig. 10a, b. Again, please define "ncRNA" and "TE". I assume the left half panels show RAD51 and RPA1A profiles and the right half panels show H3K27me1 profiles. How could the patterns of RAD51 and RPA1A at ncRNA/TE, and H3K27me1 at ncRNA be almost exactly the same except for different scales?

15. Line 290-295. Based on the text, I assume that RAD51 and RPA1A were significantly enriched on two 45S rDNA loci, which are localized at chromosomes two and four (Benoit et al., 2013, Gene), but Fig. 4g shows a locus at chromosome 3.

16. Overall, all the RAD51 and RPA1A ChIP-seq analyses need to be performed in a more proper way. For example, although the rDNA loci might have more RAD51 and RPA1A enriched in atxr5 atxr6, they also have more DNA copies in atxr5 atxr6. This is clear in Fig. 4g (and also in Extended Data Fig. 10c, d) that except for H3K27me1, levels of all others including H3K9me2 and Input were increased in atxr5 atxr6. Therefore, the RAD51 and RPA1A ChIP-seq signals should be normalized to Input but not using RPKM to eliminate the influence of higher copy number background in atxr5 atxr6, so to test whether RAD51 and RPA1A are really more enriched at these loci in atxr5 atxr6 compared with Col. In addition, in Fig. 4g and 4i, it may not be necessary to use the same scale for different marks/Input as long as Col and atxr5 atxr6 share the same, in Fig. 4i the input is too low to judge if there is a difference in DNA content between Col and atxr5 atxr6.

17. Line 298-303. It is not clear what's the purpose of mentioning mbd9 atxr5 atxr6 and sac3b atxr5 atxr6 mutants. To support that unstable DNA in atxr5 atxr6 requires RAD51 and RPA1A to maintain genome integrity, the authors need to show that loss of RAD51 or RPA1A in atxr5 atxr6 causes even more unstable or copy numbers of DNA, but results in reference 15 (Davarinejad et al., 2022, Science) do not seem to support this idea.

18. Extended Data Fig. 11b, c. Please indicate how many protein-coding genes are enriched with RAD51 and RPA1A, and why RAD51-enriched top 3000 genes but not all RAD51-enriched genes were used for GO analysis.

19. Fig. 4h. Please perform statistical analyses.

20. Line 305-334. If RAD51 is generally required for active transcription, why it is selectively enriched at genes in certain pathways? Please provide an explanation for this.

21. Line 338-340. If HRR factors (presumably refer to RAD51 and RPA1A) are sequestered at unstable DNA regions in atxr5 atxr6, why their enrichment was not increased at TEs, as TEs are enriched at heterochromatin that becomes unstable in atxr5 atxr6. Please provide an explanation for this.

22. Line 341-344 and Extended Data Fig.12a. Please explain how representative samples were randomly collected at an earlier stage. Also, these results should be normalized to input as that in Fig. 3c, otherwise the reduced RAD51 enrichment on viral DNA could be due to the reduced viral DNA copy numbers in atxr5 atxr6 compared with Col, as the authors also mentioned in line 344-345.

23. Line 349-352. Please explain why in Col the decreased signals of RAD51 and RPA1A mainly originated from the PCG loci rather than rDNA regions. DNA damage rates should be the same/random at PCG and rDNA in Col, and if virus hijacks RAD51 and RPA1A, how could it selectively take them from PCG but not rDNA in Col.

24. Line 369-373 and Fig. 5b. Although RAD51 enrichment at PCG in atxr5 atxr6 was not reduced by viral infection like that in Col, a clear reduction in RAD51 enrichment at rDNA was observed in atxr5 atxr6 but not in Col. Therefore, the current data cannot prove that there is an overall less loss of RAD51 on host genome in atxr5 atxr6 compared with Col after virus infection, and this cannot be simply deduced by the higher RAD51 occupancy on DNA in atxr5 atxr6 compared with Col as it already happens in mock due to unstable DNA and RAD51 overexpression. Moreover, in my opinion the reduction of RAD51 enrichment at rDNA (heterochromatin) in atxr5 atxr6 after infection rather implies that the RAD51 occupancy at rDNA (heterochromatin) in atxr5 atxr6 does not contribute to its viral resistance.

25. Line 374-382 and Fig. 5f. Results in Fig. 5f should be performed with at least three biological replicates and statistical analysis should also be performed. Moreover, without clear proof that there is less loss of RAD51 on host genome in atxr5 atxr6 compared with Col after virus infection, the reduced RAD51 enrichment at viral DNA could be caused by indirect reasons that affect virus infection in atxr5 atxr6.

26. Line 383-386 and Fig. 5g-i. RAD51 is already overexpressed and more enriched at

heterochromatin in atxr5 atxr6 compared with Col, and even though this does not fix the heterochromatin amplification. One would expect that the overexpressed RAD51 (by 35S promoter) would also be more enriched at heterochromatin in atxr5 atxr6 compared with Col. However, overexpressing RAD51 (by 35S promoter) caused similar effects on viral infection in Col and atxr5 atxr6, this probably again rather suggests that RAD51 enrichment at heterochromatin in atxr5 atxr6 does not contribute to its viral resistance. In addition, different tags were fused with RAD51 and different transgenes were used in Col and atxr5 atxr6 (likely resulting in varied RAD51 function and overexpression levels), making their direct comparison impossible.

27. Line 397-399 and Fig. 6b. It is not clear how the HOP2-RAD51 Co-IP is performed. Based on Fig. 6b, it seems that HOP2 interacts with Rep but not RAD51.

28. All results such as "percentage of symptomatic plants", "relative amount of viral DNA" and "ChIP-qPCR" lack statistics, and some of them lack enough biological replicates.

Minor concerns

1. Please indicate what is "dpi" in figure legends.
2. Line 147-149. The cited reference 32 describes the phenotypes of h3.3 but not h3.1 mutant.
3. Extended Data Fig. 4b is not cited in the text.
5. Line 233-234. Extended Data Fig. 7c probably should be 7b.
6. Extended Data Fig. 10d is not cited in the text.

Reviewer #3 (Remarks to the Author):

I read the manuscript of Wang et al. with great interest. The authors report on the very surprising fact that the atxr5/atxr6 double mutant is resistant to geminiviral infection. The double mutant is deficient in the deposition of H3.K27me1 to heterochromatin. The mutant phenotype results in genome instability, hyperrecombination, the activation of transposable elements, the amplification of heterochromatic regions and accumulation of DSBs.

As a consequence, factors involved in DNA repair are recruited to these genetically unstable regions, especially the strand exchange protein Rad51 and RPA1A, required for homologous recombination. Geminiviruses on the other side rely on both proteins for their replication and the authors demonstrate an interaction of both with the viral rep protein. Thus, the available pool of HR factors required for virus replication is titrated down in the mutants and the plants acquire resistance to infection.

This is a fascinating story and as far as I see the authors were able to sustain every single step in their proposed model by experiments. Besides other findings they could show that indeed less Rad51 and RPA1A are loaded on viral replicons in the double mutant than in WT. Overexpression of Rad51 in the double mutant reduced resistance. Recruitment of Rad51 to the unstable genomic regions but not to the viral DNA depends on BRCA1, CYCB1;1 and HOP2. In their absence the resistance phenotype is reduced. I am really impressed by the wealth of data produced by the authors. I do not see any flaws, the paper is well written, all conclusions are justified. A very complete story that should be published without further ado.

Reviewer Comments:
Reviewer #1:

In this manuscript, Wang et al. investigated the impact of epigenetic modifications on plant-virus interaction using *Arabidopsis thaliana* as a host and the Geminivirus Cabbage Leaf Curl Virus (CaLCuV). They found that loss of function mutations in the H3 methyltransferases ATXR5 and ATXR6, which deposit H3.1K27me1 to heterochromatin, result in robust Geminivirus resistance. To investigate the underpinning molecular mechanism, the authors employed a plethora of molecular biology techniques ranging from yeast two-hybrid to RNA and Chromatin Immunoprecipitation Sequencing, and studied CaLCuV infection in the corresponding single, double, and triple mutant *Arabidopsis* plants. The cornerstone of their model (Fig 6h) is that the viral protein Rep hijacks RAD51 and RPA1A for efficient virus replication in the wild type (Col-0) plants. In the in *atxr5 atxr6* double mutant, unstable host DNA and defence-related genes retain a large amount of DNA repairing machinery including RAD51 and RPA1A, leading to reduced virus replication. However, the data supporting this hypothesis are mostly indirect. The only direct evidence is an immunoprecipitation essay (Fig 5f), which shows very little difference in the loading efficiency of RAD51 and RPA1A on viral genome between the infected Col-0 and *atxr5 atxr6*. In addition, further clarification is required to explain the corresponding data.

Response: Many thanks to this reviewer for thorough reviewing and many insightful comments on our ms.

Regarding the point of different loading efficiencies of RAD51 and RPA1 onto viral chromatin in WT vs *atxr5/6*, we think that the context was misunderstood, and we should have explained it more clearly!

As we stated on page 12, line 341 (in the original submission), “our initial ChIP-qPCR detected a significant enrichment of RAD51 and RPA1A in the viral genome in Col-0 but barely detected any signal in the *atxr5 atxr6* background when representative samples were randomly collected at an earlier stage (Extended Data Fig. 12a--the figure below). This result clearly showed that RAD51 ChIP signal over the virus in *atxr5 atxr6* was almost identical with negative control (ChIP performed with IgG) since *atxr5 atxr6* had significantly reduced infection rate and viral titers than Col-0”.

(Be noted: this was the original Extended Data Fig.12a)

We should have emphasized the figure and context and presented them in a clearer way so that this reviewer would have not missed the results. Here we re-did the ChIP-qPCR with the samples collected at an early stage with three biological replicates and organized the data by including statistical significance. We calculated the loading efficiency of RAD51 by normalizing ChIP signal to the viral titers, and the data is now presented in the new Figure 5a in the main text. Indeed, there is a significant difference in loading efficiency of RAD51 on the viral genome between Col-0 (average RAD51 ChIP-signal /viral input 1.69%) and *atxr5 atxr6* (average 0.48%)—this reflected ~3.5 fold loading difference between Col-0 and *atxr5 atxr6*.

New Fig. 5a. ChIP-qPCR assays showed that RAD51 enrichment was significantly increased over rDNA but decreased over viral DNA in *atxr5 atxr6* than that in Col-0 when the samples were randomly collected.

We think that the misunderstanding might have come from the original Figure 5f where the RAD51 and RPA1A loading efficiencies on the viral chromatin between Col-0 and *atxr5 atxr6* appeared to be small. However, Figure 5f was in the original context described in lines 345 -347, page 12: “To account for differences in virus titer, we arbitrarily (purposely) increased the number of *atxr5 atxr6* symptomatic plants to artificially mimic the infection ratio of Col-0 for (convenience of) ChIP-seq assays to test distribution of RAD51 and RPA1A among virus and host”.

Here we would like to reiterate the context and Figure 5f: The purpose of this experiment was to identify the precise loci of RAD51 and RPA1A binding onto the host and viral chromatin in WT and *atxr5/6*. However, RAD51 and RPA1A are barely loaded onto viral chromatin on the *atxr5/6* mutant in a normal situation (*atxr5 atxr6* mutants displayed a significantly lower number of virus-infected plants).

To circumvent this issue, we purposely increased the number of virus-infected *atxr5 atxr6* plants to reach that of virus-infected Col-0 (Otherwise, we could not run RAD51 and RPA1A ChIP-seq for the mutant due to low viral titers). Through this strategy, we could pinpoint the binding loci of two proteins onto host and viral chromatin in WT and *atxr5/6*. However, the tradeoff was that the binding capacity of RAD51 and RPA1A onto the viral genome was exaggeratedly increased in *atxr5 atxr6* than the actual situation (the viral DNA amount in *atxr5 atxr6* from new Fig 5a was around 12% of that in Col-0 whereas the viral DNA amount in *atxr5 atxr6* from new Fig 5b, was arbitrarily increased to around 80% of that in Col-0).

In the other word, the purposely increased numbers of symptomatic *atxr5 atxr6* would mask the true difference of the RAD51 binding signals onto the viral chromatin between Col-0 and *atxr5 atxr6* as shown in new Fig. 5b. To avoid future misunderstandings, we would like to present the ChIP-qPCR data for representative samples first and then emphasize the rationale and trade-off of purposely increasing the number of virus-infected *atxr5 atxr6* plants (line370 to 373 in the new ms), and also the details about collecting samples (line 738 to 744 in the new ms) in the revision.

New Fig. 5b. ChIP-qPCR assays showed that the loading of RAD51 over rDNA was still significantly increased but decreased over viral DNA in *atxr5 atxr6* than that in Col-0 when the ratios of symptomatic plants were comparable in two backgrounds.

Importantly, to further address the criticism from this reviewer and reviewer #2, we have designed and conducted a new set of experiments by taking advantage of recently published transgenic lines expressing H3.1S28A. The H3.1S28A transgenic plants display heterochromatic DNA amplification and activation of DNA repair (Davarinejad et al., 2022, Science) and increased rDNA (new Fig. 6e) compared to Col-0. Importantly, the transgenic plants showed robust virus resistance against CaLCuV. Thus, the new result (in new Fig. 6e-h) provided a new piece of direct evidence to consolidate our model that unstable genomic elements will retain HRR factors and prevent them from being recruited onto the viral genome to suppress viral DNA amplification.

New Fig. 6 e-h, Transgenic plants (S28A) with heterochromatin amplification displayed viral resistance phenotype, mimicking *atxr5 atxr6*.

Davarinejad, H., Huang, Y. C., Mermaz, B., LeBlanc, C., Poulet, A., Thomson, G., Joly, V., Munoz, M., Arvanitis-Vigneault, A., Valsakumar, D., Villarino, G., Ross, A., Rotstein, B. H., Alarcon, E. I., Brunzelle, J. S., Voigt, P., Dong, J., Couture, J. F., and Jacob, Y. The histone H3.1 variant regulates TONSOKU-mediated DNA repair during replication. *Science* 375, 1281-1286 (2022).

Some suggestions are:

1. To verify their model, the authors could generate Rep mutants with reduced RAD51 and RPA1A binding and analyse the composition of Geminivirus mini-chromosomes by mass spectrometry.

Response: We thank this reviewer for sharing the idea. But we felt that it is not doable. First of all, replication-associated protein (Rep, or C1 and AL1) is a highly conserved protein across the family *Geminiviridae*, and essential for viral DNA replication (Hanley-

Bowdoin et al., 2013, Nat Rev Microbiol). Mutation of the Rep protein abolishes the virus DNA replication (Elmer et al., 1988, Nucleic Acids Res). Thus, CaLCuV mutants that encode mutated Rep with compromised binding with RAD51 and RPA1A will unlikely be able to infect *Arabidopsis*. Second, while we are a biochemistry lab, we do not feel that it would be a feasible strategy to isolate Geminivirus mini-chromosomes and then conduct quantitative mass spectrometry to detect RAD51/RPA1A in the minichromosome for three reasons: 1) RAD51 and RPA1A were expressed at relative low levels (new Fig. 3d); 2) these proteins are not stably incorporated into chromatin; and 3) the viruses are barely detected in the *atxr5/6* mutant.

The advice from this reviewer, however, prompted us to pursue two alternative approaches. Briefly, to investigate whether the interaction between Rep and HRR factors was interrupted in *atxr5 atxr6*, we first co-transfected nYFP-HOP2, which is one of the HRR factors to repair the DNA damage of *Arabidopsis*, and cYFP-RAD51 into protoplasts of Col-0 and *atxr5 atxr6*. We then evaluated YFP signal intensity as a proxy of their interaction. The complementation of nYFP-HOP2 and cYFP-RAD51 showed YFP signal in the nuclei. Indeed, a stronger YFP signal was clearly and repeatedly detected in the protoplast of *atxr5 atxr6* than that in Col-0 (new Fig. 6a, b and Extended data Fig.13d). This result indicated that increased HRR factors were indeed recruited to repair the DNA damage caused by the unstable heterochromatic elements in *atxr5 atxr6*. Such a process will in turn interferes with the interaction between RAD51 and viral protein Rep.

Fig. 6 a, b BiFC assays showed that mutations of ATXR5/6 promoted the interaction between RAD51 and HOP2, one of HRR factors.

In parallel, we assessed the direct interaction of RAD51 and Rep using BiFC assays in the protoplast. The results showed that the interaction between RAD51 and Rep was significantly suppressed in *atxr5 atxr6* compared to that in Col-0 (new Fig. 6c, d and Extended data Fig. 13e). We have vigorously examined these interactions by providing a large number of biological repeats. These new results further strengthened our model that HRR factors are efficiently sequestered onto the genome of in *atxr5 atxr6* to repair the increased DNA damage, preventing them from being used by the viral Rep for pathogen replication. The new data is now presented in new Fig. 6a-d.

Fig. 6 c, d BiFC assays showed that interaction between RAD51 and Rep was compromised in *atxr5 atxr6* vs Col-0.

Elmer, J. S., Brand, L., Sunter, G., Gardiner, W. E., Bisaro, D. M., and Rogers, S. G. (1988). Genetic analysis of the tomato golden mosaic virus. II. The product of the AL1 coding sequence is required for replication. *Nucleic Acids Res* 16, 7043-7060.

Hanley-Bowdoin, L., Bejarano, E. R., Robertson, D., and Mansoor, S. (2013). Geminiviruses: masters at redirecting and reprogramming plant processes. *Nat Rev Microbiol* 11, 777-788.

2. Statistical analyses are missing in most figures!

Response: Thanks. We have provided statistical analyses for most figures now.

3. Better introduction to Geminivirus replication cycle would help non-expert readers (minichromosomes etc.)

Response: We added the following sentences in line 62 to 65 (new ms).

“Geminiviruses pack their DNA with host-encoded histone octamers to form episomes that are named the viral mini-chromosome. Mini-chromosomes serve as intermediates for replication via rolling circle replication (RCR) and recombinant-dependent replication (RDR) (Jeske et al., 2001, EMBO J)”.

Jeske, H., Lutgemeier, M., and Preiss, W. (2001). DNA forms indicate rolling circle and recombination-dependent replication of Abutilon mosaic virus. EMBO J 20, 6158-6167.

4. Page 2 lane 91 typo “a still a”

Response: Corrected and thanks.

5. Transfection needs further clarification (Agrobacterium delivers t-DNA, not plasmid). What was detected in the Extended Data Fig2- integrated T-DNA or remaining bacterium DNA on the leaf surface including plasmid DNA? Reporter gene expression analysis, such as GUSint or GFPint would be a better method to assess the impact of atxr5 atxr6 on agrobacterium-mediated gene transfer. Additionally, it is not indicated whether inoculated or systemic tissues were used. Furthermore, the presented data seems odd, since one would expect an increasing amount of viral DNA in Col-0 due to viral amplification and not more or less a flat line.

Response: Thanks for the comment.

We now streamlined our writing to make it clear. Briefly, for geminivirus inoculation, we first extracted the plasmid from E. coli (DH5 α) that contains the geminivirus genome and transformed it into agrobacteria. Fresh agrobacteria containing geminivirus genome together with silicon carbide powder was sprayed by a pump onto the center of *Arabidopsis* (which were newly emerged rosette leaves) at 80 psi. During the process, some mild wounds were created, and agrobacteria penetrated plant tissues. We then sprayed sterilized water to wash away the agrobacteria together with silicon carbide powder 16 hours later. We have added the details in line 724-728 (new ms).

We performed the southern blot assay with a probe specific to the geminivirus genome and used the plasmid containing the geminivirus genome as technical control. So the bands on the top of southern blots in the new Extended data Fig. 2a were the plasmid containing the geminivirus genome.

We initially asked whether the agrobacteria with the geminivirus genome left in *atxr5 atxr6* was less than that in Col-0. The Southern blot and q-PCR assays in the Extended data Fig. 2 ruled out this possibility.

We collected whole plants only for experiments performed in Extended data Figure 2 since there were no systemic infected leaves at 3, 6, and 9 days post-inoculation. For all the other mock-treated and virus-infected samples, we collected systemically infected leaves according to the methods described in line 731-732 in the new ms: “To assess systemic infection, we harvested the eight newly emerged rosette leaves of CaLCuV-infected plants.”.

For the quantification of viral titers in new Extended data Fig 2c and 2e, we first normalized viral DNA amount to the internal control *Tubulin* in individual samples, and then arbitrarily assigned the internally normalized viral titer of replicate 1 of virus-infected Col-0 as a value of 1 for each of the time points of 3, 6, 9, and 13 dpi, respectively. We added the normalization details in line 971-975 (new ms).

6. Page 6, lane 178. “Intriguingly, *mbd9 atxr5 atxr6*, *sac3b atxr5 atxr6* and *brca1 atxr5 atxr6* mutants all showed a similar ratio of symptomatic plants relative to Col-0” The conclusion for *mbd9 atxr5 atxr6* and *brca1 atxr5 atxr6* is not supported by the data presented in Fig1f.

Response: Thanks for the comment.

The difficult issue is that virus infection results may vary among different time points, especially at the early stage when the virus-induced symptoms are not strong, leading to

the potential inaccuracy of phenotyping-counting. That would be likely the cases for 9 and 11 dpi in Fig 1f (in the original submission), however, the data at the later time point 13 dpi should be more convincing as the viral symptoms would be more obvious. That being said, we did not observe any statistically significant difference among Col-0, *sac3b atxr5 atxr6*, and *brca1 atxr5 atxr6* at 13 dpi.

We also repeated the experiment with three biological replicates for the following RNA-Seq and presented the results in the new Fig. 1f, g. We are confident in our conclusion.

New Fig. 1 f, g The loss of *MBD9*, *SAC3B*, or *BRCA1* could all restore viral susceptibility of *atxr5 atxr6*.

7. It is not clear why southern-blot was used to quantify viral DNA level and not qPCR, the latter would be quantitative opposed to the former. The authors should refrain from using the expressions such as comparable and similar. Use statistics.

Response: Thanks for the comment. This project has gone through several postdocs and graduate students in the 9 years as reflected in the long list of authorships. In the early stage, we typically used Southern blot assays to detect viral DNA levels. In the late stage, we switched to q-PCR as the price of ³²P-ATP has increased approximately 10-fold in the past few years.

We have performed the statistical analysis and rewrote the sentence.

8. Page 7, lane 183. "Susceptibility of *brca1 atxr5 atxr6*, *mbd9 atxr5 atxr6*, and *sac3b atxr5 atxr6* mutants was attributed to effect of *BRCA1*, *SAC3B* and *MBD9* on molecular features in *atxr5 atxr6*, but not to their distinct functions in the wild-type condition." *Mdb9* shows higher susceptibility and viral titre (Fig 1fg compared to wild type). Hence the conclusion does not seem to be valid.

Response: Thanks for the comment.

In Fig 1f, since we only had 2 replicates for *mbd9* but had 3 replicates for Col-0 at this particular time (due to the space limitation in the growth chamber), we did not perform the statistical test. Here, the seemingly increased titers in *mbd9* mutant might be due to variation in samples during the virus infection and Southern blot analysis. However, we

later repeated the experiment, and the results are listed in the new Fig. 1f, g. We are confident in our conclusion (Please also refer to the answer to question 6).

9. Page 7, lane 185. “Given that three triple mutants had similar viral infection profiles, thus, heterochromatin amplification and TE reactivation were not directly responsible for Geminivirus resistance in *atx5 atx6* mutants.” Since the triple mutants had different heterochromatin amplification profile (Fig 1c) and TE amplification pattern (Fig 1d), it is not clear how the above conclusion can be made.

Response: Thanks for the comment. This is the exact point of this work—it is DNA repairing proteins, not the heterochromatin amplification and TE reactivation, that account for the viral resistance phenotypes in *atx5/6* vs WT.

Genotype	DNA re-replication	TEs reactivation	Activation of DDR	Virus resistance
Col-0	No	No	No	No
atx5 atx6	Yes	Yes	Yes	Yes
brca1 atx5 atx6	Yes	Yes	No	No
mbd9 atx5 atx6	No	No	No	No
sac3b atx5 atx6	No	No	No	No

To make our point clearer, we have summarized the effect of mutations of *BRCA1*, *MBD9*, and *SAC3B* on molecular phenotypes and virus resistance in the *atx5 atx6* background. From the chart above, we could see that only activation of DDR is linked to the virus resistance phenotype despite of various molecular features in *atx5 atx6*.

The detailed explanation is as below: the flow cytometry results in Fig. 1c showed that the peak width of *atx5 atx6* at 8 and 16C was broader than that in Col-0, which indicates there was extra DNA content in the 8 and 16C cells of *atx5 atx6*. The amplified extra DNA in 8 and 16C cells of *atx5 atx6* mainly belongs to the heterochromatic regions (Jacob et al., 2010). We could observe a decreased peak width in 8 and 16 C cells of *mbd9 atx5 atx6* compared to those of *atx5 atx6* (Fig. 1c), but slightly increased peak width in the 8 and 16 C cells of *brca1 atx5 atx6* compared to *atx5 atx6* (Fig. 1c), indicative of opposite levels of heterochromatic amplification in three triple mutants. This result is consistent with previous data (Hale et al., 2016). A similar pattern was observed for the expression of TEs shown by the RNA-seq in Fig. 1d.

So, if the amplication and heterochromatic region and TE activation would cause the viral resistance of *atx5 atx6*, we would expect a lower virus-infected ratio and virus titer in *brca1 atx5 atx6* compared to that in *atx5 atx6*. Meanwhile, we would expect higher virus-infected ratios and virus titers in *mbd9 atx5 atx6* and *sac3b atx5 atx6* compared to that in *atx5 atx6*. Instead, the virus-infected ratios and virus titers in all the triple mutants were all higher than that in *atx5 atx6*. Thus, the heterochromatic amplification and TE reactivation were not associated with the viral resistance of *atx5 atx6*, we could

uncouple them with viral resistance phenotypes of *atxr5 atxr6* and then move forward to pinpoint the molecular phenotype that is associated with viral resistance of *atxr5 atxr6*.

Jacob, Y., Stroud, H., Leblanc, C., Feng, S., Zhuo, L., Caro, E., Hassel, C., Gutierrez, C., Michaels, S.D., and Jacobsen, S.E. (2010). Regulation of heterochromatic DNA replication by histone H3 lysine 27 methyltransferases. Nature 466, 987-U117.

*Hale, C.J., Potok, M. E., Lopez, J., Do, T., Liu, A., Gallego-Bartolome, J., Michaels, S. D., and Jacobsen, S. E. (2016). Identification of Multiple Proteins Coupling Transcriptional Gene Silencing to Genome Stability in *Arabidopsis thaliana*. Plos Genetics 12.*

10. Why *sog1* was chosen for the analysis in Fig2? It has not been introduced.

Response: Thanks for the comment.

The suppressor of gamma response 1 (SOG1) governs the transcriptional activation of DDR factors. The latest chromatin immunoprecipitation sequencing (ChIP-seq) reveals that SOG1 directly binds to the loci of 11 co-expressed genes including DNA repair, cell cycle, and transcriptional factors to regulate their transcription (Yoshiyama et al., 2009, PNAS; Bourbousse et al., 2018, PNAS). We aimed to test whether and/or which upstream DNA repair factors are required for efficient viral DNA replication. We selected mutants related to ATM, ATR, and SOG1 that are in hands to see how mutations of these three genes would affect viral DNA replication. We have introduced SOG1 in the line 232-233 (new ms).

Yoshiyama, K., Conklin, P. A., Huefner, N. D., and Britt, A. B. (2009). Suppressor of gamma response 1 (SOG1) encodes a putative transcription factor governing multiple responses to DNA damage. Proc Natl Acad Sci U S A 106, 12843-12848.

*Bourbousse, C., Vegesna, N., and Law, J. A. (2018). SOG1 activator and MYB3R repressors regulate a complex DNA damage network in *Arabidopsis*. Proc Natl Acad Sci U S A 115, E12453-E12462.*

11. Why the percentage of Col-0 symptomatic plants at 11 dpi differs between Fig 1 and Fig 2? (80 vs 30%)

Response: Thanks for the comment. Because virus inoculation is affected by many factors including agrobacterial growth stages, light, temperature, humidity, and variation of soil lots. It is normal to see variations in virus studies. To avoid these issues, we only compared the samples inoculated at the same time.

12. Page 9, lane 245, CHR31 or TK1A were not introduced

Response: Thanks for the comment.

We found mutations of upstream HRR factors such as ATM, ATR, and downstream factors such RAD51 and RPA1A compromised the viral DNA amplification. RAD51 and RPA1A are *bona fide* Rep partners, and we are curious about whether mutations of

downstream HRR factors that do not interact with Rep will affect the viral amplification or not. Since we could not rule out the possibility that CHR31 and TK1A would not interact with Rep at this stage, we replaced the results of CHR31, and TK1A with those of HOP2, BRCA1, and CYCB1 that did not interact with Rep (in the Y2H and Co-IP, with RAD51 as positive control); and provided the related introduction in line 82-85 and line 223-224 (new ms) instead.

13. Page 9, lane 248. "Given that *atx5 atx6* and *atx5 atx6 rad51* had the same viral resistance phenotype, we concluded that RAD51 and RPA1A were downstream effectors that accounted for reduced viral replication in *atx5 atx6*." However, RAD51 and RPA1A are overexpressed in the *atx5 atx6* double mutant (Fig 3d). Consequently, they should promote viral replication and not reduce it. Please clarify.

Response: Thanks for the comment.

Because RAD51 and RPA1A are key components for DNA repair and have higher ChIP signal over rDNA and defense-related PCGs in *atx5 atx6* (New Fig. 5d). The immunofluorescence signal intensity of γ -H2AX that represents the level of DNA double-strand break (or DNA damage) in *atx5 atx6* is higher than that in Col-0 (Fig. 2 Feng et al., 2017). This fact indicates that the increased expression of RAD51 and RPA1A in *atx5 atx6* is still not sufficient to repair the endogenous DNA damage in *atx5 atx6*. Therefore, RAD51 and RPA1A will be recruited to the genome of *atx5 atx6* to repair the host DNA damage and cannot be efficiently recruited onto the viral genome to promote the viral DNA amplification.

Feng, W., Hale, C. J., Over, R. S., Cokus, S. J., Jacobsen, S. E., and Michaels, S. D. (2017). Large-scale heterochromatin remodeling linked to overreplication-associated DNA damage. *Proc Natl Acad Sci U S A* 114, 406-411.

14. What does inf mean in Extended data Fig 8?

Response: Inf in the Extended data Fig 8 represents the virus-infected. We will annotate it in the figure legend as "Moc represents the mock-treated sample and inf represents the virus-infected samples". To be accurate. We have changed virus-infected to virus-inoculated (ino) samples in the revision and annotated them in the related ChIP-seq analysis.

15. Page 10, lane 75. Typo figure 4C

Response: Thanks, Type corrected.

16. Page 11, lane 301. "Collectively, these results strongly support a model where unstable DNA in *atxr5 atxr6*, especially rDNA loci, requires RAD51 and RPA1A, among other HRR factors, to maintain genome integrity." Unjustified conclusion since genome integrity was not investigated.

Response: Thanks for the comment.

The DNA damage level in *atxr5 atxr6* and Col-0 has been examined by the immunofluorescence assay (immunofluorescence signal intensity of γ -H2AX represents the level of DNA double-strand break) and showed that the DNA damage level in *atxr5 atxr6* is much higher than that in Col-0 (Fig. 2, Feng et al., 2017, PNAS, please refer the answer to question 13). The corresponding author of that publication is also a co-author in this manuscript.

Feng, W., Hale, C. J., Over, R. S., Cokus, S. J., Jacobsen, S. E., and Michaels, S. D. (2017). Large-scale heterochromatin remodeling linked to overreplication-associated DNA damage. Proc Natl Acad Sci U S A 114, 406-411.

17. Page 12, lane 344. Awkward sentence: "This result clearly resulted from that the fact that."

Response: Thanks, corrected.

18. It is not clear how the authors calculated RAD51 and RPA1A loading on the viral genome (Fig 5f).

Response: Thanks for the comment.

We first calculated the ratio of the RAD51 or RPA1-ChIP-seq reads mapped to the viral genome to the total mapped reads (including reads mapped to virus and host) in the RAD51 or RPA1A ChIP-seq. Then calculated the ratio of the reads mapped to the viral genome to the total mapped reads in the DNA-sequencing from input samples. The IP efficiency is calculated as below:

$$\text{IP efficiency} = \frac{\text{ratio of reads mapping to virus from RAD51 ChIP Seq}}{\text{ratio of reads mapping to virus from Input}}$$

19. Extended data Fig 8. Why did the author use different constructs for the overexpression lines? FLAG-MYC FLAG-MYC-RAD51 in Col-0 or Myc-RAD51 in *atxr5 atxr6*.

Response: Thanks for the comment.

Different vectors were selected because different plants or mutant backgrounds have different antibiotics resistance such as Basta and kanamycin. For instance, we could use FLAG-MYC vector, which could carry either Kan or Basta to screen transgenic plants in Col-0 background. Since the *atxr5 atxr6* is resistant to both Basta and kanamycin so it is not compatible with the Flag-Myc vector, we needed to use another vector that is resistant to hygromycin to express Myc-RAD51 in *atxr5 atxr6* (result was presented in New Extended data Fig. 13c).

Reviewer #2 (Remarks to the Author):

In this manuscript, Wang et al., discovered increased viral resistance in *atxr5 atxr6* compared with Col-0. They found that Rep interacts with RAD51 and RPA1A, and might hijack them for virus replication. Because in *atxr5 atxr6* mutant RAD51 and RPA1A are enriched at rDNA loci, this together with the lesser binding of RAD51 at viral DNA makes the authors propose that in *atxr5 atxr6*, unstable host DNA and defense-related genes retain a large amount of DNA repairing machinery to prevent its routing to viral genome, leading to low efficient replication of the virus. However, the data provided cannot support the conclusions regarding the molecular mechanisms, and also I have several concerns regarding the biological significance of the work and data analyses.

Major concerns

1. Line 40-42. Current results in this work only suggest that in **a special case** (*atxr5 atxr6* mutant), impaired genome stability might suppress viral infection. To demonstrate the claim that *host genome could retain DNA repairing proteins via sacrificing its genome stability to suppress viral infection*, **relevant data need to be provided in virus-infected wild type plants**, otherwise the biological significance of this work is missing.

Response: Thanks for the insightful comment.

In our manuscript, we report the resistance to Geminivirus infection in the *atxr5/6* mutant plants and we provide data that shows the requirement of RAD51 and RPA1A recruitment to the geminiviral genome via their interaction with the viral protein, Rep. We propose the resistance to Geminivirus infection in *atxr5/6* is due to the genome retention of RAD51 and RPA1A which outcompetes Rep and their recruitment to the viral genome. Our model predicts that genomic recruitment of RAD51 and RPA1A, such as in the case of DNA damage, would result in resistance to infection; furthermore, higher availability of RAD51 and RPA1A would increase susceptibility to Geminivirus infection. The latter was directly tested in our manuscript. Consistent with our model, plants overexpressing RAD51 are indeed, more susceptible to Geminivirus infection (New Fig. 5 f-h)

We understand the importance of testing our model in WT plants. We have now pursued three strategies to address this important criticism:

First, we learned from recent surveys that there is an inverse correlation between UV intensity and the prevalence of geminivirus species in China among other counties (Li, et al., 2022, Sci. China Life Sci.; Liu et al., 2017, Renew. Sust. Energ. Rev). These surveys

suggest that UV is one important negative factor that impacts the prevalence of geminiviruses in natural conditions. Triggered by these results, we tried to treat the plants with ultraviolet light; aiming at priming the plants to induce DDR and reduce their resistance to Geminivirus. However, ultraviolet (UV) radiation targets nucleic acids, proteins, plant growth regulators, and photosynthesis pigments (Hollosy, 2002, Micron). In our hands, UV treatment at 50 mJ or 100 mJ affected the growth and development of *Arabidopsis* to a great extent, causing the death of irradiated tissue. We realized that treatment dosages are key to this experiment but difficult to be optimized to complete the experiments in the allowed time.

The effects of UV on *Arabidopsis*, pictures are captured 17 days post-treatment, scale bars, 1cm

Hollosy, F. Effects of ultraviolet radiation on plant cells. *Micron* 33, 179-97 (2002).

Li, F., Qiao, R., Wang, Z., Yang, X., & Zhou, X. (2022). Occurrence and distribution of geminiviruses in China. *Science China. Life sciences*, 65(8), 1498–1503.

Liu, H., Hu, B., Zhang, L., Zhao, X. J., Shang, K. Z., Wang, Y. S., and Wang, J. Ultraviolet radiation over China: Spatial distribution and trends. *Renewable and Sustainable Energy Reviews* 76, 1371-1383 (2017).

Second, we appealed to a different approach: the plant immune hormone, salicylic acid (SA), can induce DNA damage in the absence of a genotoxic agent (Yan et al., 2013). Since SA triggers DNA damage and also induces the expression of defense genes, we consider this treatment as a natural condition that mimics the physiological condition in *atx5/6* that contributes to its resistance to geminivirus infection. We treated Col-0 plants with either 0, 0.1, or 0.5 mM SA for 4h before inoculation. We confirmed the induction of DDR by assessing the expression of BRCA1 and RAD17 with qRT-PCR. Consistent with our prediction, SA-treated wild-type plants showed increased resistance to geminiviral infection related to mock-treated plants (new Fig. 6i, j). Notably, multiple reports have also shown the repression of SA signaling upon Geminivirus infection and the SA-induced resistance to infection (Chen et al., 2010, Plant J). That being said, our new result and prior publications further support our model in WT condition.

New Fig. 6 i, j SA treatment activates the DDR response and suppressed the viral DNA amplification

Chen, H., Zhang, Z., Teng, K., Lai, J., Zhang, Y., Huang, Y., Li, Y., Liang, L., Wang, Y., Chu, C., Guo, H., and Xie, Q. Up-regulation of LSB1/GDU3 affects geminivirus infection by activating the salicylic acid pathway. *Plant J* 62, 12-23 (2010).

Last, we took advantage of transgenic lines expressing H3.1S28A that have been newly generated by Dr. Jacob's laboratory. The H3.1S28A transgenic plants displayed heterochromatic DNA amplification (Fig. 4a, Davarinejad et al., 2022, Science) and increased rDNA (in new Fig. 6e) compared to Col-0. Importantly, the transgenic plants showed robust virus resistance against CaLCuV vs WT. Thus, the new result (in new Fig. 6e-h) provided another new piece of evidence, in addition to the case of *atxr5/6* mutants, to support our model that unstable genomic elements will retain HRR factors and prevent them from being recruited onto the viral genome to suppress viral DNA amplification. Please refer to our reply to the main criticism by Reviewer #1.

Of note, the results that plant possessing unstable segments in certain loci in genomes (like *atxr5/6* mutants and the H3.1S28A line) display robust resistance to CaLCuV infection without major physiological and developmental defects could be potentially useful to improve agricultural traits.

Despite these promising results, we could not extensively try many other kinds of DNA damage methods to see if they have a similar boosting effect on viral resistance as the work is clearly beyond the scope of the current manuscript. We plan to lower our tone of the biological significance in the evolutionary perspective in this manuscript.

2. Line 88-90. Results in reference 15 (Davarinejad et al., 2022, Science) show that loss of RAD51 in *atxr5 atxr6* does not enhance heterochromatin amplification.

Response: Thanks for the comment. Robust CV of flow cytometry analysis is the standard method to analyze heterochromatic amplification in *atxr5 atxr6*. Figure 3g of Davarinejad et al., 2022, Science shows that the robust CV of *rad51 atxr5 atxr6* is broader than that in *atxr5 atxr6*, and the significant difference between *atxr5 atxr6* and *rad51 atxr5 atxr6* was supported by the statistical analysis (unpaired two-tailed student t-test, *, **,*** and ****, *P* value <0.05, 0.01, 0.001 and 0.0001, respectively) with original data (our collaborator, the corresponding author of cited paper, Dr. Jacob kindly provided the original data).

Davarinejad, H., Huang, Y. C., Mermaz, B., LeBlanc, C., Poulet, A., Thomson, G., Joly, V., Munoz, M., Arvanitis-Vigneault, A., Valsakumar, D., Villarino, G., Ross, A., Rotstein, B. H., Alarcon, E. I., Brunzelle, J. S., Voigt, P., Dong, J., Couture, J. F., and Jacob, Y. The histone H3.1 variant regulates TONSOKU-mediated DNA repair during replication. *Science* 375, 1281-1286 (2022).

3. It would be necessary to examine the DNA re-replication and TE-reactivation phenotypes in *atm;atxr5;atxr6* mutant to further decipher the connection between *atxr5;atxr6*-induced phenotypes, DDR and viral resistance.

Response: Thanks for the comment. In Fig 4B and C and D Feng et al., 2017, PNAS, it has been shown that *atxr5 atxr6 atm* still maintains the DNA re-replication and increased level of DNA damage (γ -H2AX signal is an indicator of DNA double-strand break, EdU is an indicator of DNA replication).

Genotype	DNA re-replication	TEs reactivation	Activation of DDR	Virus resistance
Col-0	No	No	No	No
atxr5 atxr6	Yes	Yes	Yes	Yes
brca1 atxr5 atxr6	Yes	Yes	No	No
mbd9 atxr5 atxr6	No	No	No	No
sac3b atxr5 atxr6	No	No	No	No
atm atxr5 atxr6	Yes	?	?	Yes

As summarized in the table above, results from *mbd9 atxr5 atxr6*, *sac3b atxr5 atxr6*, and *brca1 atxr5 atxr6* are sufficient to decouple the heterochromatic replication and expression of TEs from the viral resistance (New Fig. 1c-g). Since we have already used three triple mutants (*mbd9 atxr5 atxr6*, *sac3b atxr5 atxr6*, and *brca1 atxr5 atxr6*) to decouple the heterochromatic replication and expression of TEs from the viral resistance, and since *atm atxr5 atxr6* was used to only bridge the genetic pathway of DNA damage repair to the virus replication, and since *atm atxr5 atxr6* has been clearly

shown to possess the DNA replication and DNA damage features, we feel there is no need to further invest effort on deciphering the connection among *atxr5;atxr6*-induced phenotypes, DDR and viral resistance.)

4. The figure legends of Extended Data Fig. 4c, d does not explain what exactly is the difference between 4c and 4d. In addition, what is the “mock” condition in viral infection? Can an experiment from previous work (GSE77735) be considered “mock”?

Response: Thanks for the careful reading of ms.

Genes from Extended Data Fig. 4d in the original text (new Extended Data Fig. 4f) were selected from Extended Data Fig. 4c in the original text (new Extended Data Fig. 4e). We rewrote the figure legend in the following sentence: “e, Heat map showing the overall change of transcript level for 1625 DEGs in *atxr5 atxr6*, *mbd9 atxr5 atxr6*, and *sac3b atxr5 atxr6*. RNA-seq data were mined from GSE77735. The quantification was done with de-seq2. f, Heat map showing the transcriptional level change for 240 genes selected from 1625 DEGs in indicated backgrounds.”

“Mock” is data from a set of plants treated exactly in the same way as the infected ones but lacking agrobacteria to inoculate with the virus. Mock plants were sprayed with a solution containing 10mM MgCl₂, 150μM ASG, and silicon carbide powder onto the rosette center. As with infected plants, the residues were washed away 16 hours later and plants were harvested at around 14 days post-treatment.

This is not the case for the dataset GSE77735, which corresponds to work performed by our group (Ma et al., 2018, Development cell). To avoid confusion, we changed the manuscript. We named our new ChIP-seq data as “Mock”, and the data from GSE77735 is referred to as “Untreated”.

Since all other growth conditions (temperature, light intensity, humidity, etc.,) of plants in previous publication (Ma et al., 2018, Development cell) were similar to what we used for this work, and DNA re-replication, higher level expression of TE and DNA repair genes were also observed in *atxr5 atxr6* (Fig 5 F, G and H of Ma et al., 2018, Development cell), we think that the biology is comparable and that it is reasonable to use the data from GSE 77735 to further explain our Mock ChIP-seq data.

Ma, Z., Castillo-González, C., Wang, Z., Sun D., Hu, X., Shen, X., Potok, M. E., and Zhang, X. Arabidopsis Serrate coordinates histone methyltransferases ATXR5/6 and RNA processing factor RDR6 to regulate transposon expression. Dev Cell 45, 769-784.e6 (2018)

5. Line 231-232. Are these 20 selected DDR factors differentially expressed in RNA-seq analysis? It is not clear what is the purpose of Extended Data Fig. 7a, in which transcripts of only six genes are shown. Moreover, PARP2 and CHR31 were not tested in Y2H assay, and PARP2 and TK1A are repeatedly shown in Figure 2c and Extended Data Fig. 7a.

Response: Thanks for the comment.

Yes, we selected 20 DDR factors because they were differentially expressed in RNA-seq analysis upon virus inoculation. We just selected a few examples to show the representative IGV files and could not show all of them due to space limitations.

Most constructs we used for the Y2H assay were purchased from Arabidopsis Biological Resource Center. The constructs with PARP2 and CHR31 coding sequences are not available so we did not perform Y2H with those two genes.

Since IGV files just serve as examples for genes involved in the homologous recombination repair pathway, three representative genes were good enough. We used HOP2, BRCA1, and CYCB1 to replace the three genes in the original Fig. 2C and deleted other IGV files in the original Extended Fig. 7a.

6. Extended Data Fig. 7b. The interaction between REP and RAD51 was tested twice in yeast (4th row and 17th row), but different results were generated. This raises concerns regarding the quality of the data.

Response: Thanks for the careful proofreading and comment.

AD-RAD51 and BD-Rep, together with AD-Rep and BD-RAD51, serve as positive controls in many of our experiments, and the combinations were typically plated on the 4th row of the plate. This arrangement was the same as in the upper two plates. For this panel, we mistakenly labeled the rows of the positive controls with Rep/EMB1379. We have thoroughly re-checked our notebook for the 16 colony screening results and we are confident that NSE1 (EMB1379) does not interact with Rep.

We apologize for mislabeling the figure and we have now corrected it in the new Extended data Fig. 7d.

7. Figure 3f. There are two dark grey columns in Figure 3f.

Response: Thanks for the comment. We have corrected the figure listed in the new Fig. 3f.

Fig. 3f HRR factors are required to promote the virus amplification.

8. Line 244-245 and Extended Data Fig. 7d. It seems that chr31 and tk1a mutants are less susceptible to viral infection, please provide statistics to support the conclusion that their mutations did not affect viral pathogenesis.

Response: Thanks for the comment. We showed that upstream factors of DNA repair such as ATM and ATR are required for efficient viral DNA amplification, then we investigated whether the downstream DDR factors affect viral DNA amplification. We found that Rep-interacting proteins, RAD51 and RPA1A, are required for efficient viral DNA amplification.

We then asked whether or not mutations of other downstream DNA repair proteins that do not interact with viral protein Rep might also affect virus amplification. To this end, we tested the effect of mutations of several other downstream DNA repair factors (including CHR31, TK1A, HOP2, CYCB1, and BRCA1) on the virus amplification. BRCA1, HOP2, and CYCB1 do not show interaction with Rep but we are not sure of CHR31 and TK1A. Thus, we replaced this panel for the results obtained with *brca1*, *hop2*, and *cycb1* mutants.

The results were listed in the new Extended data Fig.7 e, f.

New Extended Data Fig. 7e, f Mutations of HOP2, BRCA1, and CYCB1 do not affect viral performance

9. Line 256-257. It is not clear how this conclusion is reached. Results in Figure 3 only show that RAD51 and RPA1A associate with Rep and viral genome and are required for viral infection, but there is no evidence to support their participation in virus replication.

Response: Thanks for the comment. Replication-associated protein (Rep, or C1 and AL1) is a highly conserved protein across the family *Geminiviridae*, and essential for viral DNA replication. It has been reported that Rep protein nicks virus DNA in a site-specific manner, and catalyzes the DNA unwinding to initiate the rolling circle replication (RCR). That being said, we did not pursue further effort in this direction to show that RAD51 and RPA1A directly promote the rolling circle replication or recombination-dependent replication, as the work will be clearly beyond the scope of the ms. However, we cited the paper and also discussed several possibilities in the discussion part. We also changed the virus replication to virus amplification in the ms.

10. Line 260-262 and Extended Data Fig. 8a-c. It is very confusing regarding the antibody test here. Based on line 724-725, ChIP experiments were performed with anti-RAD51 and anti-RPA1A antibodies. But figure legends of Extended Data Fig. 8a-c do not provide any information in terms of how the antibody tests were performed and seem that Myc and HA but not RAD51 and RPA1A antibodies were used for blotting. 1) Extended Data Fig. 8a, what's the purpose of examining FLAG-Myc-RAD51 and Myc-RAD51 with Myc antibody here? 2) Extended Data Fig. 8b, these blots do not show the quality/specificity of the antibody. Why not simply use RAD51/RPA1A antibody for blotting with Col and *rad51/rpa1a* mutant? Since ChIP was performed with Col plants but not with these Myc-tagged lines. 3) Extended Data Fig. 8c, what's the purpose of detecting HA IPed HA-RPA1A with HA antibody? Overall, these antibody testing results raise concerns regarding the quality of RAD51 and RPA1A antibodies used in this study.

Response: Thanks for the comment. We should have written the experiments more clearly.

These sets of experiments were performed many years ago to validate the transgenic lines or antibodies when a graduate student just started the work.

1) The purpose of Extended Fig 8a in the original text was to examine transgenic plants expressing Flag-Myc-RAD51 and Myc-RAD51 with Myc antibody. The plants were used to perform virus inoculation assay in the new Fig. 5f-h.

2) The initial purpose of Extended Fig 8b in the original text was to test the quality and specificity of anti-RAD51 polyclonal antibodies with transgenic plants; and the experiments were performed many years ago. Briefly, we immunoprecipitated Myc-RAD51 protein from the transgenic lines with an anti-RAD51 antibody and probed the western blot with anti-Myc. The clear enrichment of Myc-RAD51 in *atxr5 atxr6:35S-MYC-RAD51* but not in *atxr5 atxr6* demonstrated the viability of the antibody. Be noted that the expected band MYC-RAD51 was indicated with an arrow and the heavy chain of RAD51 antibody is around 50 kd.

New Extended Data Fig. 8e Western blot assays via an anti-Myc antibody detected the immunoprecipitated Myc-RAD51 with an anti-RAD51 antibody.

Following this reviewer's advice, we re-did the experiment to validate the antibodies. We isolated nuclei and performed western blot assays with an anti-RAD51 antibody and found one weak specific band at 37kd (molecular weight of RAD51 is 37kd) when *atxr5 atxr6* is treated with UV. Furthermore, we conjugated the anti-RAD51 antibody with protein A-magnetic beads and then performed the immunoprecipitation. This time we detected the enriched RAD51 in *atxr5 atxr6* but not *atxr5 atxr6 rad51* without the contamination of heavy chains. The results are listed in the New Extended Fig. 8a-b. Collectively, we are confident in the antibody quality for immunoprecipitation or ChIP.

New Extended Data Fig. 8a, b Western blot assays validated the specificity of an anti-RAD51 antibody. RAD51 was immunoprecipitated from nuclear extraction with a protein A-magnetic bead-conjugated RAD51 antibody, and then probed with the anti-RAD51 antibody.

Extended Data Fig. 8c, since the expression of RPA1A is also very low (Fig. 3d), so we tested RPA1A antibody with tobacco carrying transiently expressed HA-RPA1A and tested efficiency by the HA antibody (FM-RAD51 serves as a negative control). The IP is performed with HA (serve as one positive control) and RPA1A antibody (which has been cropped out). Western blot is performed with anti-HA.

New Extended Data Fig. 8f Western blot assays via an anti-HA antibody validated the immunoprecipitated HA-RPA1A with an anti-RPA1A antibody.

Again, per the advice from the reviewer, we re-did the experiment: We immunoprecipitated RPA1A with its antibody (conjugated to protein A-magnetic beads) and detected the enriched RPA1A in infected Col-0 but not *rpa1a*. The results are listed in the New Extended Fig. 8c, d.

Collectively, we are confident about the antibody quality for immunoprecipitation or CHIP.

New Extended Data Fig. 8c, d. Western blot assays validated the specificity of an anti-RPA1A antibody. RPA1A was immunoprecipitated from nuclear extraction with a protein A-magnetic bead-conjugated RPA1A antibody, and then probed with the anti-RPA1A antibody.

11. Line 264-266 and Extended Data Fig. 9a, b. Increased number of peaks in heterochromatin for what? RAD51 or RPA1A? But in Extended Data Fig. 9a, b RAD51 is clearly enriched at euchromatin, and RPA1A data are not shown. Also, it is not clear whether these data are derived from Col mock or any other conditions.

Response: Thanks for the comment.

In line 264-266 (In the original text) we mainly observed numerous peaks (circled by red rectangles below) at the heterochromatic regions (the heterochromatic regions in Arabidopsis are marked by the H3K27me1 and H3K9me2) when we used the method from our reference (Jacob et al., 2010, Nature). However, these peaks were not observed when only uniquely mapping reads were aligned (new Extended Data Fig. 9c, d).

We have added the data for RPA1A in the new Extended Data Fig. 9c,d.

We aimed to check the how distribution of RAD51 and RPA1A enriched peaks overlapped with heterochromatin markers (H3K9me2, H3K27me1) and euchromatin markers (H3K27ac and H3K4me3). All the data related to H3k9me2, H3K27me1, H3K27ac, and H3K4me3 from references 10, 13, and 14 were derived from untreated Col-0. Thus, the data we used for RAD51 and RPA1A were derived from Col-0 Mock.

New Extended data Fig. 9 c (the analysis was performed with total reads including multiple mapping reads), d (the analysis was performed with unique mapping reads) Chart diagrams show peak distributions of RPA1A, RAD51, H3K27me1, H3K9me2, H3K4me3, and H3K27Ac over genome of Col-0.

Jacob, Y., Stroud, H., Leblanc, C., Feng, S., Zhuo, L., Caro, E., Hassel, C., Gutierrez, C., Michaels, S.D., and Jacobsen, S.E. Regulation of heterochromatic DNA replication by histone H3 lysine 27 methyltransferases. *Nature* 466, 987-U117 (2010).

12. Line 273-278, Fig. 4c and Extended Data Fig. 9c. Data here does not tell whether RAD51 and RPA1A are enriched at any specific elements or heterochromatin or euchromatin as the percentage of different elements on the Arabidopsis/mouse genome is not provided and compared. Based on the text, RAD51 and RPA1A are localized at both heterochromatin and euchromatin, how does this general protein distribution pattern suggest conservative functions in eukaryotes?

Response: Thanks for the comment.

We agree with the reviewer that this pattern is not enough to support the statement that “supportive of their conserved function in eukaryotes”, we will change it to “indicative of their important functions in eukaryotes”.

13. Fig. 4e, f. Please describe what is “overall”, “rDNA” and “PCG”. Are they all elements, all rDNA loci and all PCG in the genome or RAD51/RPA1A-enriched “overall”, “rDNA” and “PCG” in Col/atxr5 atxr6?

Response: Thanks for the comment.

Overall represents the total 16865 peaks of RAD51 (Union of RAD51 enriched peaks from different genotypes, treatment, and replicates) or the total of 1675 of RPA1A enriched

peaks (Union of RPA1A enriched peaks from different genotypes, treatment, and replicates).

rDNA represents the two enriched -peaks that belong to 45S rDNA.

PCG represents the 9917 of RAD51- or 514 of RPA1A enriched peaks that belong to the protein-coding gene (PCG). We have added the annotation in the related profile analysis.

14. Extended Data Fig. 10a, b. Again, please define “ncRNA” and “TE”. I assume the left half panels show RAD51 and RPA1A profiles and the right half panels show H3K27me1 profiles. How could the patterns of RAD51 and RPA1A at ncRNA/TE, and H3K27me1 at ncRNA be almost the same except for different scales?

Response: Thanks for the comment.

We have included the descriptions in the figure legend as follows "ncRNA represents non-coding RNA loci. TE represents transposable elements".

Regarding the distribution patterns, we were just as stricken as the reviewer by the data. Because of that, we carefully assessed our data and in fact, we found subtle differences. Specifically, we refer the reviewer to the patterns in the red rectangles (in the figures below), which are different between RAD51 and RPA1A ChIP-signal.

However, the most interesting question is the reason for the striking similarity. As shown in Fig 4 a and b, RAD51 and RPA1A interact with each other and RPA1A enriched loci overlapped with those of RAD51, suggesting that RAD51 and RPA1A collaboratively to repair the DNA damage at those loci. We zoomed into the details and found that 391 TEs were commonly enriched in RPA1A and RAD51. Out of 409 RPA1A-enriched TEs and 1861 RAD51, the reads from overlapped TEs account for 98.5% of RPA1A-enriched TEs (In Col-0 Mock RPA1A ChIP-seq, the number of raw reads s corresponding to the overlapped 391 TEs was 86639 and the number of raw reads corresponding to 409 TEs was 87971) and 51.0 of RAD51-enriched TEs (In Col-0 Mock RAD51 ChIP-seq, the number of raw reads corresponding to the overlapped 391 TEs was 122393 and the number of total raw reads corresponding to 1861 TEs was 242799).

We observed a similar pattern in ncRNA reads: there were 53 overlapping ncRNAs, out of 53 RPA1A-enriched (In Col-0 Mock RPA1A ChIP-seq, Col-0 Mock RPA1A, the number of raw reads corresponding to the overlapped 53 ncRNA was 198247 and the number of raw reads corresponding to 53 ncRNA was 198247) and 630 RAD51-enriched ncRNAs (In Col-0 Mock RAD51 ChIP-seq, the number of raw reads corresponding to the overlapped 53 ncRNA was 213861 and the number of raw reads corresponding to 630 ncRNA was 249388). The reads corresponding to 53 overlapped ncRNAs accounted for 100% and 85.7% RPA1A- and RAD51-enriched ncRNAs, respectively.

This pattern is not observed in PCGs (In Col-0 Mock RAD51 ChIP-seq, the number of raw reads corresponding to the overlapped 514 PCGs is 162679 whereas the reads corresponding to 9917 PCGs is 2223142, which is only 7.3% of reads mapped to 9917 PCGs). Thus, PCGs could serve as an internal reference here.

Original text Extended data Fig. 10 a, b Distribution of normalized RAD51 and RPA1A ChIP-signal (RPKM) from Mock-treated Col-0 and *atxr5 atxr6* over different categories.

And this has also been observed in the New Extended Data Fig. 10b, c.

15. Line 290-295. Based on the text, I assume that RAD51 and RPA1A were significantly enriched on two 45S rDNA loci, which are localized at chromosomes two and four (Benoit et al., 2013, Gene), but Fig. 4g shows a locus at chromosome 3.

Response: Thanks for the comment.

As is shown below, Zapata et al., 2016 updated that the rDNA can be found at chr2, chr3, chr4, and chr5 of one ecotype of *Arabidopsis-Ler* (Fig 1, Zapata et al., 2016, PNAS). RAD51 and RPA1A were significantly enriched on two rDNA loci, we used the sequence from Rbanal et al., 2017 to perform a sequence blast and find that the DNA sequence of RAD51 and RPA1A enriched peaks over rDNA loci corresponding to 45S rRNA located in the chr2 and chr3 of Col-0. The location of 45S rDNA differs among different ecotypes.

Zapata, L., Ding, J., Willing, E. M., Hartwig, B., Bezdán, D., Jiao, W. B., Patel, V., Velikkakam James, G., Koornneef, M., Ossowski, S., and Schneeberger, K. Chromosome-level assembly of *Arabidopsis thaliana* Ler reveals the extent of translocation and inversion polymorphisms. *Proc Natl Acad Sci U S A* 113, E4052-60 (2016).

Rabanal, F.A., Nizhynska, V., Mandakova, T., Novikova, P. Y., Lysak, M. A., Mott, R., and Nordborg, M. Unstable Inheritance of 45S rRNA Genes in *Arabidopsis thaliana*. *G3 (Bethesda)* 7, 1201-1209 (2017).

We annotated the 45S rDNA and placed the 45S rDNA on chr2 at new Fig. 4f and the other one on chr3 at new Extended Fig. 10a

New Fig. 4f and Extended data Fig. 10a IGV files of normalized ChIP signals of H3K9me2, H3K27me1 (RPKM), RAD51, RPA1A, H3K27ac, and H3K4me3 on a 45S rDNA locus on chr2 (Fig. 4f) and chr3 (Extended data Fig. 10a).

16. Overall, all the RAD51 and RPA1A ChIP-seq analyses need to be performed in a more proper way. For example, although the rDNA loci might have more RAD51 and RPA1A enriched in *atxr5 atxr6*, they also have more DNA copies in *atxr5 atxr6*. This is clear in Fig. 4g (and also in Extended Data Fig. 10c, d) that except for H3K27me1, levels of all others including H3K9me2 and Input were increased in *atxr5 atxr6*. Therefore, the RAD51 and RPA1A ChIP-seq signals should be normalized to Input but not using RPKM to eliminate the influence of higher copy number background in *atxr5 atxr6*, so to test whether RAD51 and RPA1A are really more enriched at these loci in *atxr5 atxr6* compared with Col.

Response: Thanks for sharing this idea.

We understand the reviewer's perspective of the traditional ChIP-seq analysis. However, the analysis (ChIP-seq reads/input reads) is not suitable to address the questions posed by our research. Specifically, we are not questioning the affinity of protein binding to the nucleic acid and the relative binding difference of RAD51 or RPA1A within the host genome. What we need to detect is the total amount of protein retained in the genome. This is our question because we have more DNA in the double mutant, not a change in the biochemical properties of the proteins, nor the relative changes in binding loci within the host genome. Thus, normalizing the ChIP signal to the amount of DNA would ignore and mask the biology of our subject, the *atxr5/6* double mutant.

If we would normalize the ChIP signal to the input (the figure is shown below), we would see that this method would produce a result that is inconsistent with previous research and other experiments. Specifically, this analysis shows an even marginally reduced RAD51 signal in *atxr5 atxr6* than that in Col-0. This is inconsistent with the demonstrated increased expression of RAD51 in *atxr5/6* (new Fig. 3d) and the increased DNA damage and RAD51 foci in the double mutant (Fig 2, Feng et al., 2017). Thus, we further confirmed that our pipeline and believed that our ChIP-seq method, not the canonical analysis, is appropriate in this case.

RAD51 distribution over 16865 enriched loci in Col-0 and *atxr5 atxr6* revealed via a classic ChIP-seq.

New Fig. 3d Elevated expression of RAD51 in *atxr5 atxr6*

We have mined the data used for peaking calling (H3K27ac from GSE14612610 and H3K4me3 from GSE16689714) and found that the signal of H3K27ac and H3K4me3 remain unchanged in *atxr5 atxr6* over rDNA (new Fig. 4f and Extended Fig. 10a, please refer the picture shown for the same question and question 15) compared to that in Col-0.

We have searched the literature and found that the H3K9me2 is required to maintain the heterochromatin amplification phenotype in *atxr5 atxr6*. KYP (SUVH4), SUVH5, and SUVH6 redundantly catalyze the deposition of H3K9me2 over *Arabidopsis* (Liu et al., 2010, Annu. Rev. Plant Biol.). Deletions of SUVH4/5/6 will suppress the heterochromatin amplification in *atxr5 atxr6* (Fig. 2F, Stroud et al., 2012, PLoS Genet.).

Liu, C., Lu, F., Cui, X., & Cao, X. (2010). Histone methylation in higher plants. Annual review of plant biology, 61, 395–420.

Stroud, H., Hale, C. J., Feng, S., Caro, E., Jacob, Y., Michaels, S. D., & Jacobsen, S. E. (2012). DNA methyltransferases are required to induce heterochromatic re-replication in *Arabidopsis*. *PLoS genetics*, 8(7), e1002808.

In addition, in Fig. 4g and 4i, it may not be necessary to use the same scale for different marks/Input as long as Col and *atx5 atx6* share the same, in Fig. 4i the input is too low to judge if there is a difference in DNA content between Col and *atx5 atx6*.

Response: Thanks for sharing this idea.

We have changed the scales accordingly and listed the new figure (new Fig. 4g). In the new figure, it is evident that the input for the two PCGs (protein-coding genes) in *atx5 atx6* remains the same as that in Col-0. Moreover, the elevated RAD51 signal over PCGs in Col-0 (protein-coding genes) co-occurs with the higher expression of PCGs.

New Fig. 4f, g IGV files of normalized CHIP signals (RPKM) of H3K9me2, H3K27me1, RAD51, RPA1A, H3K27ac, and H3K4me3 on a 45S rDNA locus on chr2 (Fig. 4f) and selected PCGs (4g).

17. Line 298-303. It is not clear what's the purpose of mentioning *mbd9 atx5 atx6* and *sac3b atx5 atx6* mutants. To support that unstable DNA in *atx5 atx6* requires RAD51 and RPA1A to maintain genome integrity, the authors need to show that loss of RAD51 or RPA1A in *atx5 atx6* causes even more unstable or copy numbers of DNA, but results in reference 15 (Davarinejad et al., 2022, Science) do not seem to support this idea.

Response: Thanks for the comment. The purpose of mentioning *mbd9 atx5 atx6* is to support the concept that lower H3K27me1 leads to aberrant replication of heterochromatin amplification including rDNA. Our previous results summarize that reduced H3K27me1 over the heterochromatin region in *atx5 atx6* will induce heterochromatin amplification including rDNA. If this is true, then in the triple mutants with restored H3K27me1 level to that in Col-0, the mutants should lose the heterochromatin amplification including rDNA. According to Fig. 4E in reference 14 (Potok et al., 2022, PNAS), the mutations of MBD9 and SAC3B restore the H3K27me1 level in *atx5 atx6* to that in Col-0. In our hands, the mutations of *MBD9* suppress the heterochromatin amplification in *atx5 atx6*, and mutations of *MBD9* and *SAC3B* restored the elevated level of rDNA in *atx5 atx6* back to that in Col-0 (new Fig. 1c and

Extended data Fig. 11a). Thus, our results are well in line with previously published data and support the idea that H3K27me1 serves as a negative factor to regulate the replication of 45S rDNA.

Potok, M.E., Zhong, Z., Picard, C. L., Liu, Q., Do, T., Jacobsen, C. E., Sakr, O., Naranbaatar, B., Thilakarathne, R., Khnkoyan, Z., Purl, M., Cheng, H., Vervaeet, H., Feng, S., Rayatpisheh, S., Wohlschlegel, J. A., O'Malley, R. C., Ecker, J. R., and Jacobsen, S. E. The role of ATXR6 expression in modulating genome stability and transposable element repression in Arabidopsis. *Proc Natl Acad Sci U S A* 119 (2022).

Figure 3G of reference 15 (Davarinejad et al., 2022, Science) clearly showed that *rad51 atrx5 atrx6* has a higher robust CV than that in *atrx5 atrx6*, suggestive of increased DNA content. Please refer to our reply to the question #2.

However, we do not have data to test whether mutation of RPA1A will affect the heterochromatin amplification. To be more accurate, we would change the sentence to “45s rDNA loci among other unstable DNA segments in *atrx5 atrx6* recruit RAD51 and RPA1A to repair the increased DNA damage”.

18. Extended Data Fig. 11b, c. Please indicate how many protein-coding genes are enriched with RAD51 and RPA1A, and why RAD51-enriched top 3000 genes but not all RAD51-enriched genes were used for GO analysis.

Response: Thanks for the comment.

To comprehensively analyze the RAD51 and RPA1A signal distribution, we used the union of RAD51 or RPA1A enriched peaks as our references to perform all analysis. We used the union of all RAD51 enriched protein-coding genes from mock-treated Col-0 and *atrx5 atrx6*, the total number is 6988 genes. The website we used to perform GO analysis only takes up to 3000 genes. So we ranked the RAD51-enriched genes by *P*-value of peak calling and perform GO analysis with the top 3000 genes with low *P*-value. RPA1A enriched genes from mock-treated Col-0 and *atrx5 atrx6* is 467. Since RPA1A has other four orthologs in *Arabidopsis* (Aklilu et al., 2014, Nucleic Acids Res.), the number of RPA1A enriched peaks is lower than RAD51 enriched peaks.

Aklilu, B.B., Soderquist, Ryan S., and Culligan, Kevin M. Genetic analysis of the Replication Protein A large subunit family in Arabidopsis reveals unique and overlapping roles in DNA repair, meiosis, and DNA replication. *Nucleic Acids Research* 42, 3104-3118 (2014).

19. Fig. 4h. Please perform statistical analyses.

Response: Thanks for the comment. we have performed the Welch *t*-test, and the *P*-value is listed in the new Fig. 4h. For this figure, we aimed to investigate the RAD51 signal indeed associated with the transcription of genes, so we first selected the genes that showed a lower RAD51 signal ($\text{Log}_2\text{FC} < -0.5$, $P < 0.05$) in *atxr5 atxr6* than that in the Col-0 and checked the transcription levels of genes in *atxr5 atxr6* and Col-0. Then we selected the 959 that showed unchanged RAD51 signal between Col-0 and *atxr5 atxr6* ($-0.2 < \text{Log}_2\text{FC} < 0.2$) and check how the corresponding transcription levels of genes changed in *atxr5 atxr6*. The number of genes in the group circled by the red rectangle is 207, whereas the genes circled in the blue rectangle are 959. So we performed Welch's approximate *t*-test for two groups (Welch's approximate *t*-test is used to analyze the data when variances are not assumed equal).

New Fig. 4h, Violin plot shows the expression changes of transcripts from the loci with reduced and unchanged RAD51 ChIP signal in mock-treated *atxr5 atxr6* vs Col-0.

20. Line 305-334. If RAD51 is generally required for active transcription, why it is selectively enriched at genes in certain pathways? Please provide an explanation for this.

Response: Thanks for this insightful comment.

Recent studies reported a mechanism termed “transcription-associated homologous recombination repair” (TA-HRR) in human cells, pointing out the positive correlation between HRR and transcription. RAD52 promotes the processing of the R-loops and the formation of RPA, RAD51, and BRCA1 foci to protect the actively transcribed regions. They proposed that TA-HRR is responsible for around 5% of total DSBs (DNA double-strand breaks) (Yasuhara et al., 2018, cell). Interestingly, during SA-triggered innate immune response, RAD51 can directly bind to promoter elements of defense genes and enhance the gene expression in a BRCA2 and SA-dependent manner reference 27 (Wang, 2010, PNAS). Moreover, RAD17 and ATR are required to enhance the expression of the SA-activated genes and effective immune response reference 28 (Yan et al., 2013, Molecular cell).

Yasuhara, T., Kato, R., Hagiwara, Y., Shiotani, B., Yamauchi, M., Nakada, S., Shibata, A., and Miyagawa, K. (2018). Human Rad52 Promotes XPG-Mediated R-loop Processing to Initiate Transcription-Associated Homologous Recombination Repair. *Cell* 175, 558-570 e511.

Yan, S., Wang, W., Marques, J., Mohan, R., Saleh, A., Durrant, W. E., Song, J., and Dong, X. Salicylic acid activates DNA damage responses to potentiate plant immunity. *Mol Cell* 52, 602-10 (2013).
Wang, S., Durrant, W. E., Song, J., Spivey, N. W., and Dong, X. Arabidopsis BRCA2 and RAD51 proteins are specifically involved in defense gene transcription during plant immune responses. *Proc Natl Acad Sci U S A* 107, 22716-21 (2010).

21. Line 338-340. If HRR factors (presumably refer to RAD51 and RPA1A) are sequestered at unstable DNA regions in *atrx5 atrx6*, why their enrichment was not increased at TEs, as TEs are enriched at heterochromatin that becomes unstable in *atrx5 atrx6*. Please provide an explanation for this.

Response: Thanks for the comment.

We normally performed virus inoculation with the 3-week-old plant and collected the samples 14 days post-inoculation. However, the heterochromatin amplification phenotype decreases at a later stage (Fig. S2, Potok et al., 2022, PNAS).

In other words, the samples we collected were around 5-6 week-old plants. The heterochromatin amplification phenotypes in the rosette leaves in our collected samples were not as strong as the cotyledon at 12-day-old seedlings.

That being said, the input for RAD51 or RPA1A enriched TEs, different from that of ncRNAs and 45S rDNA, remains largely unchanged in *atrx5 atrx6* vs Col-0 (new Extended data Fig. 10 d, e). Thus, the DNA damage level over TEs in *atrx5 atrx6* is comparable to (or marginally different from) that in Col-0 as the DNA damage level is associated with the DNA copy number and sequence feature. Consequently, RAD51 and RPA1A signal over TEs remain largely unchanged in *atrx5 atrx6*.

New Extended Data Fig. 10 d, e, Distributions of normalized input reads (RPKM) of RAD51 (d) and RPA1A (e) on different regions of chromosomes in *atxr5 atxr6* vs Col-0.

Potok, M.E., Zhong, Z., Picard, C. L., Liu, Q., Do, T., Jacobsen, C. E., Sakr, O., Naranbaatar, B., Thilakarathne, R., Khnkoyan, Z., Purl, M., Cheng, H., Vervaeet, H., Feng, S., Rayatpisheh, S., Wohlschlegel, J. A., O'Malley, R. C., Ecker, J. R., and Jacobsen, S. E. The role of ATXR6 expression in modulating genome stability and transposable element repression in *Arabidopsis*. *Proc Natl Acad Sci U S A* 119(2022).

22. Line 341-344 and Extended Data Fig.12a. Please explain **how** representative samples were randomly collected at an earlier stage. Also, these results should be **normalized to input** as that in Fig. 3c, otherwise the reduced RAD51 enrichment on viral DNA could be due to the reduced viral DNA copy numbers in *atxr5 atxr6* compared with Col, as the authors also mentioned in line 344-345

Response: We thank the reviewer for the insightful questions.

First, the way of "representative samples" here refers to the harvest of a whole tray of 36 inoculated individuals, which conforms to one biological replicate. This is, there is no randomization or selection, but the harvest of the whole population. We corrected the text to indicate this.

Second, we have redone this analysis with three replicates and normalized the results to the virus input, the new results are listed below. These results indicated that a significant reduced association of RAD51 on viral genome but higher RAD51 signal over rDNA in *atxr5 atxr6* than that in Col-0 when samples were collected at an earlier stage. Please refer to the response to the main criticism by Reviewer #1.

New Fig. 5a. ChIP-qPCR assays showed that RAD51 enrichment was significantly increased over rDNA but decreased over viral DNA in *atxr5 atxr6* than that in Col-0 when the samples were randomly collected.

23. Line 349-352. Please explain **why** in Col the decreased signals of RAD51 and RPA1A mainly originated from the PCG loci rather than rDNA regions. DNA damage rates should be the same/random at PCG and rDNA in Col, and if virus hijacks RAD51 and RPA1A, how could it selectively take them from PCG but not rDNA in Col.

Response: Thanks for this extremely insightful comment.

First of all, virus hijacks RAD51 and RPA1 from both rDNA and PCGs loci. Careful comparison of the RAD51-ChIP-seq reads in the viral-infection condition vs mock treatment indeed revealed a subtle decrease over rDNA (417 RPM) but a much bigger decrease in PCGs loci (by 47216 RPM). This change was not visible due to the relatively large y-scale of rDNA. Importantly, the ChIP-seq analysis was based on the RPKM (internal normalization) and only reflected the relative changes of RAD51-binding patterns among the loci within one example and did not reflect the changes cross different samples (the original Fig. 5b). However, this reviewer's comment prompted us to re-analyze ChIP-seq based on internal contaminants (mitochondrial DNAs where RAD51 does not bind) that can serve as unbiased internal controls. The new analysis showed that the hijacking of RAD51 and RPA1A from both rDNA and PCGs by virus vs mock is more obvious than the patterns generated from the classic RPKM method (i.e., RAD51 signal over two 45S rDNA in Col-0 decreased by 15411; RAD51 signal over 9917 PCGs in Col-0 decreased by 406503. Also see presented in new Fig. 5c, d. and the reply to question #24 below).

New Fig. 5c Overview of RAD51 signal change in *atxr5 atxr6* and Col-0 upon virus inoculation. ncRNA represents for non-coding RNA loci. TEs represent transposable elements.

New Fig. 5d Distribution of ChIP-seq signals of RAD51, and RPA1A normalized on internal mitochondrial reads in different genome categories in mock-treated and virus-inoculated Col-0 and *atxr5 atxr6*.

Second, there is the difference in hijacking patterns from rDNAs vs PCGs loci, and such difference is due to the following three facts:

- 1) As Col-0 does not undergo heterochromatin amplification in a normal condition, the RAD51 signal over rDNA in Col-0 may represent only a basal level. By contrast, RAD51 has mainly distributed to PCGs loci as the pool of RAD51 ChIP-reads over PCGs is way more than the amount over rDNA. Such pool is already sufficient for the viral DNA amplification at the early stage (the raw reads mapping to the virus is only 233777 whereas the reads mapping to the host in the infected Col-0 is 12164781).
- 2) RPA and RAD51 are major players in transcription-associated homologous recombination repair” (TA-HRR). It has been found that transcription inhibitors including DRB could significantly reduce RPA and RAD51 foci formation without affecting their protein accumulation (Fig. 1, Yasuhara et al., 2018, Cell). During the TA-HRR process, these HRR factors will be depleted when the transcription is suppressed. Here virus infection counter-defenses the host immunity by suppressing the transcription of defense-related genes including Jasmonic acid and s-adenosylmethionine (SAM, the donor for DNA methylation). Suppression of transcription would release RAD51 and RPA1A (one ortholog of RPA in *Arabidopsis*) from the PCG loci. Consequently, the virus will readily pick up RAD51 that is laid off from PCGs of the *Arabidopsis* genome for the viral DNA amplification (new Extended data 12a, b).

New Extended Data Fig.12 a, b Viral inoculation suppressed the transcription of defense-related genes and deplete RAD51 signal from host.

Yasuhara, T., Kato, R., Hagiwara, Y., Shiotani, B., Yamauchi, M., Nakada, S., Shibata, A., and Miyagawa, K. (2018). Human Rad52 Promotes XPG-Mediated R-loop Processing to Initiate Transcription-Associated Homologous Recombination Repair. *Cell* 175, 558-570.

3) The virus infection can also induce the expression of RAD51 and RPA1A as such the reservoir of two proteins is further elevated (New Fig. 3d). But the RAD51 signal in inoculated Col-0 over 16865 enriched loci decreased by 369787, and RPA1A signal in inoculated Col-0 decreased by 131639, indicating that the elevated RAD51 in Col-0 upon virus inoculation was efficiently recruited to the viral genome to facilitate the viral DNA amplification. Combined the elevated expressed RAD51 and RPA1A and those were laid off from the transcription of PCGs will be quickly recruited onto the viral genome sufficient enough to support the viral replication.

New Fig. 3d Viral inoculation elevated the expression of RAD51 and RPA1A.

24. Line 369-373 and Fig. 5b. Although RAD51 enrichment at PCG in *atxr5 atxr6* was not reduced by viral infection like that in Col, a clear reduction in RAD51 enrichment at rDNA was observed in *atxr5 atxr6* but not in Col. Therefore, the current data cannot prove that there is an overall less loss of RAD51 on host genome in *atxr5 atxr6* compared with Col after virus infection, and this cannot be simply deduced by the higher RAD51 occupancy on DNA in *atxr5 atxr6* compared with Col as it already happens in mock due to unstable DNA and RAD51 overexpression. Moreover, in my opinion the reduction of RAD51 enrichment at rDNA

(heterochromatin) in *atxr5 atxr6* after infection rather implies that the RAD51 occupancy at rDNA (heterochromatin) in *atxr5 atxr6* does not contribute to its viral resistance.

Response: We think that our points might have been misunderstood. We should have clearly described the experiments and results in a better way.

First of all, the ChIP-seq analysis was based on the RPKM (internal normalization) and only reflected the relative changes of RAD51-binding patterns among the loci within one example and did not allow the direct comparison of RAD51 association with host chromatin across different samples and treatments (the original Fig. 5b).

However, this reviewer's comment prompted us to re-analyze ChIP-seq based on internal contaminants control (mitochondrial DNAs where RAD51 and RPA1A do not bind). As shown in the new Fig. 5c, d, the new ChIP-seq analysis clearly showed that there is more than "an overall less loss of RAD51 on host genome in *atxr5 atxr6* compared with Col-0 after virus infection". In fact, the overall RAD51 signal at 16865 loci including 9917 PCGs and the two 45S rDNA, and 630 ncRNA loci in *atxr5 atxr6* increased upon virus inoculation. By contrast, the RAD51 signal over the loci in Col-0 decreased. Thus, the new results (New Fig. 5c, d) clearly indicated that RAD51 occupancy at host genome including rDNA in *atxr5 atxr6* does contribute to its viral resistance.

Please also refer to the reply to the question #23.

25. Line 374-382 and Fig. 5f. Results in Fig. 5f should be performed with at least three biological replicates and statistical analysis should also be performed. Moreover, without clear proof that there is less loss of RAD51 on host genome in *atxr5 atxr6* compared with Col after virus infection, the reduced RAD51 enrichment at viral DNA could be caused by indirect reasons that affect virus infection in *atxr5 atxr6*.

Response: Thank you for this criticism.

The results in the original Fig. 5f is calculated as this way. We first calculated the ratio of the RAD51 or RPA1-ChIP-seq reads mapped to the viral genome to the total mapped reads (including reads mapped to virus and host) in the RAD51 or RPA1A ChIP-seq. Then calculated the ratio of the reads mapped to the viral genome to the total mapped reads in the DNA-sequencing from input samples. The IP efficiency is calculated as below:

$$\text{IP efficiency} = \frac{\text{ratio of reads mapping to virus from RAD51 ChIP Seq}}{\text{ratio of reads mapping to virus from Input}}$$

During the revision, we performed two new sets of ChIP-qPCR (one set the relative viral DNA amount in *atxr5 atxr6* is 12% of that in Col-0 (newly collected samples), the other set the viral DNA amount in *atxr5 atxr6* purposely increased to 80% of that in Col-0, which is previously ChIP-seq samples). The reason why we purposely increased the viral titers in *atxr5 atxr6* was listed in the line 370-373 in the new ms, The results are presented in the New Fig. 5a and Fig. 5b.

Please see the response to the major concern of reviewer #1.

Regarding “without clear proof that there is less loss of RAD51 on host genome in *atxr5 atxr6* compared with Col after virus infection”, **please refer to our reply to the question #24. Please also refer to our response to the main question by reviewer #1 (the interaction between RAD51 and viral protein Rep was attenuated in *atxr5 atxr6*, whereas the interaction between RAD51 and host HRR factors HOP2 increased in *atxr5 atxr6*).**

26. Line 383-386 and Fig. 5g-i. RAD51 is already overexpressed and more enriched at heterochromatin in *atxr5 atxr6* compared with Col, and even though this does not fix the heterochromatin amplification. One would expect that the overexpressed RAD51 (by 35S promoter) would also be more enriched at heterochromatin in *atxr5 atxr6* compared with Col. However, overexpressing RAD51 (by 35S promoter) caused similar effects on viral infection in Col and *atxr5 atxr6*, this probably again rather suggests that RAD51 enrichment at heterochromatin in *atxr5 atxr6* does not contribute to its viral resistance. In addition, different tags were fused with RAD51 and different transgenes were used in Col and *atxr5 atxr6* (likely resulting in varied RAD51 function and overexpression levels), making their direct comparison impossible.

Response: We think that our points might have been misunderstood!

As we showed in Figure 6 in the original text, the loading of RAD51 onto rDNA and defense-related genes in *atxr5 atxr6* depends on the RAD51 partners such as BRCA1, CYCB1, and HOP2 (DNA repair factors of *Arabidopsis*). Since the expression of the RAD51 factors (BRCA1, HOP2, and CYCB1) remains unchanged (or not coordinately accumulated) between *atxr5 atxr6* and *atxr5 atxr6; 35S-RAD51* lines, overexpressed RAD51 in *atxr5 atxr6; 35S:RAD51* will not be efficiently loaded to rDNA and defense-related genes. As a consequence, Rep will recruit the overexpressed RAD51 to the virus genome and promote the virus accumulation.

Please refer to the response to the question # 19 by Reviewer #1, regarding the reason for using different constructs of RAD51 in Col-0 and mutant backgrounds. For the virus infection assays with the transgenic plants carrying overexpressed RAD51 in Col-0 and *atxr5 atxr6* backgrounds, we only compared their viral phenotypes of tagged-RAD51 transgenic lines to each own's background, and did not perform cross-comparison in different backgrounds. Our results clearly confirmed that overexpressed RAD51 did promote the virus accumulation in both Col-0 and *atxr5 atxr6* backgrounds.

27. Line 397-399 and Fig. 6b. It is not clear how the HOP2-RAD51 Co-IP is performed. Based on Fig. 6b, it seems that HOP2 interacts with Rep but not RAD51.

Response: Thanks for the comment.

We are sorry for the confusion and careless mistake. We have repeated this experiment and validated that HOP interacts with RAD51, but not Rep. The result was listed in the new Fig. 7b

28. All results such as “percentage of symptomatic plants”, “relative amount of viral DNA” and “ChIP-qPCR” lack statistics, and some of them lack enough biological replicates.

Response: Thanks for the criticism. We have now done statistical analyses for most figures. We have also repeated experiments or recovered previous results to provide more replicates to support the key conclusions where replicates were not enough, the new results were listed in the new Fig. 1 f, Fig. 2f, Fig. 2g, Fig. 5a, Fig. 5b.

Minor concerns

1. Please indicate what is “dpi” in figure legends.

Response: dpi represents days post-inoculation. Corrected in the revision.

2. Line 147-149. The cited reference 32 describes the phenotypes of h3.3 but not h3.1 mutant.
Response: Thanks for the comment. we replaced the cited reference 32 with reference 10 and 16.

3. Extended Data Fig. 4b is not cited in the text.

Response: Corrected.

5. Line 233-234. Extended Data Fig. 7c probably should be 7b.

Response: Corrected.

6. Extended Data Fig. 10d is not cited in the text.

Response: Corrected.

Reviewer #3 (Remarks to the Author):

I read the manuscript of Wang et al. with great interest. The authors report on the very surprising fact that the atxr5/atxr6 double mutant is resistant to geminiviral infection. The double mutant is deficient in the deposition of H3.K27me1 to heterochromatin. The mutant phenotype results in genome instability, hyperrecombination, the activation of transposable elements, the amplification of heterochromatic regions and accumulation of DSBs.

As a consequence, factors involved in DNA repair are recruited to these genetically unstable regions, especially the strand exchange protein Rad51 and RPA1A, required for homologous recombination. Geminiviruses on the other side rely on both proteins for their replication and the authors demonstrate an interaction of both with the viral rep protein. Thus, the available pool of HR factors required for virus replication is titrated down in the mutants and the plants acquire resistance to infection.

This is a fascinating story and as far as I see the authors were able to sustain every single step in their proposed model by experiments. Besides other findings they could show that indeed less Rad51 and RPA1A are loaded on viral replicons in the double mutant than in WT.

Overexpression of Rad51 in the double mutant reduced resistance. Recruitment of Rad51 to the unstable genomic regions but not to the viral DNA depends on BRCA1, CYCB1;1 and HOP2. In their absence the resistance phenotype is reduced. I am really impressed by the wealth of data produced by the authors. I do not see any flaws, the paper is well written, all conclusions are justified. A very complete story that should be published without further ado.

Reviewer #1 (Remarks to the Author):

The manuscript went through an extensive revision and the authors have addressed most of the reviewers' concerns. The added data strengthened this complex, yet important work.

Reviewer #2 (Remarks to the Author):

The authors have addressed the majority of my questions, and I appreciate their efforts. However, I still have some concerns, which are listed below:

1. Comment 1 revolves around the importance of demonstrating that the wild type host plant could sacrifice its genome stability after virus infection for the suppression of viral replication, without relying on any other artificial treatments or mutations (UV, SA, atxr5;atxr6 mutation, H3.1S28A, etc.). Otherwise, I do not see the biological significance of the mechanism described in this manuscript for resisting viral infection under natural conditions. While I appreciate that the authors have moderated their tone in the revised manuscript, I still think that they could test whether there is any evidence of DNA damage or activation of the DDR following virus infection in wild type plants.

The SA treatment-induced DNA damage and increased resistance to viral infection could be correlative. SA may induce other pathways that suppress virus replication, independent of DNA damage.

2. Comment 4, Extended Data Fig. 4e and 4f are still labeled as "Mock".

3. Comment 5, If 18 genes were tested for Y2H, the authors should state "18" in the manuscript and not "20" (line 245).

4. Comment 11, Please indicate why multiple mapped reads were used for analysing RAD51 and RPA1A enrichment. Usually, only unique mapped reads are used for ChIP-seq data analysis, even for heterochromatin-enriched marks such as H3K27me1 and H3K9me2. More signals will likely be observed for RPA1A and RAD51 at heterochromatin when multiple mapped reads are used since heterochromatin is enriched with repetitive sequences.

5. Comment 16, The explanations provided by the authors are not entirely convincing. Indeed, RAD51 and RPA1A expression is induced in atxr5 atxr6 mutant, but the extra RAD51/RPA1A may distribute across the genome without strong preference, and the "increased" enrichment of RAD51/RPA1A at rDNA loci could just because there are more rDNA copies in atxr5 atxr6. This idea is actually proved by the authors' analysis with input for normalization. The authors claim that "Specifically, we are not questioning the affinity of protein binding to the nucleic acid and the relative binding difference of RAD51 or RPA1A within the host genome." But their conclusion is "RAD51 and RPA1A preferentially bind to ribosomal DNA (rDNA) in atxr5 atxr6" (Line 265)", which is not sufficiently supported by their data.

6. Comment 23 and 24, While the authors' efforts to re-analyse the ChIP-seq data are appreciated, the method section lacks details on how the data were analysed with mitochondrial DNAs for normalization. Additionally, it is worth noting that the ChIP-seq experiments were performed with nuclei extracts, which should have little to no mitochondrial contamination. Normally, a fixed amount of alien chromatin is added across samples for spike-in normalization, but in this case, it is unclear how the starting amounts of mitochondrial DNA were controlled to ensure it can serve as a proper reference.

Reviewer Comments:

Reviewer #1:

The manuscript went through an extensive revision and the authors have addressed most of the reviewers' concerns. The added data strengthened this complex, yet important work.

Reviewer #2 (Remarks to the Author):

The authors have addressed the majority of my questions, and I appreciate their efforts. However, I still have some concerns, which are listed below:

1. Comment 1 revolves around the importance of demonstrating that the wild type host plant could sacrifice its genome stability after virus infection for the suppression of viral replication, without relying on any other artificial treatments or mutations (UV, SA, *atxr5*;*atxr6* mutation, H3.1S28A, etc.). Otherwise, I do not see the biological significance of the mechanism described in this manuscript for resisting viral infection under natural conditions. While I appreciate that the authors have moderated their tone in the revised manuscript, I still think that they could test whether there is any evidence of DNA damage or activation of the DDR following virus infection in wild type plants.

Response: Thanks very much for sharing this great idea, which we should have thought about! But much appreciated!

With this insightful advice, we revisited our RNA-seq data. Indeed, virus infection can trigger DNA damage and activation of the DNA damage response (DDR) pathway as at least 18 DDR genes are activated upon the virus inoculation (New Extended data Fig. 6f). We also mined published microarray data of geminivirus-inoculated plants (Ascencio-Ibañez, et al., Plant Physiol. 2008. 148-436-454), and found that CaLCuV-inoculation significantly enhances the expression of 17 DDR genes with 9 overlapping with 18 genes in New Extended data Fig. 6f). Thus, the two independent assays from two labs clearly show that CaLCuV indeed triggers DNA damage or activation of the DDR pathway upon the virus infection in wild-type plants.

To clarify the physiological connection, here we will add the sentence “Of note, the 18 of 19 DDR genes and 16 of 17 genes related to DNA repair were upregulated in Col-0 upon virus inoculation (Extended Data Fig. 6f). A similar result was also observed in an early microarray assay³⁰. Importantly, the expression of DDR genes induced by virus inoculation was further enhanced in *atxr5 atxr6* in lines 222-226” during this revision.

For the convenience for readers, we have extracted the 19 DDR genes in Fig. 2b and drawn a subset heatmap (New Extended data Fig. 6f). We can observe the activation of 18 DDR genes in Col-0 after virus infection.

New Extended data Fig. 6f Heat map shows the accumulation change of 19 DNA damage response genes in Fig. 2b.

The SA treatment-induced DNA damage and increased resistance to viral infection could be correlative. SA may induce other pathways that suppress virus replication, independent of DNA damage.

Response: Thanks for sharing this idea. We have tried the effect of DNA damage reagent bleomycin (BLM) and found that the application of BLM (4 $\mu\text{g/ml}$, concentration adopted from Yan et al., 2013) can enhance plant immunity against geminivirus infection. We have added the results in the Extended data Fig. 14a, b during this revision. However, the price for this chemical (228\$/100mg) and its life-threatening feature make it unsuitable for application on agricultural products.

New Extended data Fig. 14a, b a, Representative phenotypes of CaLCuV-inoculated plants in the indicated genotypes and treatments at 15 dpi. **b**, Percentages of symptomatic plants induced by CaLCuV infection in indicated backgrounds at 11, 13, and 15 dpi.

Yan, S., Wang, W., Marques, J., Mohan, R., Saleh, A., Durrant, W. E., Song, J., and Dong, X. Salicylic acid activates DNA damage responses to potentiate plant immunity. *Mol Cell* 52, 602-10 (2013).

2. Comment 4, Extended Data Fig. 4e and 4f are still labeled as “Mock”.

Response: Thanks for the comment. The data for Extended Data Fig. 4e and 4f were mined from previous work GSE77735 (Hale et al., 2016), the samples were collected without any treatment. Extended Data Fig. 4e and 4f are labeled as “Untreated” in the revision.

Hale, C.J., Potok, M. E., Lopez, J., Do, T., Liu, A., Gallego-Bartolome, J., Michaels, S. D., and Jacobsen, S. E. Identification of Multiple Proteins Coupling Transcriptional Gene Silencing to Genome Stability in *Arabidopsis thaliana*. *Plos Genetics* 12(2016).

3. Comment 5, If 18 genes were tested for Y2H, the authors should state “18” in the manuscript and not “20” (line 245).

Response: Thanks for the comment. Corrected in the revision.

4. Comment 11, Please indicate why multiple mapped reads were used for analysing RAD51 and RPA1A enrichment. Usually, only unique mapped reads are used for ChIP-seq data analysis, even for heterochromatin-enriched marks such as H3K27me1 and H3K9me2. More signals will likely be observed for RPA1A and RAD51 at heterochromatin when multiple mapped reads are used since heterochromatin is enriched with repetitive sequences.

Response: Thanks for the comment. We should have emphasized the reason why we used reads maps to multiple locations in the genome to perform ChIP-seq analysis in our manuscript.

Briefly, Jacob et al., 2010 analyzed their ChIP-seq library with two strategies: 1) keeping reads that uniquely mapped to the genome; 2) keeping the total mapping reads that include uniquely mapped and the reads that mapped to the multiple locations of the genome. They found that strategy 1 resulted in no signals in the majority of re-replicating regions in *atxr5 atxr6*.

Thus, when the study focused on the over-replicated elements in *atxr5 atxr6* compared to those in Col-0 such as rDNA, TEs, and other repetitive sequences in heterochromatic regions, scientists tended to use multiple-mapped reads for mapping to the reference genome. For instance, Jacob et al.,(2010) uses this way to analyze ChIP-seq libraries of H3K27me1 and H3-enriched DNA, and so did Pontvianne et al.,(2012) with their ChIP-seq libraries prepared with anti-H3K27me1, anti-H3K9me2, and anti-Pol I-enriched DNA.

Here we compared the two specific 45S rDNA loci and found that the majority of reads corresponding to the 45S rDNA were filtered out when unique mapped reads were aligned to the genome. Our conclusion is consistent with that of Dr. Jacob's lab. Clearly, the 45S rDNA has many repetitive sequences, and majority reads would have been filtered out when the unique reads are aligned to the reference genome. Combined all together, we performed the ChIP-seq analysis with reads mapped to multiple locations.

We added one sentence "since the analysis with uniquely mapped reads barely detected signals in the majority over-replicated regions that harbor many repetitive sequences in *atxr5 atxr6*" with a citation of Jacob's paper in the revision (line 925-927).

Jacob, Y., Stroud, H., Leblanc, C., Feng, S., Zhuo, L., Caro, E., Hassel, C., Gutierrez, C., Michaels, S.D., and Jacobsen, S.E. Regulation of heterochromatic DNA replication by histone H3 lysine 27 methyltransferases. *Nature* 466, 987-U117 (2010).

Pontvianne, F., Blevins, T., Chandrasekhara, C., Feng, W., Stroud, H., Jacobsen, S. E., Michaels, S. D., and Pikaard, C. S. Histone methyltransferases regulating rRNA gene dose and dosage control in Arabidopsis. *Genes Dev* 26, 945-57 (2012).

5. Comment 16, The explanations provided by the authors are not entirely convincing. Indeed, RAD51 and RPA1A expression is induced in *atxr5 atxr6* mutant, but the extra RAD51/RPA1A may distribute across the genome without strong preference, and the “increased” enrichment of RAD51/RPA1A at rDNA loci could just because there are more rDNA copies in *atxr5 atxr6*. This idea is actually proved by the authors’ analysis with input for normalization. The authors claim that “Specifically, we are not questioning the affinity of protein binding to the nucleic acid and the relative binding difference of RAD51 or RPA1A within the host genome.” But their conclusion is “RAD51 and RPA1A preferentially bind to ribosomal DNA (rDNA) in *atxr5 atxr6*” (Line 265), which is not sufficiently supported by their data.

Response: Thanks for the comment. We should have explained our point clearly.

The reads mapped to the two 45S rDNA loci in *atxr5 atxr6* account for 17.09% of the reads corresponding to all 16865 RAD51 enriched peaks, whereas the two 45S rDNA loci in Col-0 account for 6.17% of the reads corresponding to all 16865 RAD51 enriched peaks. A similar pattern has been observed in the RPA1A ChIP-seq. The point we tried to emphasize is that more RAD51 and RPA1A, which are downstream factors of

homologous recombination repair, are recruited by the over-replicated 45S rDNA of *atx5 atx6*. This pattern is reminiscent of a recent discovery that more TONSOKU (TSK, one key factor to initiate homologous recombination repair) is recruited to resolve stalled/broken replication forks caused by re-replication and maintain genomic stability (Davarinejad et al., 2022).

Our ChIP-qPCR detected higher ChIP signals on rDNA in *atx5 atx6* than that in Col-0 ($P < 0.05$, Fig. 5b). However, we agreed that it is not sufficient to conclude that RAD51 and RPA1A preferentially bind to ribosomal DNA (rDNA) in *atx5 atx6*. We re-worded “RAD51 and RPA1A preferentially bind to ribosomal DNA (rDNA) in *atx5 atx6*” to “Over-replicated rDNA in *atx5 atx6* recruited more RAD51 and RPA1A to the loci” (line 267 in the revision).

Fig. 5b. ChIP-qPCR assays show that the loading of RAD51 over rDNA was significantly increased but decreased over viral DNA in *atx5 atx6* than that in Col-0 when the ratios of symptomatic plants were comparable in two backgrounds.

Davarinejad, H., Huang, Y. C., Mermaz, B., LeBlanc, C., Poulet, A., Thomson, G., Joly, V., Munoz, M., Arvanitis-Vigneault, A., Valsakumar, D., Villarino, G., Ross, A., Rotstein, B. H., Alarcon, E. I., Brunzelle, J. S., Voigt, P., Dong, J., Couture, J. F., and Jacob, Y. The histone H3.1 variant regulates TONSOKU-mediated DNA repair during replication. *Science* 375, 1281-1286 (2022).

6. Comment 23 and 24, While the authors' efforts to re-analyse the ChIP-seq data are appreciated, the method section lacks details on how the data were analysed with mitochondrial DNAs for normalization. Additionally, it is worth noting that the ChIP-seq experiments were performed with nuclei extracts, which should have little to no mitochondrial contamination. Normally, a fixed amount of alien chromatin is added across samples for spike-in normalization, but in this case, it is unclear how the starting amounts of mitochondrial DNA were controlled to ensure it can serve as a proper reference.

Response: Thanks for bringing up several interesting points regarding ChIP-seq analysis.

Typically, scientists use the internal normalization based on total reads for ChIP-seq analysis. For instance, two standardized labs of Drs. Steven E. Jacobsen (Gallego-Bartolome, et al., 2019. Cell 176, 1068, Ichino et al., 2021. Science 372, 1434) and Robert Martienssen (Lee, et al., 2023. Cell 10.1016/j.cell.2023.08.001. among many other long lists). This type of normalization could reflect the relative increase and decrease of ChIP-seq signals within samples.

2) For comparison across samples, computational scientists developed two strategies to normalize the ChIP-seq signal: First, resemble the “housekeeping” approach with signals from loci that do not change between samples. For instance, Allhoff, M. et al. developed one brilliant method of normalization on the selected housekeeping genes (2016. NAR). This method has been well accepted by the field, and used in significant publications exemplified by Eckert., et al., 2019. Nature. <https://doi.org/10.1038/s41586-019-1173-8>; Willcockson, et al., 2020. Nature. <https://doi.org/10.1038/s41586-020-3032-z>.

Second, to apply some reference controls (spike-in control). While some people used this method happily, we should share our prior experience with this reviewer: 1) one could not find a good dosage (or amount) of the spike-in that could cover the multiple latitudes of signals from different loci in the ChIP-seq for proper comparison; 2) one does not know the sequences preference/bias for ligation of the specific alien controls into adaptors. In fact, when the Pro. Xiuren Zhang was a postdoc in Drs. Nam-Hai Chua and Thomas Tuschl’s labs, he and his colleagues had thoroughly compared the spiking-in methods in different sequencing methods (i.e. RNA-seq, small-RNA-seq, and ChIP-seq). The conclusion was that using the internal contaminants or housekeeping genes is way better than using the spike-in method.

Specific for our ChIP-seq analysis, we consider the mitochondrial DNA as “housekeeping genes” to normalize the ChIP signals based on two facts. First, the ratio of reads mapping to mitochondria in our input samples varies from 1.92% to 2.11% of total reads. The relative stabilized ratios of the mitochondria reads among samples were also comparable with earlier reports (i.e., reads mapping to mitochondria in the input samples varies from 2.00% to 2.59%) (Ichino et al., 2021). Second, we observed a reduced ChIP-seq signal of RAD51 (and RPA1A but to a lesser extent) mapping to mitochondria relative to the input signal. These results indicated that the IP process did enrich the RAD51 and RPA-bound loci in the nuclei from mitochondria. Furthermore, none of the 16865 RAD51-enriched or the 1675 RPA1A-enriched peaks are located in the mitochondria, which suggests that the ChIP-seq of RAD51 and RPA1A worked well.

Regarding to this note: “Additionally, it is worth noting that the ChIP-seq experiments were performed with nuclei extracts, which should have little to no mitochondrial contamination”.

We compared our ChIP-seq data with RAD51 and RPA1A with the ChIP-seq data with MBD5/6 and H3K9me2/3 from two distinguished labs. The ChIP-seq signals mapping to mitochondria are comparable even using different antibodies—so, a low amount of reads mapping to contaminants are quite normal (see table below for the comparison of IP contaminants across different labs).

In summary, to better explain our ChIP-seq analysis, we have added the sentence “Normalization of the ChIP-seq signals was performed as previously described⁷³ with mitochondrial DNA as a reference. Be noted that the reads mapping to mitochondria were relatively stabilized from 1.92% to 2.11% of total reads among our input samples and thus could serve as internal “housekeeping” controls. Furthermore, none of 16865 RAD51-enriched or 1675 RPA1A enriched peaks were mapped into the mitochondrial genome, indicative of successful enrichment of RAD51 and RPA1A-bound chromatin in nuclei.” in line 937-942 during revision.

Ichino, L., Boone, B. A., Strauskulage, L., Harris, C. J., Kaur, G., Gladstone, M. A., Tan, M., Feng, S., Jami-Alahmadi, Y., Duttke, S. H., Wohlschlegel, J. A., Cheng, X., Redding, S., and Jacobsen, S. E. MBD5 and MBD6 couple DNA methylation to gene silencing through the J-domain protein SILENZIO. *Science* (2021).

Allhoff, M., Sere, K., J, F.P., Zenke, M., and I, G.C. (2016). Differential peak calling of ChIP-seq signals with replicates with THOR. *Nucleic Acids Res.* 44, e153.

Ratio: Mitochondria reads/Total reads

sample_name	Ratio
atxr56_ino_rep1_input	2.00%
atxr56_ino_rep2_input	2.09%
atxr56_ino_rep1_RAD51_mito	0.55%
atxr56_ino_rep1_RPA1A_mito	1.60%
atxr56_ino_rep2_RAD51_mito	0.56%
atxr56_ino_rep2_RPA1A_mito	1.53%
atxr56_mock_rep1_input	2.11%
atxr56_mock_rep2_input	2.03%
atxr56_mock_rep1_RAD51_mito	0.89%
atxr56_mock_rep1_RPA1A_mito	1.55%
atxr56_mock_rep2_RAD51_mito	0.77%
atxr56_mock_rep2_RPA1A_mito	1.52%
Col.0_ino_rep1_input	1.92%
Col.0_ino_rep2_input	1.95%
Col.0_ino_rep1_RAD51_mito	0.70%
Col.0_ino_rep1_RPA1A_mito	1.63%
Col.0_ino_rep2_RAD51_mito	0.80%
Col.0_ino_rep2_RPA1A_mito	1.48%
Col.0_mock_rep1_input	1.96%
Col.0_mock_rep2_input	1.96%
Col.0_mock_rep1_RAD51_mito	0.68%
Col.0_mock_rep1_RPA1A_mito	1.42%
Col.0_mock_rep2_RAD51_mito	0.74%
Col.0_mock_rep2_RPA1A_mito	1.46%

Our data

sample_name	Ratio
GSM5026044_MBD5_WT_L23	2.20%
GSM5026045_MBD5_RR_L23	2.34%
GSM5026046_MBD6_WT_L23	2.00%
GSM5026047_MBD6_RR_L23	2.59%
GSM5026048_Col0_L23	2.44%
GSM5026049_MBD5F_line5_dnai4plus_L26c_L23	2.80%
GSM5026050_MBD5F_line5_dnai4minus_L26c_L23	2.25%
GSM5026051_MBD6F_line2_dnai4plus_L26c_L23	2.29%
GSM5026052_MBD6F_line2_dnai4minus_L26c_L23	2.14%
GSM5026053_Col0_L26c_L23	2.50%
GSM5026054_DNAJ4F_mbd56plus_1_L26b_L23	2.60%
GSM5026055_DNAJ4F_mbd56minus_1_L26b_L23	2.60%
GSM5026056_Col0_correct_L26b_L23	3.34%
GSM5026057_DNAJ4_WT_L30_L30	2.33%
GSM5026058_DNAJ4_H94Q_L30_L30	2.14%
GSM5026059_Col0_FLAG_L30_L30	2.80%

Ichino, L. *et al.* (2021) MBD5 and MBD6 couple DNA methylation to gene silencing through the J-domain protein SILENZIO. *Science* 372, 1434.

sample_name	Ratio
GSM4914863_amiR.fie.1_ChIP	0.28%
GSM4914864_amiR.fie.2_ChIP	0.30%
GSM4914865_amiR.fie.3_ChIP	0.29%
GSM4914866_Ctrl.1_ChIP	0.07%
GSM4914867_Ctrl.2_ChIP	0.06%
GSM4914868_Ctrl.3_ChIP	0.06%
GSM4914869_GFP.FIE.1_ChIP	0.12%
GSM4914870_GFP.FIE.2_ChIP	0.12%
GSM4914871_GFP.FIE.3_ChIP	0.13%
GSM4914872_NST.1_ChIP	0.19%
GSM4914873_NST.2_ChIP	0.20%
GSM4914874_NST.3_ChIP	0.18%
GSM4914875_Col.0.H3K27.1	0.07%
GSM4914876_Col.0.H3K27.2	0.08%
GSM4914877_Col.0.H3K27.3	0.08%
GSM4914878_tor.H3K27.1	0.20%
GSM4914879_tor.H3K27.2	0.20%
GSM4914880_tor.H3K27.3	0.20%
GSM4914881_Col.0.H3K9.1	1.17%
GSM4914882_Col.0.H3K9.2	1.21%
GSM4914883_Col.0.H3K9.3	1.16%
GSM4914884_tor.H3K9.1	0.98%
GSM4914885_tor.H3K9.2	0.87%
GSM4914886_tor.H3K9.3	0.91%

Ye, R.Q. *et al.* (2022) Glucose-driven TOR-FIE-PRC2 signalling controls plant development. *Nature* 609, 986.

Table: Comparison of IP contaminants among different labs shows that a low amount of mitochondrial contaminants is very normal during the ChIP-seq analysis. The ratios of mitochondria-mapped reads from ChIP-seq with different antibodies are framed in red.

Reviewer #2 (Remarks to the Author):

The authors have addressed my concerns. However, I recommend that they include the ratio of mitochondrial reads and provide specific details about the spike-in calculation in the supplementary materials. This additional information would enhance the clarity and enable readers to better understand and replicate their analysis.

Reviewer Comments:

Reviewer #2 (Remarks to the Author):

The authors have addressed my concerns. However, I recommend that they include the ratio of mitochondrial reads and provide specific details about the spike-in calculation in the supplementary materials. This additional information would enhance the clarity and enable readers to better understand and replicate their analysis.

Response: The ratio of mitochondrial reads was provided as Supplementary Table 3 in Supplementary Information File. The detailed normalization process was now provided in Methods part (Lines 914-920).